# Pull-back Geometry of Persistent Homology Encodings

**Shuang Liang**                                                    *liangshuang@g.ucla.edu*
*Department of Statistics & Data Science*
*UCLA*

**Renata Turkeš**                                          *renata.turkes@uantwerpen.be*
*Department of Mathematics & Computer Science*
*University of Antwerp*

**Jiayi Li**                                                              *jiayi.li@g.ucla.edu*
*Department of Statistics & Data Science*
*UCLA*

**Nina Otter**                                             *nina-lisann.otter@inria.fr*
*DataShape, Inria-Saclay;*
*Laboratoire de Mathématiques d'Orsay, Université Paris-Saclay*

**Guido Montúfar**                                          *montufar@math.ucla.edu*
*Departments of Mathematics and Statistics & Data Science, UCLA;*
*Max Planck Institute for Mathematics in the Sciences*

**Reviewed on OpenReview:** *https://openreview.net/forum?id=7yswRA8zzw*

## Abstract

Persistent homology (PH) is a method for generating topology-inspired representations of data. Empirical studies that investigate the properties of PH, such as its sensitivity to perturbations or ability to detect a feature of interest, commonly rely on training and testing an additional model on the basis of the PH representation. To gain more intrinsic insights about PH, independently of the choice of such a model, we propose a novel methodology based on the pull-back geometry that a PH encoding induces on the data manifold. The spectrum and eigenvectors of the induced metric help to identify the most and least significant information captured by PH. Furthermore, the pull-back norm of tangent vectors provides insights about the sensitivity of PH to a given perturbation, or its potential to detect a given feature of interest, and in turn its ability to solve a given classification or regression problem. Experimentally, the insights gained through our methodology align well with the existing knowledge about PH. Moreover, we show that the pull-back norm correlates with the performance on downstream tasks, and can therefore guide the choice of a suitable PH encoding.

*Keywords:* Persistent homology, data representation, Jacobian spectrum, pull-back geometry, sensitivity analysis

## 1 Introduction

Persistent homology (PH) is a well-established technique in applied and computational topology (Carlsson, 2009; Oudot, 2015). At its core, PH seeks to create *representations* of data that highlight topological aspects. Recently, they have been found to also capture purely geometric aspects of the data, such as curvature (Collins et al., 2004; Bubenik et al., 2020) and convexity (Turkeš et al., 2022). PH representations have been used particularly in applications where multiscale homological features can be expected to capture relevant information, such as in the prediction of biomolecular properties (Cang & Wei, 2017; Cang et al.,

2018; Wang et al., 2020), quantification of similarity in materials (Lee et al., 2017), medical imaging (Singh et al., 2023), or the analysis of tree-like brain artery structures (Bendich et al., 2016). PH also increasingly interfaces with machine learning (see, e.g., Carriere et al., 2020; Hensel et al., 2021). Beside serving as a data representation technique, PH has been used, for instance, to investigate the decision boundaries of neural networks (Ramamurthy et al., 2019) or the transformations of data sets across the layers of a deep neural network (Naitzat et al., 2020). Other work has explored training neural networks to approximate PH features (Hofer et al., 2019; Montúfar et al., 2020; de Surrel et al., 2022), which can facilitate faster computations or serve as a basis for fine-tuning PH representations. Along this thread, a topology-encoding neural network based on PH was proposed by Haft-Javaherian et al. (2020).

In applications of PH to data classification or regression tasks, it is common to employ a model, such as a support vector machine (SVM) or a neural network, on the PH features. Such performance-based testing comes with two main drawbacks. Firstly, additional time and effort are needed to choose the classifier or regression model and tune their hyperparameters: this typically involves a grid search over all PH parameters, *but also* over models (e.g., SVM and neural networks), and over the model's parameters (e.g., regularization parameter of SVM, and a much larger list of hyperparameters for neural networks), which also requires training and testing for each combination of the three groups of parameter values. Secondly, the conclusions drawn regarding the effectiveness of PH are contingent upon the choice of the model and its specific parameters.

In this work, we propose a novel methodology aimed at gaining a more intrinsic understanding of PH encodings, irrespective of a particular classification or regression model. Here, a PH *encoding* denotes the mapping from data to a vectorized PH representation. We use the pull-back geometry induced by the PH encoding map to investigate its sensitivity to any particular data variation. A data variation is represented by a vector field in the data space, and can therefore be understood as an umbrella term that includes both perturbation vector fields reflecting data perturbations (e.g., translation or dilation of a point cloud) and gradient vector fields resulting from data features (e.g., a label indicating the presence of an anomaly or disease).

The Jacobian of an encoding mapping characterizes the behavior of the encoding in response to data variations. Specifically, the rank and eigenvectors of the Jacobian characterize the number of independent data variations and the most significant data variations captured by the encoding, respectively. The average pull-back norm of a vector field quantifies the sensitivity of PH to the corresponding data variation, and thus it helps assess to what extent PH is sensitive to a given perturbation or how effective it is at detecting a given feature (which in turns translates to its ability to solve a given problem). Furthermore, optimizing the average pull-back norm can guide the choice of a suitable PH encoding (choice of filtration, PH representation, and their parameters): one only needs to evaluate the pull-back norm over the different choices of PH parameters. This approach eliminates the need to train and select a classifier on top of PH features, at the same time providing insights that are more intrinsic to the underlying problem. Indeed, if the performance of a particular model on PH features is poor, one can hardly make any claims about the *PH representation itself* (since the problem could be that the model is poor). On the other hand, if the pull-back norm of the vector field is close to zero, we are more confident that the representation cannot recognize the given perturbation or feature. We provide a schematic diagram in Figure 1 that illustrates the pipeline of our proposed method and compares it with performance-based methods.

We center our attention on a widely used PH representation known as the persistence image (PI) (Adams et al., 2017), which is an image-like representation of the input data in terms of multiscale homological features. Our methodology, however, extends to other PH representations and, more broadly, to any differentiable encoding whose representation space can be endowed with a Riemannian manifold structure. In our experiments, we illustrate this generality by applying our approach to the PointNet encoding, a benchmark deep learning model for point clouds. We note that the insights about the (PH) encodings obtained through our approach depend on the specifics of the data set. Nonetheless, by evaluating different datasets one may be able to draw certain conclusions that hold with some generality: for instance, conclude that a particular encoding captures a particular feature in datasets of a particular type. This is an interesting prospect that can be facilitated by our proposed approach.

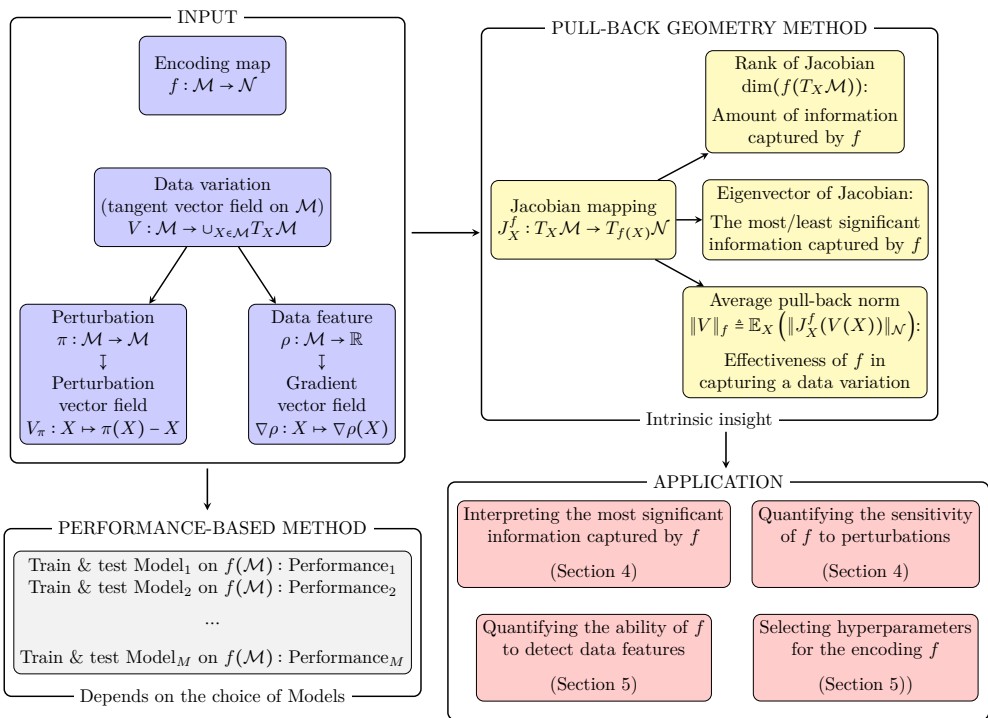

Figure 1: Schematic pipeline of our proposed method (comparing it with performance-based testing).

**Main contributions**

- We present an approach that can be used to investigate persistence images and their induced pull-back geometry on the manifold of input data sets in terms of the rank, spectrum, and eigenvectors of the Jacobian, as well as the pull-back norm of tangent vectors on the data manifold (Section 3).[1]

- We show how the above approach can be used to identify which data perturbations are captured by the encodings and which are ignored on given data sets. We also show how this facilitates an intrinsic comparison of PH encodings built with different filtrations. We experimentally demonstrate the insights gained via our approach align well with the existing knowledge about PH (Section 4).

- We show how the above approach can be used to quantify to what extent a PH encoding can recognize a data feature of interest on given a data set (*sex* feature in a data set of brain artery trees). We also show how this quantitative evaluation can guide the selection of hyperparameters for the encodings. Finally, we show that the pull-back norm is predictive for the performance on a downstream task (Section 5).

**Related work**  Our discussion falls within the general subject of interpreting a complex nonlinear map by investigating the effect that local input perturbations have on the output. This is conceptually related to topics such as sensitivity analysis (Saltelli, 2002), interpretable machine learning (Samek et al., 2021), sensitivity of outputs to input perturbations (Molnar et al., 2020), activation maximization (Simonyan et al., 2013), relevance propagation (Montavon et al., 2018), adversarial robustness (Engstrom et al., 2019), interpretable controls in implicit generative models (Härkönen et al., 2020), or function parametrizations in artificial neural networks (Du et al., 2019; Kornblith et al., 2019).

Hauser & Ray (2017) investigated the Riemannian geometry on the data manifold that is induced by representations learned using artificial neural networks and show, in particular, that the metric tensor can be

---

[1]Data and code developed in this research are available at `https://github.com/shuangliang15/pullback-geometry-persistent-homology`.

found by backpropagating the coordinate representations learned by the network. This shares similarities with our approach, as the induced geometry is essentially the pull-back geometry from representation space by the Jacobian map. Meller & Berkouk (2023) proposed a graph representation for neural networks using a singular value decomposition of the weight matrices. While they tackle the nonlinearity by studying linear maps contained in the nonlinear map consecutively, our emphasis lies in the local linear approximation of the nonlinear map.

There exist a few studies that investigate the sensitivity of PH representations to perturbations and their ability to recognize specific features, as we do in this work. For example, Turkeš et al. (2021) study the sensitivity of a number of PH representations to different types of transformations (such as rotation, translation, change of image brightness or contrast, as well as Gaussian and salt and pepper noise). However, the main method to assess sensitivity is the performance of an SVM trained on the representations of the original data and tested on the representations of data under transformations, and thus requires training and testing, and depends on the choice of a particular classifier. Bubenik et al. (2020) showed in theory and experiments that persistence landscapes can be used to detect curvature of an underlying set based on a sampled point cloud. Turkeš et al. (2022) conducted investigations towards identifying fundamental types of tasks for which PH representations might be most useful. They showed that beside curvature and number of holes, PH representations can be used for detecting convexity. However, they focus on three specific tasks (detecting number of holes, convexity, and curvature), whereas we study the alignment of the representations with *any* given feature defined on the space of input data sets. Moreover, in that work, the performance of PH is experimentally evaluated via the SVM accuracy, whereas our approach does not require training and testing of any model.

More generally, in the context of inverse problems in persistence theory, there are several lines of work that study conditions under which persistence diagram maps are surjective or injective; see the survey of Oudot & Solomon (2020) and references therein. Within this context, our work can be seen as providing a framework related to the study of the injectivity of specific PH encodings. The work of Xenopoulos et al. (2022) deals with local explainability using topological representations. In contrast, we deal with the representations. McGuire et al. (2023) measured the dissimilarity between representations learned by neural networks trained on PH encodings and networks trained on raw data. They experimentally demonstrate that networks learn considerably different representations when processing PH encodings instead of raw data. Rieck (2023) compared the expressivity of PH against the Weisfeiler-Lehman hierarchy of graph isomorphism tests, and explored the potential of PH to capture certain graph structures and characteristic properties. Finally, an important line of work in the study of PH encodings is concerned with developing computable notions of optimal representative cycles for persistent homology classes; see, e.g., the survey of Li et al. (2021).

## 2  Preliminaries on Persistent Homology

The key idea behind PH is to construct a filtration, i.e., a sequence of topological spaces by gradually adding simplices (vertices, edges, triangles, etc.) to the data, and to study the evolution of topological features (components, holes, voids, and higher-dimensional voids) across the filtration. In this section, we give a brief overview of the different filtrations that we consider in this work (Section 2.1), and the persistence image that we will use to represent PH information (Section 2.2).

### 2.1  Filtration

Let $X = \{x_i \in \mathbb{R}^D \mid i = 1, \ldots, N\}$ denote a point cloud consisting of $N$ points in $\mathbb{R}^D$. To build a filtration on $X$, we commonly construct a sequence of simplicial complexes on $X$.

A simplicial complex can be thought of as a space obtained by taking a union of vertices, edges, triangles, tetrahedra and higher-dimensional simplices. Formally, a collection $K \subset 2^X$ of subsets of $X$ is called a *simplicial complex* if $\sigma \in K$ and $\tau \subset \sigma$ imply $\tau \in K$. An element $\sigma$ in a simplicial complex is called a $(|\sigma| - 1)$-simplex, where $|\sigma|$ is the cardinality of $\sigma$. Specifically, one can think of 0-simplices as vertices (i.e., elements of $X$); 1-simplices as edges (i.e., pairs of elements of $X$); 2-simplices as triangles, etc. We note that according to this definition of a simplicial complex, not every element of $X$ is necessarily a 0-simplex; this is important for the types of filtrations that we consider in our work, such as the DTM filtration.

In the following we focus on simplicial complexes obtained as the clique complex of an $R$-neighborhood graph. The *clique complex of the R-neighborhood graph* on a point cloud $X$ consists of all subsets $\sigma$ of $X$ such that the distance between any pair of points in $\sigma$ is at most $R$:

$$\mathrm{Cl}(X, R) = \{\sigma \in 2^X \mid B_E(x_i, R) \cap B_E(x_j, R) \neq \varnothing, \forall x_i, x_j \in \sigma\},$$

where $B_E(x, R) = \{y \in \mathbb{R}^D \mid d_E(x, y) \leq R\}$ denotes the Euclidean $R$-ball centered at $x$ and $d_E$ denotes the Euclidean distance.

A *filtration* with respect to the clique complex is an indexed collection $\{K_r\}_{r \in \mathbb{R}^{\geq 0}}$ of subsets $K_r \subset \mathrm{Cl}(X, R)$ satisfying the condition that $K_{r_1} \subset K_{r_2}$ if $r_1 \leq r_2$. The construction of a filtration is equivalent to assigning a *filtration value* $\phi(\sigma)$ to each simplex $\sigma$ in $\mathrm{Cl}(X, R)$ in the following sense. Given $\{K_r\}_{r \in \mathbb{R}^{\geq 0}}$, one can define the filtration value for any simplex $\sigma$ as $\phi(\sigma) = \inf\{r : \sigma \in K_r\}$. Conversely, given a filtration value for every simplex, one can define the collection of subsets as $K_r = \{\sigma \in \mathrm{Cl}(X, R) : \phi(\sigma) \leq r\}$.

We will focus on some common filtrations built upon $\mathrm{Cl}(X, R)$:

1. The Vietoris-Rips filtration (Vietoris, 1927). This defines the filtration value for each simplex as its diameter: $\phi(\sigma) = \mathrm{Diam}(\sigma) = \max_{x,y \in \sigma} d_E(x, y)$.

2. The distance-to-measure (DTM) filtration (Anai et al., 2020). This defines the filtration value for each vertex as $\phi(\{x_i\}) = \frac{1}{K} \sum_{x_j \in \mathrm{KNN}(x_i)} d_E(x_i, x_j)$, where $\mathrm{KNN}(x_i)$ denotes the set of $k$ nearest neighbors of $x_i$ in $X$, so that the outliers have a large filtration function value and appear late in the filtration. Then it defines the filtration value for edges as $\phi(\{x_i, x_j\}) = \phi(\{x_i\}) + \phi(\{x_j\}) + d_E(x_i, x_j)/2$, and for simplices with degree $(|\sigma| - 1)$ greater than one as $\phi(\sigma) = \max_{x_i, x_j \in \sigma}\{\phi(\{x_i, x_j\})\}$.

3. The height filtration with respect to a hyperplane. Let $v \in \mathbb{R}^n$ be the unit normal vector of a hyperplane. The corresponding height filtration defines the filtration value of vertices as $\phi(\{x_i\}) = \langle x_i, v \rangle$, and the filtration value of any other simplices as $\phi(\sigma) = \max_{x \in \sigma}\{\phi(\{x\})\}$. We note that this means that, in order to capture features of interest, one needs to set the maximum length for an edge to be present in the height filtration (for details, see Appendix F.3).

## 2.2 Persistence image

The $k$-dimensional homology group, or $k$-dimensional homology, of a simplicial complex characterizes the $k$-dimensional holes in the complex. Each non-zero $k$-dimensional homology class in the $k$-dimensional homology group uniquely characterizes a $k$-dimensional hole. As we introduced earlier, the set $K_r$ includes more and more simplices as the parameter $r$ increases. In persistent homology, we are interested in how the homology groups of $K_r$ change as we vary the parameter $r$. By the matrix reduction algorithm (Edelsbrunner et al., 2002), one can identify a birth-death pair $(b, d) \in [0, R]^2$ for every non-trivial homology class that appears in the filtration. Roughly, the homology class first "appears" in $K_b$ and "persists" until $K_d$, degenerating to the trivial class afterwards. In addition to the bounded intervals, the matrix reduction algorithms gives infinite intervals $(b, \infty)$, which one can interpret as homology classes that appear at $K_b$ and do not degenerate to the trivial class for any filtration value considered. The persistence diagram (PD) is a summary of such information.

Formally, the $k$-dimensional PD is the multiset of birth-death pairs for all $k$-dimensional homology classes that appear in the filtration. Although the space of PDs can be endowed with a metric structure, PDs do not lend themselves to processing with techniques that require a Hilbert space structure, including support vector machines and principal component analysis (PCA) (Reininghaus et al., 2015). Hence, one often considers vector representations of PDs. The most commonly used ones include persistence images (Adams et al., 2017) and persistence landscapes (Bubenik, 2015).

In our work, we will focus on persistence images (PIs). Let PD be a $k$-dimensional persistence diagram in birth-death coordinates. One converts this to a multiset $\eta(\mathrm{PD})$ in birth-lifespan coordinates by applying the linear map $\eta(b, d) = (b, l)$ with $l = d - b$ to each birth-death pair $(b, d)$. Given an infinite interval $(b, \infty)$, in practice, one sets the death value to the maximum filtration value $R$. Given a kernel function $g_{(b,l)}(x, y)$ on

$\mathbb{R}^2$ and a weighting function $\alpha(b, l)$, the persistent surface is the function $\psi : \mathbb{R}^2 \to \mathbb{R}$ defined by

$$\psi(x, y) = \sum_{(b,l) \in \eta(\mathrm{PD})} \alpha(b, l) g_{(b,l)}(x, y).$$

A persistence image (PI) is a finite-dimensional representation of $\psi$ obtained as follows. One splits a subdomain of $\psi$ by a $P \times P$ grid of regions. Then the PI of resolution $P$ is the matrix whose $(i, j)$-th entry or pixel is the integration value of $\psi$ over the $(i, j)$-th region. Note that, since the death time $d$ cannot be smaller than the birth time $b$, the birth-death pairs $(b, d)$ always lie above the diagonal line, i.e., $\mathrm{PD} \subseteq \{(x, y) \in \mathbb{R}^2 : y \geq x \geq 0\}$. The transformed birth-lifespan pairs $(b, l)$ lie in the first quadrant, i.e., $\eta(\mathrm{PD}) \subseteq \{(x, y) \in \mathbb{R}^2 : x \geq 0, y \geq 0\}$ (see an illustration in Figure 2).

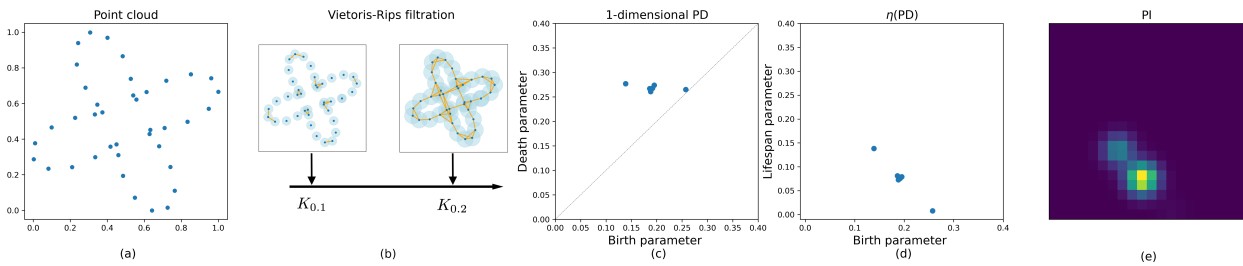

Figure 2: The pipeline for constructing a persistence image described in Section 2.2. From left to right: (a) input point cloud; (b) Vietoris-Rips filtration built on the point cloud; (c) 1-dimensional persistence diagram; (d) birth-lifespan pairs (transformed 1-dimensional persistence diagram); and (e) persistence image.

For constructing PIs, one needs to choose 1) the resolution $P$, 2) the kernel function $g_{(b,l)}(x, y)$ and its associated parameters, and 3) the weighting function $\alpha(b, l)$. One of the main difficulties in working with PIs is that there is no canonical way to choose these hyperparameters (Adams et al., 2017). Adams et al. (2017) studied the effects of PI parameters on the performance of certain classifiers ($K$-medoids classifiers) that take PIs as inputs. However, we note this approach heavily depend on the choice of downstream model. This motivates us to investigate what kind of information of the input data is intrinsically captured by the PI under different choices of the hyperparameters.

A motivation for considering PIs is that they provide differentiable PH representations,[2] and that, with an appropriate choice of metric, the space of PIs has a Euclidean structure, which simplifies computations (see Section 3.1). We will later consider derivatives of the mapping from input data to PIs. These can be obtained using existing automatic differentiation packages and libraries, such as topologylayer (Gabrielsson et al., 2020) and Gudhi (The GUDHI Project, 2020). We provide further details about this in Appendix C.

## 3 Methods: Sensitivity of PH Encoding to Data Variations

We consider an encoding map $f : \mathcal{M} \to \mathcal{N}$, where $\mathcal{M}$ is a space of point clouds and $\mathcal{N}$ is the space of persistence images (Section 3.1). We conceptualize input data variations (Section 3.2) and the resulting changes of the encoding output (Section 3.3).

### 3.1 Input space and output space

We let $\mathcal{M}$ be the space of point clouds in $\mathbb{R}^D$ that contain exactly $N$ points,

$$\mathcal{M} = \{X \subset \mathbb{R}^D : |X| = N\}.$$

A point cloud $X$ is a finite subset in $\mathbb{R}^D$ with cardinality $|X| = N$. A point cloud may be regarded as an unordered list of points, determined only up to permutation. It can also be regarded as a probability

---

[2]Precisely, the map from point clouds to PIs is generically differentiable (see a detailed discussion in Appendix C).

distribution on $\mathbb{R}^D$. Hence we can equip $\mathcal{M}$ with the 2-Wasserstein distance (see, e.g., Peyré et al., 2019):

$$d_W(X,Y) = \min_{\omega \in \Omega(X,Y)} \left( \sum_{x \in X} d_E^2(x, \omega(x)) \right)^{\frac{1}{2}}.$$

Here $\Omega(X,Y)$ denotes the set of bijections $\omega : X \to Y$ between the sets $X$ and $Y$, and $d_E$ denotes the Euclidean distance on $\mathbb{R}^D$. The 2-Wasserstein distance induces a metric topology on $\mathcal{M}$. Further, $\mathcal{M}$ can be endowed with a Riemannian manifold structure with $\dim(\mathcal{M}) \triangleq m = D \times N$. For simplicity of presentation, in the following we treat $\mathcal{M}$ as an Euclidean space. Nonetheless, our discussion is consistent with the Riemannian manifold structure and applies in that level of generality (see Appendix B). For a detailed discussion regarding the Riemannian structure and differential calculus on a Wasserstein space we refer the reader to Ambrosio et al. (Chapter 8, 2005) and Villani et al. (2009).

We let $\mathcal{N}$ be the space of persistence images of fixed resolution $P$. Thus we can interpret $\mathcal{N}$ as a submanifold embedded in $\mathbb{R}^{P \times P}$ and endowed with the canonical Euclidean distance, with $\dim(\mathcal{N}) \triangleq n = P^2$.[3] Here again, other choices of metric on the space of PIs are possible.[4]

## 3.2 Data variations

To characterize local variations of the input data, we consider tangent vectors on the data manifold. We conceptualize the intuitive concepts of perturbations and feature variations in terms of corresponding vector fields on the data manifold.

The *tangent space* at $X \in \mathcal{M}$, denoted $T_X\mathcal{M}$, is the vector space of all vectors emanating from $X$ and tangential to the data manifold $\mathcal{M}$. The dimension of $T_X\mathcal{M}$ is equal to the dimension of the data manifold, $\dim(T_X\mathcal{M}) = \dim(\mathcal{M})$. Each *tangent vector* $v \in T_X\mathcal{M}$ characterizes a local variation of a single point cloud $X$. A *vector field* specifies a variation for each point cloud in $\mathcal{M}$. More specifically, a vector field $V$ on $\mathcal{M}$ is a smooth map $V : \mathcal{M} \to \sqcup_X T_X\mathcal{M}$, assigning to each $X$ in $\mathcal{M}$ a tangent vector $V(X) \in T_X\mathcal{M}$.

A *perturbation* is a modification of a point cloud in the data manifold, e.g., by rotation or shearing. This can be described by a map $\pi : \mathcal{M} \to \mathcal{M}$ taking data $X \in \mathcal{M}$ to a perturbed data $\pi(X) \in \mathcal{M}$. The perturbation vector field $V_\pi$ associates to each $X \in \mathcal{M}$ a tangent vector $V_\pi(X)$ capturing the difference between $\pi(X)$ and $X$ (see Figure 3, left).[5]

**Definition 1** (Perturbation vector field). Let $\mathcal{M}$ be a manifold, $T_X\mathcal{M}$ the tangent space at $X \in \mathcal{M}$, and $\pi : \mathcal{M} \to \mathcal{M}$ a perturbation map. The perturbation vector field induced by $\pi$ is defined as

$$V_\pi : \mathcal{M} \to \sqcup_X T_X\mathcal{M}; \quad X \mapsto V_\pi(X) = \pi(X) - X.$$

A *feature* $\rho$ is a real-valued smooth function defined on the data manifold, $\rho : \mathcal{M} \to \mathbb{R}$, assigning a feature value to each $X$. Discrete-valued (categorical) features can be converted to continuous ones by considering probability distributions or logits of the feature values. For instance, the "cat-or-dog" feature can be converted to a continuous feature $\rho(X) = \text{Prob}(X \text{ is cat}) \in [0,1]$.

The gradient of a feature introduces a vector field on the data manifold. The gradient vectors point in the direction of steepest increase of the feature, with magnitude indicating the rate (see Figure 3, right).

**Definition 2** (Gradient vector field). Let $\mathcal{M}$ be a manifold, $T_X\mathcal{M}$ the tangent space at $X \in \mathcal{M}$, and $\rho : \mathcal{M} \to \mathbb{R}$ a real-valued feature. The gradient vector field of $\rho$ is the vector field on $\mathcal{M}$ defined as

$$\nabla\rho : \mathcal{M} \to \sqcup_X T_X\mathcal{M}; \quad X \mapsto \nabla\rho(X),$$

such that $\frac{\nabla\rho(X)}{\|\nabla\rho(X)\|} = \text{argmax}_{v \in T_X\mathcal{M}:\|v\|=1} |\frac{\partial}{\partial v}\rho(X)|$ and $\|\nabla\rho(X)\| = \max_{v \in T_X\mathcal{M}:\|v\|=1} |\frac{\partial}{\partial v}\rho(X)|$. Here $\frac{\partial}{\partial v}$ is the directional derivative along $v$.

---

[3]We note that if one considers the 1-Wasserstein distance on the space of PDs, and any of the $L_1, L_2$ or $L_\infty$ norms on the space of PIs, then PIs are known to be stable (Adams et al., 2017, Theorem 5). On the other hand, PIs, together with the $L_2$ norm, are unstable if one instead considers $p$-Wasserstein distances on the space of PDs (Adams et al., 2017, Remark 6).

[4]This can be of interest to try to establish more general stability results for PIs. An example would be a Wasserstein distance between persistence images assigning an appropriate cost to $b$ and $l$ directions.

[5]Figure 3 is a schematic illustration. It is not intended to imply that the kind of depicted point clouds indeed form a torus.

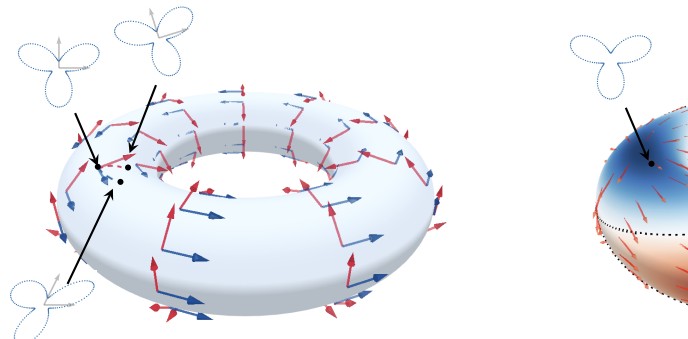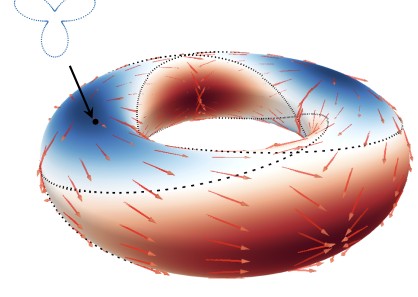

Figure 3: The space of point clouds forms a manifold, which in this figure is depicted as a torus; each point on this manifold is a point cloud. Left: vector fields on the data manifold correspond to variations of the point clouds; in this illustration, the red arrows correspond to "rotation" and the blue arrows to "shearing". Right: a continuous feature on the data manifold induces a gradient vector field; the figure illustrates a binary feature, where the dashed line is the class boundary, and the continuous feature value represents the probability of the data point belonging to the "red" class.

Definition 1 and Definition 2 are given for the case that $\mathcal{M}$ is a Euclidean space. We provide definitions of perturbation vector fields and gradient vector fields for the case of general Riemannian manifolds in Appendix B.3. Further, we provide details on how to estimate such vector fields using finite data sets in Appendices F.3 and F.4.

### 3.3 Encoding variations

Having characterized data variations in terms of vector fields, the next step is to describe the behavior of the encoding map $f$ in response to these variations. Specifically, we are going to introduce the average pull-back norm of a vector field to quantify the sensitivity of the encoding map to the corresponding data variation. At the outset of this subsection, we emphasize that whether or not it is desirable to have an encoding that is sensitive to a particular data variation depends on the specific practice scenario and whether this variation is perceived as valuable information or as noise that one would like to filter out in the encoding.

**Jacobian**   The *Jacobian* of an encoding map $f$, denoted by $J_X^f$, is a linear transformation between tangent spaces that characterizes the local behavior of $f$. While a tangent vector $v \in T_X\mathcal{M}$ describes one type of data variation at $X$, the image tangent vector $J_X^f(v)$ describes the resulting variation of the encoding $f(X)$,

$$J_X^f : T_X\mathcal{M} \to T_{f(X)}\mathcal{N}; \quad v \mapsto J_X^f(v).$$

We may write this linear transformation in terms of a Jacobian matrix $J_X^f \in \mathbb{R}^{n \times m}$ with respect to a basis. If there is no risk of confusion, we will omit the super-/subscripts $f$ and $X$. We provide a visualization for the Jacobian map in the left panel of Figure 4.

The rank of the Jacobian is the dimension of the image of $T_X\mathcal{M}$ under the Jacobian map, $\text{rank}(J_X^f) = \dim(J_X^f(T_X\mathcal{M}))$. It corresponds to the number of degrees of freedom of the data that are captured by the encoding. For instance, $\text{rank}(J_X^f) = \dim(T_X\mathcal{M})$ indicates that $f$ is sensitive to all local data variations, whereas $\text{rank}(J_X^f) = 0$ means that $f$ is approximately invariant under all local variations and thus approximately constant near $X$.

**Pull-back norm**   To measure the encoding's effectiveness in capturing a data variation, we introduce the average pull-back norm of a vector field. The *pull-back norm* of a tangent vector $V(X)$ at $X$ is defined as[6]

$$\|V(X)\|_f = \|J_X^f(V(X))\|_{\mathcal{N}} = \sqrt{V(X)^T \cdot G_X^f \cdot V(X)}.$$

---

[6]Strictly speaking this is a semi-norm, as it may vanish for non-zero tangent vectors.

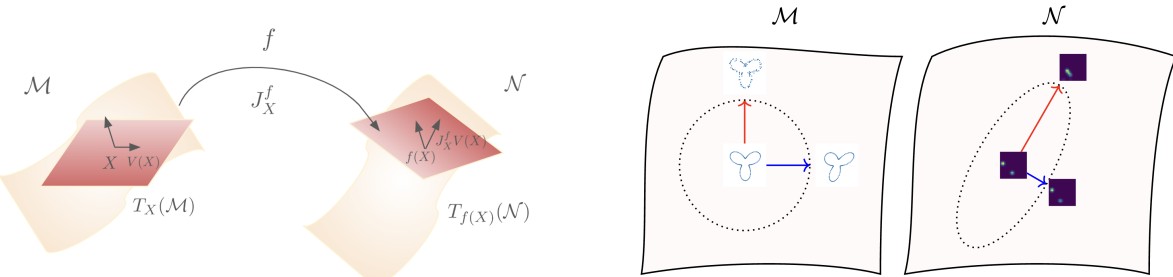

Figure 4: A visualization of the Jacobian map and the pull-back norm. Here $f$ denotes an encoding map from the input space $\mathcal{M}$ to the output space $\mathcal{N}$. Left: the Jacobian of the encoding sends tangent vectors in the tangent space $T_X\mathcal{M}$ of $\mathcal{M}$ to tangent vectors in the tangent space $T_{f(X)}\mathcal{N}$ of $\mathcal{N}$. Right: the pull-back norm of a tangent vector on $\mathcal{M}$ measures by what amount the output of the encoding would change in response to the variation of the input by that tangent vector. In this schematic illustration, the pull-back norm of the red vector ("noising") is larger than the pull-back norm of the blue vector ("shearing").

Here $\|\cdot\|_\mathcal{N}$ denotes the vector norm in output space $\mathcal{N}$ and $G_X^f = (J_X^f)^T J_X^f$ is the Gram matrix of the encoding $f$ at $X$. While in the above definition, we consider the Euclidean metric for the output space of PIs, our approach can be applied for other choices of metric as well. We present the definition of pull-back norm for any differential encoding mapping between Riemannian manifolds in Appendix B.4. We also provide a visualization for the pull-back norm in the right panel of Figure 4.

The pull-back norm of $V(X)$ measures the sensitivity of $f$ to the variation $V(X)$ at $X$. To measure the sensitivity across different inputs, we take the average with respect to a distribution on $\mathcal{M}$. In practice, we use the empirical distribution of a given data set $\mathcal{D} = \{X_i\}_{i=1,\ldots,|\mathcal{D}|}$.

**Definition 3** (Average pull-back norm). The average pull-back norm of a vector field $V$ with respect to an encoding map $f$ and a data set $\mathcal{D} = \{X_i\}_{i=1,\ldots,|\mathcal{D}|}$ of cardinality $|\mathcal{D}|$ is

$$\|V\|_f = \frac{1}{|\mathcal{D}|} \sum_{X \in \mathcal{D}} \|V(X)\|_f.$$

Please note that in Definition 3, $V(X)$ denotes a tangent vector at $X$ in the space of point clouds. Specifically, a vector $V(X) \in T_X\mathcal{M}$ corresponds to a "vector field" on $X$ which assigns a vector to each point $x \in X$ in the point cloud $X$. We say that an encoding can detect a data variation characterized by a vector field $V$ if the encoding is sensitive to $V$, which alludes to the average pull-back norm of $V$.

**Singular value decomposition** To gain a more fine-grained insight into the properties of an encoding, we consider the singular value decomposition (SVD) of the Jacobian matrix,

$$J = \tilde{Q}\Lambda Q^T.$$

Here $\tilde{Q} \in \mathbb{R}^{n \times n}, Q \in \mathbb{R}^{m \times m}$ are orthogonal matrices, and $\Lambda \in \mathbb{R}^{n \times m}$ is a diagonal matrix containing in its diagonal the singular values in decreasing order $\lambda_1 \geq \lambda_2 \geq \cdots \geq \lambda_{\min(m,n)}$. This sequence of ordered singular value is the *spectrum* of the Jacobian $J$. Accordingly, the Gram matrix has eigendecomposition $G = J^T J = Q\Lambda^2 Q^T$. We denote the right singular vectors, i.e., the columns of $Q$, by $q_1,\ldots,q_m$. We will refer to these $q_i$'s as the eigenvectors of the encoding. Any tangent vector $v \in T_X\mathcal{M}$ can be written as $v = \sum \langle v, q_i \rangle q_i$, and its pull-back norm as $\|v\|_f = \sqrt{\sum \lambda_i^2 \langle v, q_i \rangle^2}$. With this, the pull-back norm is decomposed as two parts: the spectrum of Jacobian and the alignment between $v$ and eigenvectors, which is described by the inner product. In particular, the pull-back norm is large if $v$ is aligned with $q_i$'s that have large singular values. This also implies that the eigenvectors with top largest eigenvalues can be regarded as the data variations that the encoding considers most "important". We offer a visualization in Appendix D.

**Comparison between encodings**   Later we will compare different encodings by examining their sensitivity to specific data variations. To place different encodings on the same scaling level, we consider the normalized average pull-back norm $[\sum \|V(X)\|_f / \lambda_1^f]/|\mathcal{D}|$. We provide details about this normalization technique in Appendix F.2. Another way of comparing different encodings is via the Bures-Wasserstein distance (Bhatia et al., 2019) between their Gram matrices. The Bures-Wasserstein distance quantifies the alignment between the eigendecompositions of the Gram matrices. For two positive definite matrices $A$ and $B$, the Bures-Wasserstein distance is computed as:

$$d_{\mathrm{BW}}(A, B) = \left[ \mathrm{Tr} A + \mathrm{Tr} B - 2\mathrm{Tr}(A^{1/2} B A^{1/2})^{1/2} \right]^{1/2}.$$

For matrices that are not strictly positive definite we use the same definition after adding a small multiple of the identity matrix.

## 4   Identifying What Is Recognized

In this section, we seek to identify, for fixed PH encodings, which data variations are recognized and which are ignored. Specifically, we investigate the total amount of data variations that are captured by PH encodings (Section 4.1), and among those captured variations we interpret the most significant ones (Section 4.2). Then, we quantify the "importance" for any data variation (Section 4.3), and measure the dissimilarity of the captured information across different PH encodings (Section 4.4). It is important to notice that, even though our approach eliminates potential inference biases induced by downstream models, our results still significantly depend on the data set under consideration. Therefore, we emphasize that this section is dedicated to investigating "what is recognized" by PH encodings *within specific data sets*.

**Synthetic data**   Throughout this section we consider a synthetic data set of point clouds in $\mathbb{R}^2$ sampled from curves in the Radial Frequency Pattern (RFP) family. A point cloud of this type is shown in Figure 2. The curve $\mathrm{RFP}_{(a,w)}$ is parametrized by $\rho(\theta) = 1 + a\cos(w\theta), \theta \in (0, 2\pi]$. Loosely speaking it represents the shape of a flower with $w$ petals of size characterized by $a$. We take $w$ in $\{3, 4, \ldots, 10\}$ and 10 values of $a$ evenly distributed on the interval $[0.5, 0.9]$. For each curve $\mathrm{RFP}_{(a,w)}$, we evenly sample $N = 150$ points to obtain a point cloud, which is then scaled to the unit square $[0, 1]^2$ (see examples in Appendix F.3). Notably, each curve in the RFP family has the same topology. This allows us to validate the ability of PH to capture information beyond topology. The RFP data set has also been used in studying the importance of specific shape features in shape recognition and object perception (Schmidtmann et al., 2015).

**PH encodings**   We investigate PH encodings constructed on 3 different filtrations: Vietoris-Rips (Rips) filtration, DTM filtration, and Height filtration (with respect to the hyperplane with normal vector $[1,0]^T$). For each filtration we extract the 1-dimensional PDs, and convert them to PIs with the same PI parameters. In the following discussion, we sometimes refer to these encodings by the name of the filtration on which they are constructed, denoting for instance the PH encoding constructed on Rips filtration simply as Rips. For reference we also include a PointNet encoding. PointNet (Qi et al., 2017) is a deep neural network architecture designed for processing point clouds directly. We train the network to predict the number of petals $w$, achieving a test accuracy of 100%. We take the output of the second-to-last layer of the trained PointNet as the output of the PointNet encoding. More details concerning the filtration and PI parameters, and the PointNet encoding are provided in Appendix F.3.

### 4.1   Spectrum of the Jacobian

As explained in Section 3.3, the rank of the Jacobian indicates the maximum amount of information, in a dimension sense, that can be captured by the encoding. The spectrum, i.e., the sequence of ordered singular values $\lambda_1 \geq \lambda_2 \geq \cdots \geq \lambda_m$, provides a fuller picture, indicating to what extent different eigenvectors are highlighted.

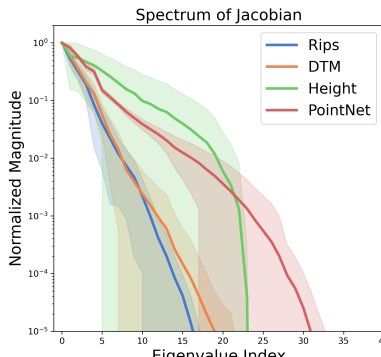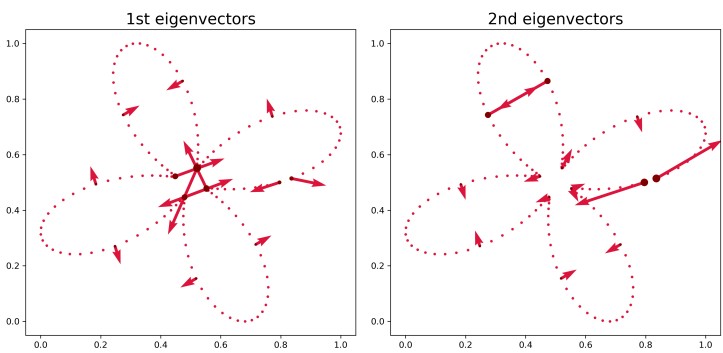

Figure 5: Left: The normalized spectrum of the Jacobian for different encodings. Shown is the mean and standard error of the ordered normalized singular values over different input point clouds. Right: The top two eigenvectors of the Jacobian for the PH encoding constructed on the Rips filtration at a particular input point cloud.

The left plot in Figure 5 reports the normalized spectrum of the Jacobian for different encodings, which is the sequence of ordered normalized singular values

$$1 \geq \frac{\lambda_2}{\lambda_1} \geq \cdots \geq \frac{\lambda_{\mathrm{rank}(J)}}{\lambda_1} > 0.$$

The first observation is that the rank of the Jacobian is much smaller than the dimension of the point cloud space and the PI space. Indeed, note that the dimension of $\mathcal{M}$ is $m = D \times N = 2 \times 150 = 300$, and the dimension of $\mathcal{N}$ is $n = P \times P = 20 \times 20 = 400$. On the other hand, the normalized singular values decay to $10^{-5}$ before index 40. We conclude that the four encodings under consideration capture only a small set of variations in the input data and discard many others. Secondly, while Rips, DTM, and Height all have a similar number of singular values larger than $10^{-5}$, Rips and DTM exhibit a sharper initial decay than Height. This implies that Height has a larger effective rank,[7] while Rips and DTM concentrate their attention more specifically on a few variations. This difference in decay rates may stem from the fact that the size of holes, which is captured by Rips and DTM, is influenced by fewer variations compared to the position of holes, which is information retained by Height.

The middle and right plots in Figure 5, show the top two eigenvectors for Rips at an example point cloud $X$. An eigenvector corresponds to a list of vectors attached to all individual points in the point cloud. These eigenvectors provide insight into the nature of the "important" data variations. More technically, these variations correspond to the most effective way to change the birth/death parameters of certain homology classes.[8] For instance, in the middle plot of Figure 5, the vectors on the petals depict variations that narrow/broaden the petals, which in turn change the death parameter of the corresponding homology classes. The eigenvectors can also be used to obtain point saliency maps, which we discuss in Appendix E. However, we observe that the eigenvectors do not necessarily have an obvious intuitive description. Hence, interpretations are needed to bridge the gap between abstract eigenvectors and human-understandable concepts.

## 4.2 Alignment between eigenvectors and perturbation tangent vectors

To interpret the eigenvectors of the encoding, we consider their alignment with different perturbation vector fields.

**Perturbations on the data manifold** We consider eight types of perturbations applied to the input data, illustrated in Figure 6. The first two, *rotation* and *translation*, are Euclidean motions, i.e., transformations that preserve the Euclidean distances between the points in a point cloud. They are used to test the fact that

---

[7]The effective rank is the number of singular values that have a similar order of magnitude as the top singular value.
[8]This bears some resemblance to adversarial perturbations considered in neural networks.

pointwise distance-based encodings, namely Rips and DTM, should remain invariant under such variations. The *dilation*, *strech_x*, and *shearing* variations serve to test the sensitivity of the encoding to invertible linear transformations of the point clouds. The next two variations are used to test the robustness of PH encoding against noise: the *noising* variation adds coordinate-wise Gaussian noise at each point in the point cloud; the *wiggly* variation adds a sine-type noise at every point in the point cloud along the normal direction. Lastly, the *convex* variation transforms the point cloud towards the boundary of its convex hull through a linear interpolation. We present a visualization of the effects of *shearing* and *convex* on the PH associated with Rips filtration and Height filtration in Appendix F.3.

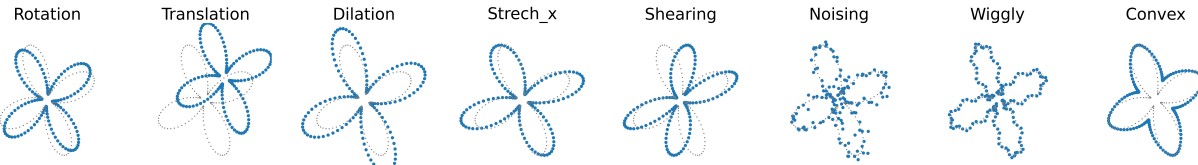

Figure 6: Eight types of perturbations on a RFP pointcloud.

We examine the angles between the top eigenvectors of the different encodings and different perturbation tangent vectors. In Figure 7, we record the average inner product between the perturbation tangent vectors and the top four eigenvectors of each encoding method:

$$\frac{1}{|\mathcal{D}_{\text{RFP}}|} \sum_{X \in \mathcal{D}_{\text{RFP}}} \left| \left\langle \frac{V_\pi(X)}{\|V_\pi(X)\|}, q_i^f \right\rangle \right|, \quad i = 1, 2, 3, 4.$$

The eigenvectors $q_i^f$ depend on $X$. Since each encoding map $f$ induces a different orthonormal basis $\{q_1^f, q_2^f, \ldots, q_m^f\}$ on the tangent space of the data manifold, a fixed tangent vector will have different coordinates with respect to the different encodings. Figure 7 serves as a sort of dictionary, showing how each perturbation (rotation, translation, dilation, etc.) is expressed in the language of each encoding.

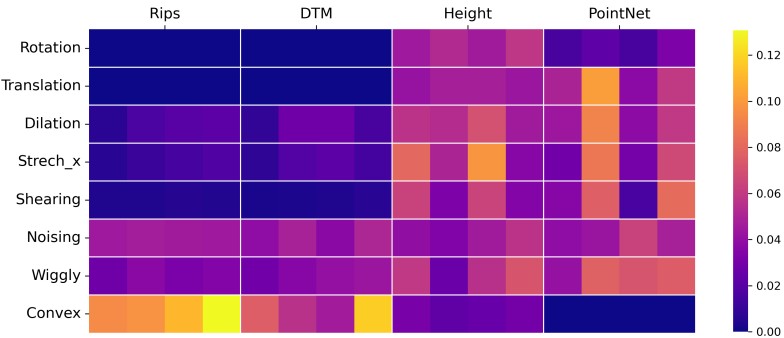

Figure 7: Absolute inner product between perturbation vectors and top four eigenvectors of different encodings. A higher value implies greater alignment, i.e., greater sensitivity to a perturbation.

For the PH encoding constructed on Rips and DTM filtration, we find that the top eigenvectors exhibit a relatively strong alignment with *convex*. The corresponding average inner product is around 0.1. This indicates that the most "important" data variation for Rips and DTM are closely related to convexity. This is consistent with the fact that these encodings capture geometric properties, such as birth values of holes in the filtration (that increase under the *convex* perturbation, see Figure 6). At the same time, the top eigenvectors of Rips and DTM are orthogonal to *rotation* and *translation*. This indicates that Euclidean motion is not as relevant in the Rips and DTM, as is to be expected from the definitions of these encodings.

For the PH encoding constructed on Height filtration, the top eigenvectors have a significant alignment with Euclidean motions and *strech_x*. This makes sense, since Height is designed to collect information on the

position of holes. On the other hand, we do not observe a strong alignment between top eigenvectors of Height and *convex*. This might seem to be in contradiction with Turkeš et al. (2022), who demonstrated that PH on height filtration can be used for detecting convexity. Note, however, that they use 0-dimensional PH on cubical complexes (or analogously, on Rips complexes on geodesic distances), which recognizes concave shapes by their multiple connected components (0-dimensional cycles) for at least some height filtration directions. 0-dimensional PH with respect to such a filtration would see two connected components (two petals) for a while, that would then merge into one at some point. This would happen earlier under *convex* perturbation (i.e., one of the connected components would die sooner), so that the alignment can be expected to be more significant in that case.

For PointNet encoding, the top eigenvector has a relatively strong alignment with *translation*. This is consistent with a previous observation by Turkeš et al. (2022) where the PointNet did not perform well in classification tasks when the test data was corrupted by translations. Note akin to the approach adopted by Turkeš et al. (2022), we do not use data augmentation techniques during the training for PointNet. This might lead to the sensitivity of the trained PointNet to *translation*. On the other side, PointNet is robust under *convex* perturbations. This can be attributed to the nature of RPF data set. Recall the RPF data set comprises point clouds defined by two independent parameters, $a$ and $w$, which characterize the size and the number of the petals, respectively. While the PointNet is trained to identify the number of petals, it can easily learn from the data set to ignore the size of petals. Notice increasing the petal size bears strong resemblance to *convex* perturbation (see examples of RPF point clouds in Appendix F.3). Therefore, one can loosely infer that the RPF data set is "inherently" augmented by *convex* perturbation, and consequently the trained PointNet might learn from the data to ignore *convex* information.

### 4.3 Pull-back norm of perturbation vector fields

In some scenarios, one is interested in the sensitivity of an encoding to certain types of perturbations (Ren et al., 2020). In Figure 8 (left) we evaluate the average pull-back norm of different perturbation vector fields with respect to different encodings. The pull-back norm takes into account not only the alignment with the encoding eigenvectors but also the magnitude of the corresponding singular values. We provide a numerical function in https://github.com/shuangliang15/pullback-geometry-persistent-homology that allows automatic computation of the average pull-back norm with respect to Vietoris-Rips filtration for any given perturbation on a given data set. Note that it is not necessary to have an explicit function description of the perturbation $\pi : \mathcal{M} \to \mathcal{M}$, since we only require the set of perturbed point clouds, i.e., $\pi(X)$ for all $X$ in the data set.

For DTM and Rips, we find that *noising*, *wiggly*, and *convex*, have a significantly larger pull-back norm than the other data variations. This is consistent with the Jacobian spectrum in Figure 5 (which indicates DTM and Rips have faster-decaying spectrum and therefore capture only few data variations), and the alignment information in Figure 7 (which indicates alignment of the top eigenvectors with these particular variations). The PH encoding constructed on Height filtration has a relatively large pull-back norm for many of the considered data variations, including dilation, stretch, and shearing. Also the pull-back norm associated with *convex* perturbations with respect to Height exhibits a moderate average value and a large variance. This implies Height is sensitive to *convex* perturbations in certain point clouds, while being less sensitive in others. This aligns with the alignment information in Figure 7, which indicates the most "important" data variations seen by Height, on average, are not closely related with *convex*. Similar to Height, PointNet also leads to relatively large pull-back norms, but with a different profile and with exception of *convex*, which has a small pull-back norm under PointNet. Rips and DTM have a faster-decaying Jacobian spectrum than Height and PointNet, indicating that they are sensitive to fewer data variations.

### 4.4 Distance between Gram matrices

Along this thread, we can also investigate the relationship between encodings. The right panel of Figure 8 shows the average Bures-Wasserstein distance between the Gram matrices of different encodings.

The average distance matrix shown in the right part of Figure 8 indicates that all encodings are different, whereby some are more similar and some are more dissimilar (see also alignment pattern in Figure 7). Rips

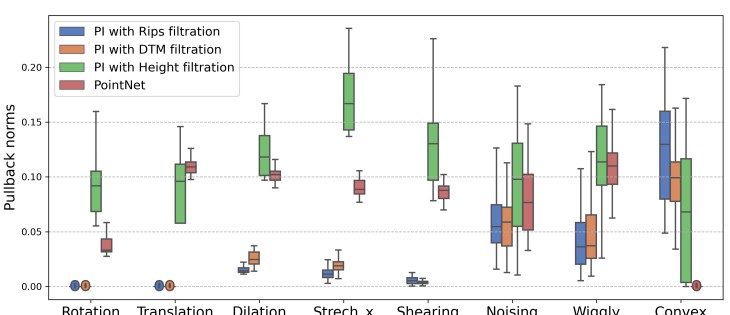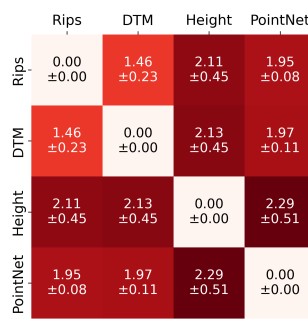

Figure 8: Left: Average pull-back norm of different perturbation vector fields with respect different encodings. Right: Bures-Wasserstein distance between Gram matrices $J^T J$ of different encodings.

and DTM are closest each other, while PointNet and Height both differ significantly from Rips and DTM. This indicates that Rips and DTM capture similar information which is different from the information that is captured by Height and PointNet. This makes sense, since the DTM filtration function is the average distance to neighbors, which approximates the distance function that underlies the Rips filtration; moreover, the data we consider does not contain outliers. We also find that although Height and PointNet give relatively similar pull-back norms for rotation, translation, and dilation, overall these two encodings are very different.

## 5 Selecting Hyperparameters

In this section, we shift our focus to the problem: how do we select the hyperparameters of the encoding in order to detect a data feature of interest. Recall that there are three major hyperparameters to choose when constructig PIs: 1) the resolution $P$, 2) the kernel function $g_{(b,l)}(x, y)$ and its associated parameters and 3) the weighting function $\alpha(b, l)$. We first focus on the first two PI parameters, namely the resolution and the variance for the Gaussian kernels. We examine their impact on the rank and spectrum of Jacobian (Section 5.1.1) and on the pull-back norm of gradient vector fields of data features of interest (Section 5.1.2). We demonstrate there is a strong correlation between the pull-back norms of gradient vector fields and the downstream task performance, where the task objective is to predict that data feature (Section 5.1.3). We then investigate the impact of weighting functions on the pull-back geometry. We introduce the *beta weighting function*, which allows highlighting persistence intervals with different length (persistence time). Then we examine the effects of the mean parameter of beta weighting function on the pull-back geometry (Section 5.2.1). Finally, again we show a significant correlation between the pull-back norm of gradient vector fields and the downstream task performance (Section 5.2.2).

**Real-world data**   In this section we utilize the brain artery tree data (Bendich et al., 2016). This data set comprises 96 artery trees in $\mathbb{R}^3$ (see Figure 9, left). These artery trees are obtained by applying a tube-tracking algorithm to Magnetic Resonance Angiography (MRA) images from 96 human subjects. We randomly subsample three point clouds from the vertices of each artery tree, with each point cloud containing $N = 500$ points. Then we normalize the sampled point clouds to the unit cube $[0, 1]^3$.

**Feature**   Each point cloud is labeled with a binary *sex* feature, based on the corresponding human subjects' medical information.

### 5.1 Resolution and variance of Gaussian kernel

**PH encoding**   We focus on the 1-dimensional PIs on the Vietoris-Rips filtration. We investigate two hyperparameters involved in the construction of PIs: 1) the resolution $P$, and 2) the variance $\gamma^2$ of the Gaussian kernel (see Figure 9, right). We set the baseline PI parameters as $P = 20$, $\gamma^2 = 10^{-4}$, and consider a linear weighting function $\alpha(b, l) = \frac{l}{\max\{l\}}$ (Adams et al., 2017).

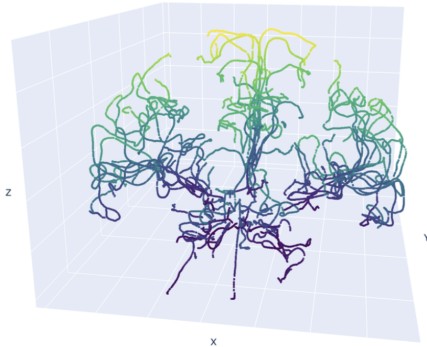
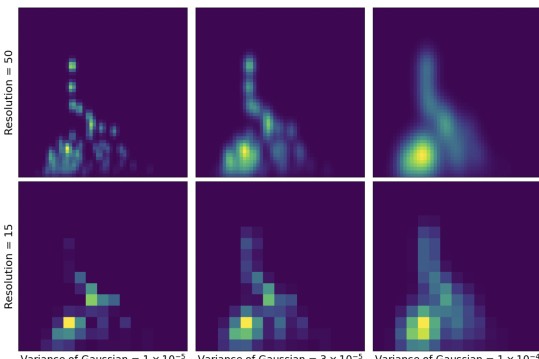

Figure 9: Left: point cloud sampled from a brain artery tree, where the color represents the $z$-coordinate. Right: the corresponding PI representation with different parameter settings.

### 5.1.1   Spectrum of the Jacobian

We begin by analyzing the effects of the resolution $P$ and variance $\gamma^2$ of the Gaussian kernel on the spectrum of the Jacobian. In each plot of Figure 10, we varied one parameter while keeping the other fixed at the baseline setting, and present the normalized spectrum. We again observe a low-rank phenomenon, since the average rank is always below 160, while in the baseline setting the dimension of the point cloud and PI spaces are respectively $m = D \times N = 3 \times 500 = 1500$ and $n = P \times P = 20 \times 20 = 400$.

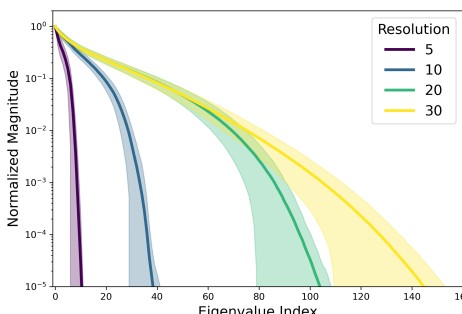
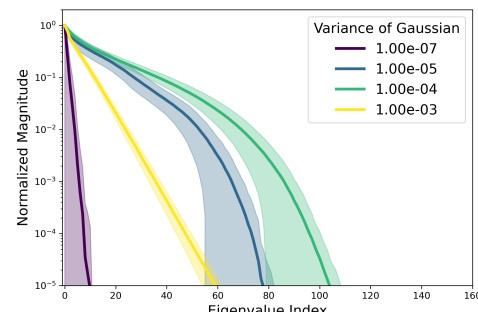

Figure 10: Spectrum of the Jacobian matrix depending on the PI parameters. Left: The effect of changing the resolution $P$ of PI. Right: The effect of changing the variance $\gamma^2$ of Gaussian kernel in PI.

In the left part of Figure 10, we see that the spectrum decays slower as the resolution increases, indicating an increase in rank. This implies that, as one would expect, higher resolution allows the PI to capture more information.

Interestingly, we observe that, as the variance $\gamma^2$ increases, the rank of the Jacobian initially increases and then decreases. For fixed resolution, very small variance results in sparse PIs (see the first column in the right panel of Figure 9), where multiple PD points $(b, l)$ may fall into one pixel and can only highlight that pixel; very large variance leads to blurred PIs (see the third column in the right panel of Figure 9), where it also becomes difficult to distinguish between PD points.

### 5.1.2   Pull-back norm of feature gradient vector fields

We now explore the effects of the resolution and variance on the pull-back norm of gradient vector fields of the following data feature.

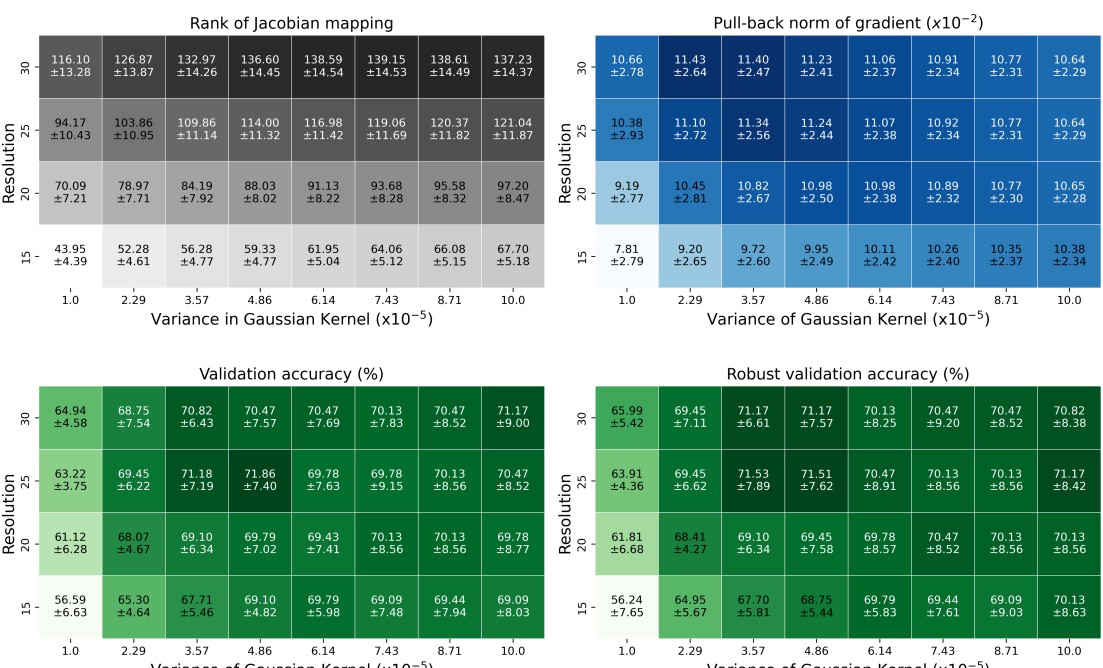

Figure 11: For the brain artery data set, shown is the effect of the resolution (vertical axis) and variance of the Gaussian kernel (horizontal axis) of the PI on the average rank of the Jacobian (upper left), average pull-back norm of the gradient vector field of the *sex* feature (upper right), as well as the test accuracy (lower left) and robust test accuracy (lower right) of the logistic regression model predicting *sex* based on the PI.

Our goal is to locate the optimal PI parameters of the PH encoding to effectively detect the *sex* feature in the brain artery point clouds. We consider the domain $(P, \gamma^2) \in [15, 30] \times [10^{-5}, 10^{-4}]$. Here the ranges for $P$ and $\gamma^2$ are selected based on the values where the spectra in Figure 10 exhibit the slowest decay.

In the upper right plot of Figure 11, we present the average pull-back norm of the gradient field of the *sex* feature under different PI parameter choices. The gradient fields are estimated via numerical methods detailed in Appendix F.4. We observe that the pull-back norm generally increases as the resolution $P$ increases, whereas the pull-back norm is not monotonic with $\gamma^2$. Moreover, the optimal value for $\gamma^2$ varies depending on the choice of resolution. Also, comparison with the upper left part of Figure 11 reveals that the maximum pull-back norm is not necessarily attained for parameters where the rank of the Jacobian is maximal, i.e., when PIs capture the most information about the point cloud.

### 5.1.3 Correlation with downstream task performance

We investigate the hypothesis that a high pull-back norm correlates with the performance of a predictor trained on the encoding. To this end we feed PIs generated with different choices of the parameters into logistic regression models and train these to predict the *sex* feature. Here we use logistic regression as the downstream model because of its simplicity, with the intention to minimize the impact of model complexity and training techniques on the task performance. We provide results for convolutional neural networks (CNN) in Appendix F.4.

We evaluate the performance in terms of validation accuracy and robust validation accuracy[9] using cross-validation, which are presented in the lower left and lower right plots in Figure 11. The validation accuracy and robust validation accuracy exhibit a similar pattern to the pull-back norm. Notably, all three quantities reach their maximum at around $P = 30$ and $\gamma^2 = 3.57 \times 10^{-5}$, and their minimum at the lower-left corner.

---

[9]Robust validation accuracy evaluates the accuracy on a test data set subject to additive Gaussian noise on the inputs.

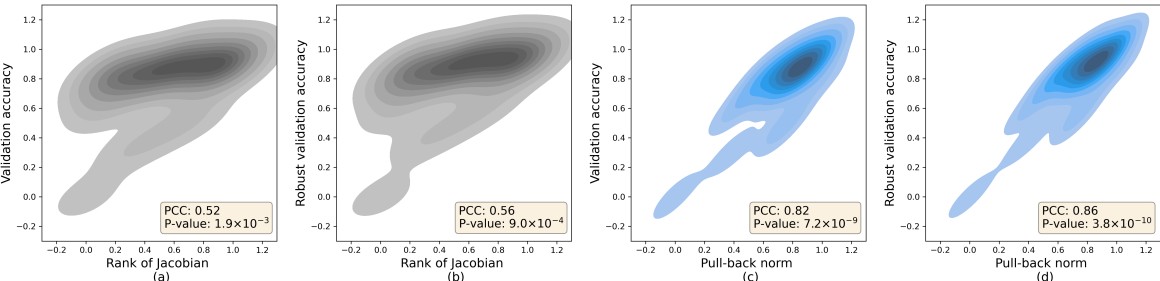

Figure 12: Gaussian kernel density estimation of the joint distribution of four pairs of variables: (a) Jacobian rank vs. validation accuracy; (b) Jacobian rank vs. robust validation accuracy; (c) pull-back norms vs. validation accuracy; and (d) pull-back norms vs. robust validation accuracy, where the downstream models are chosen as logistic regression models.

For a more quantitative comparison, Figure 12 shows a kernel density estimate of the joint distribution of four pairs of variables: Jacobian rank vs. validation accuracy, Jacobian rank vs. robust validation accuracy, pull-back norm vs. validation accuracy, and pull-back norm vs. robust validation accuracy. The plots clearly indicate a strong correlation between the pull-back norm and the performance on the downstream task. The Pearson's correlation coefficient (PCC) between the four pairs of variables, along with the p-value for a two-sided test, are presented in the lower right corner of each plot in Figure 12.

We conclude that for the considered task, there is a significant correlation between the pull-back norm and the downstream task performance. It is also interesting to interpret these results together with the rank of the Jacobian (upper left in Figure 11). The results demonstrate that the improvement in downstream performance is only somewhat correlated with including more information, but it is strongly correlated with including the most relevant information, which is precisely quantified by the pull-back norm. Therefore, we suggest that the proposed framework can be used to select appropriate PH encodings in practice. Note that the procedure is independent of the downstream model architectures and training techniques.

## 5.2 Weighting function

**PH encoding** We maintain our focus on the 1-dimensional PIs with respect to the Vietoris-Rips filtration. We set the baseline PI parameters as $P = 20$ and $\gamma^2 = 3 \times 10^{-5}$. For the weighting function, we consider the *beta weighting function* induced by the probability density function of a beta distribution:

$$\alpha(b, l) = \frac{\Gamma(\alpha + \beta)}{\Gamma(\alpha)\Gamma(\beta)}(\kappa l)^{\alpha-1}(1 - \kappa l)^{\beta-1}$$

where $\Gamma(\cdot)$ is the Gamma function and $\kappa$ is a scaling factor. We consider the mean-variance parameterization for the beta weighting function: $\alpha = k\left(\frac{k(1-k)}{s^2} - 1\right)$ and $\beta = (1 - k)\left(\frac{k(1-k)}{s^2} - 1\right)$. Here $k$ is the *mean parameter*, which controls the concentration of the weighting function, and $s^2$ is the variance parameter, which controls the "degree" of concentration. We set $s^2$ as 0.065 and $\kappa$ as 1.

We consider beta weighting function for several reasons: 1) beta weighting function is compactly supported, which is more suitable for PI; 2) it assigns zero weight to the horizontal axis, which aligns with the stability criteria proposed by Adams et al. (2017); 3) by tuning the mean parameter, one can highlight persistence intervals with different length (see Figure 13 for an illustration). This allows to investigate questions such as "are short persistence intervals more crucial to this application than long persistence intervals?"

### 5.2.1 Pull-back norm of feature gradient vector fields

We investigate the effects of the mean parameter $k$ on the rank of Jacobian and pull-back norm of the gradient vector field of the *sex* feature (see the left and middle panel in Figure 14). In the left panel of Figure 14, we observe that as the weighting function assigns more importance to longer persistence intervals, the rank

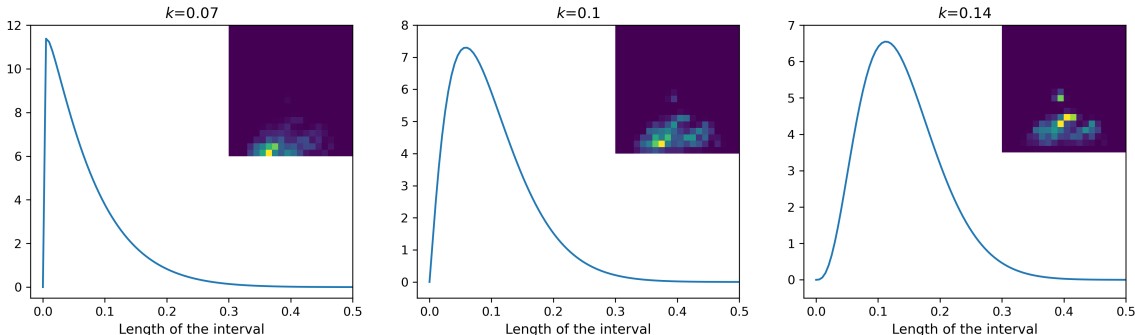

Figure 13: As a weighting function for PIs for the brain artery data, we use the beta weighting function with different values of mean parameter $k$. Larger $k$ assigns more importance to longer persistence intervals. The top right corner of each panel shows the 1-dimensional PI derived from the Rips filtration on one point cloud, illustrating the impact of the weighting function depicted in the main plot.

of the Jacobian monotonically decreases. This aligns with the fact that the number of longer persistence intervals is generally smaller than the number of shorter persistence intervals. However, the pull-back norm peaks when $k$ is set to 0.1 and then decreases as $k$ increases further.

### 5.2.2 Correlation with downstream task performance

We again examine the correlation between the pull-back norm and the performance of the logistic regression models trained on PIs. We present the validation accuracy in the right panel of Figure 14. Notably, we observe that the validation accuracy demonstrates a similar pattern to the pull-back norm. Quantitatively, the Pearson's correlation coefficient (PCC) between pull-back norm and validation accuracy is 0.839, with a two-sided test p-value of 0.009. In contrast, the PCC between rank and validation accuracy is 0.338, with a p-value as 0.413, which indicates once again that including more information in the data representation does not necessarily improve the downstream performance. These findings reinforce our conclusion that the pull-back norm is highly predictive for the downstream task performance in this task.

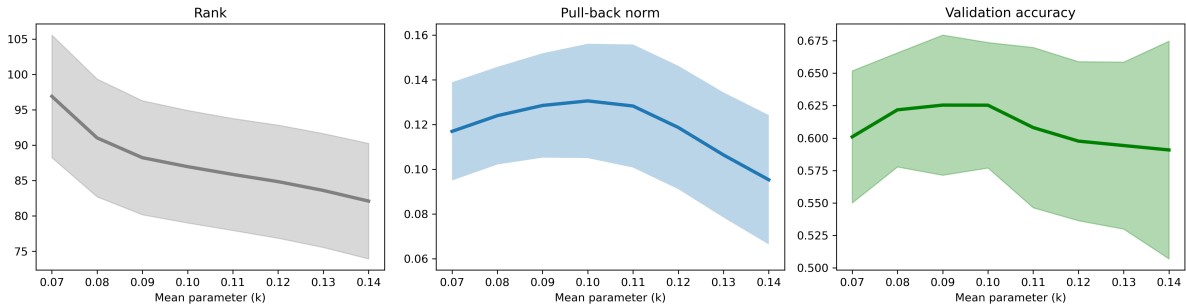

Figure 14: The impact of the mean parameter for the beta weighting function on the rank of Jacobian (left), pull-back norm of gradient field of feature *sex* (middle), and the 7-folded validation classification accuracy (right). The pull-back norms and validation accuracy are strongly correlated, and they both indicate persistence intervals with medium length are vital to classify the *sex* feature.

Interestingly, we observe that both the pull-back norm and the validation accuracy reach their maximal at an intermediate value for the mean parameter. This implies that persistence intervals of medium length are most crucial for classifying the *sex* feature, which is consistent with the observation in the original paper (Bendich et al., 2016). We note that this is an example of application for which medium length intervals in the barcode contain the most information for the problem at hand. In Appendix G, we complement this discussion by considering a real-world data set of point clouds sampled from human body meshes, on which

we compare long and short persistence intervals from a different perspective: which *part* of the point clouds is the focus of long intervals and which is the focus of short ones.

## 6  Conclusions

The methods and observations presented in this work contribute to addressing some of the main bottlenecks in the practical application of PH, namely how to identify which data variations are captured by PH encodings, how to quantify the effectiveness of these encodings in detecting particular data features, and how to select the parameters of PH encodings in order to obtain data representations that are suitable for solving a particular task.

We presented ways to analyze the most relevant features on the data manifold that are captured by persistence images with different choices of the filtration and compared the results with neural-network-based encodings. For example, in the RFP data set we found that while a pretrained PointNet had a relatively high alignment with translation and dilation, the 1-dimensional persistence image encoding with Height filtration had a high alignment with stretch, and the 1-dimensional persistence image encoding with Rips filtration had a high alignment with a data variation that makes the point clouds more convex. At the same time we observe that the response of the encodings to these perturbations is less than 10% as strong as for other more abstract data variations captured by the singular vectors of the Jacobian; for instance, the maximal value taken by inner products between unit tangent vectors representing perturbations and unit singular vectors is less than 0.1.

We demonstrated on the real-world brain artery tree data set that feature alignment as measured by the Jacobian permits PH parameter tuning without the need to train a classifier on top of the data representation in order to select the parameters based on the test accuracy. Rather, one can select the parameters based on the pull-back norm of the features of interest, and perform training using the data representation with the highest pull-back norm. Meanwhile, we found that the persistence intervals of medium length are crucial for classifying the *sex* feature on this data set. This goes against the popular belief that long intervals are the most important, at the same time confirming the findings from the original paper that employs PH on this data set.

**Limitations and future work**   Our analysis is based on the structure of the Jacobian of the data encoding, which by nature focuses only on local variations of the input data. In future it will be interesting to further advance these methods in regard to non-local data variations, where synthetic notions of derivatives such as our empirical evaluation of the vector fields, and ideas such as the application of the iterated closest point method could serve as a point of departure. Further, the analysis of non-linear transformations via Gram matrices has seen a number of recent advances in the context of artificial neural networks, and it will be interesting to explore possible synergies between those investigations and PH data encodings.

A limitation of the proposed methodology is the assumption about the differentiability of the encoding, and the need for a Riemannian manifold structure for the representation space. For this reason, our methodology cannot be directly applied to analyze certain common PH representations, such as persistence diagrams, since these cannot be endowed with a smooth structure (Leygonie et al., 2022). We note, however, that a Riemannian framework for *approximated* PDs has been introduced by Anirudh et al. (2016).

The pull-back geometry approach also faces some computational challenges, since calculating the average pull-back norm for either perturbation vector fields or gradient vector fields requires the computation of the Jacobian matrix $J_X$, which is of size $(P \times P, D \times N)$, for each data point $X$, where $P$, $D$, $N$ are respectively the resolution for PI, the dimension of points in the point cloud and the number of points in the point cloud. On the positive side, this enables insights that are more intrinsic to the problem rather than being dependent on the choice of a classifier. Performance-based methods can also involve computational challenges, due to the need to choose the downstream models and tune their hyperparameters. We regard the proposed methods not as a substitute but as complementary to performance-based methods.

**Acknowledgements**

We thank anonymous TMLR referees for helpful feedback. This project has been supported in part by NSF CAREER 2145630, NSF 2212520, DFG SPP 2298 grant 464109215, ERC Starting Grant 757983, and BMBF in DAAD project 57616814.

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

## Appendix

The appendix is organized into the following sections.

- Appendix A: Notation

- Appendix B: Riemannian manifold structure of the space of point clouds

- Appendix C: Differentiability of the mapping from point clouds to PIs

- Appendix D: Visualizing the Jacobian of the encoding over the data manifold

- Appendix E: Point saliency maps for PH encodings

- Appendix F: Details on the experiments

- Appendix G: Investigating which part of the data is highlighted by PH encodings

## A   Notation

Table 1 provides a summary of the notation.

## B   Riemannian manifold structure of the space of point clouds

Let $\mathcal{M}$ denote the collection of all point clouds in $\mathbb{R}^D$ that contain exactly $N$ points,

$$\mathcal{M} = \{X \subset \mathbb{R}^D : |X| = N\}.$$

Recall that the 2-Wasserstein distance $d_W$ between two point clouds of the same size in $\mathbb{R}^D$ is defined as

$$d_W(X,Y) = \min_{\omega \in \Omega(X,Y)} \left( \sum_{x \in X} d_E^2(x, \omega(x)) \right)^{\frac{1}{2}},$$

where $\omega$ is an bijection between $X$ and $Y$, $\Omega(X,Y)$ contains all bijections, and $d_E$ denotes the Euclidean metric on $\mathbb{R}^D$. The 2-Wasserstein distance defined above induces a metric topology on $\mathcal{M}$.

Compared to other distances in the space of point clouds, for instance the Gromov-Hausdorff distance which is commonly used in the study of stability theory of persistent homology (see, e.g., Blumberg & Lesnick, 2022), the 2-Wasserstein distance endows the space of point clouds with a favorable manifold structure. This manifold structure ensures that every small neighborhood is isometric to an Euclidean open set.

We discuss the topological manifold structure (Appendix B.1) and Riemannian manifold structure (Appendix B.2) on the space of point clouds. Then we introduce the Riemannian style definition for perturbation vector fields and gradient vector fields (Appendix B.3).

### B.1   Manifold structure

We proceed to establish a manifold structure on $\mathcal{M}$.

**Proposition 4.** *Let $\mathcal{M}$ be the set containing all point clouds in $\mathbb{R}^D$ with $N$ distinct points, and $d_W$ be the 2-Wasserstein distance on $\mathcal{M}$. For any point cloud $X \in \mathcal{M}$, there exists a Wasserstein ball $B_W(X, \varepsilon_X)$ and an injective mapping $\xi_X : B_W(X, \varepsilon_X) \to \mathbb{R}^{D \times N}$ such that*

$$d_W(Y, Z) = d_E(\xi_X(Y), \xi_X(Z)), \quad \forall Y, Z \in B_W(X, \varepsilon_X).$$

Table 1: Notations and definitions

| Notation | Definition |
|---|---|
| $\mathcal{M}$ | The manifold of point clouds |
| $m$ | Dimension of manifold $\mathcal{M}$ |
| $D$ | Dimension of the space where point clouds are located |
| $N$ | Number of points in point clouds |
| $X$ | Point cloud |
| $x$ | Point in a point cloud |
| $d_W$ | Wasserstein distance between point clouds |
| $[N]$ | The set $\{1, 2, \ldots, N\}$ |
| $\Omega(X, Y)$ | The set of bijections between point clouds $X$ and $Y$ |
| $\omega : X \to Y$ | Bijection between point clouds $X$ and $Y$ |
| $d_E$ | Euclidean distance |
| $K, K_r$ | Simplicial complex |
| $\sigma$ | Simplex |
| $\phi : K \to \mathbb{R}$ | Filtration function |
| $b$ | Birth parameter of homology classes |
| $d$ | Death parameter of homology classes |
| $l$ | Lifespan parameter of homology classes |
| $T_X \mathcal{M}$ | Tangent space at $X$ on $\mathcal{M}$ |
| $v$ | Tangent vector |
| $V : \mathcal{M} \to \sqcup_X T_X \mathcal{M}$ | Vector field |
| $V(X)$ | Tangent vector at $X$ assigned by $V$ |
| $\pi : \mathcal{M} \to \mathcal{M}$ | Perturbation mapping on the space of point clouds |
| $V_\pi$ | Vector field induced by perturbation $\pi$ |
| $\rho : \mathcal{M} \to \mathbb{R}$ | One-dimensional feature function on the space of point clouds |
| $\nabla \rho$ | Gradient vector field of function $\rho$ |
| $\mathcal{N}$ | The manifold of persistence images |
| $n$ | Dimension of manifold $\mathcal{N}$ |
| $\eta$ | Transformation on PD points from birth-death to birth-lifespan coordinate |
| $g_{b,l}$ | Gaussian kernel located at $(b, l)$ |
| $P$ | Resolution of persistence images |
| $\gamma^2$ | Variance of Gaussian kernel in persistence images |
| $\alpha(b, l)$ | Weighting function |
| $\psi$ | Persistence surface |
| $k$ | Mean parameter for the beta weighting function |
| $f : \mathcal{M} \to \mathcal{N}$ | Encoding map from $\mathcal{M}$ to $\mathcal{N}$ |
| $J_X^f, J^f, J_X, J$ | Jacobian mapping (Jacobian matrix) of map $f$ at $X$ |
| $G_X^f, G^f, G_X, G$ | Gram matrix of map $f$ at $X$ |
| $\lambda_i^f, \lambda_i$ | The $i$-th largest singular value of the Jacobian mapping |
| $q_i^f, q_i$ | The $i$-th eigenvector of the PH encoding mapping |
| $\|\cdot\|_f$ | Pull-back norm induced by mapping $f$ |
| $\mathcal{D}$ | Finite data set of point clouds |
| $d_{BW}$ | Bures-Wasserstein distance between positive-definite matrices |

*Proof.* Consider a point cloud $X = \{x_i\}_{i=1}^N$ in $\mathcal{M}$. For arbitrary $\varepsilon > 0$, we construct an injective mapping $\xi_X$ from $B_W(X, \varepsilon)$ to $\mathbb{R}^{D \times N}$, then choose a radius $\xi_X$ such that the above equation holds. Denote $[N] = \{1, 2, \ldots, N\}$. The map $\xi_X(X)$ can be characterized by a total order in $X$, $\tau : [N] \to X$, where

$$\xi_X(X) = [\tau(1), \tau(2), \ldots, \tau(N)] \in \mathbb{R}^{D \times N}.$$

$\tau$ reorders $X$ by assigning $[N] = \{1, 2, \ldots, N\}$ to $\{x_i\}_{i=1}^N$. For any other point cloud $Y \in B_W(X, \varepsilon)$, there exists an optimal transport plan between $X$ and $Y$, denoted by $\omega_{XY} : X \to Y$, satisfying

$$\omega_{XY} = \underset{\omega \in \Omega(X,Y)}{\arg\min} \left( \sum_{x \in X} d_E^2(x, \omega(x)) \right)^{\frac{1}{2}}.$$

This maps assigns each element in $X$ to a distinct element in $Y$. We define an embedding $\xi_X$ from $Y$ to $\mathbb{R}^{D \times N}$ as follows:

$$\xi_X(Y) = [\omega_{XY} \circ \tau(1), \omega_{XY} \circ \tau(2), \ldots, \omega_{XY} \circ \tau(N)] \in \mathbb{R}^{D \times N}.$$

We proceed to show that $\xi_X$ is an injective embedding. For any $Y, Z \in B_W(X, \varepsilon)$ with $\xi_X(Y) = \xi_X(Z)$, consider optimal transport plans $\omega_{XY}$ between $X$ and $Y$, and $\omega_{XZ}$ between $X$ and $Z$. Since $\xi_X(Y) = \xi_X(Z)$, we have $\omega_{XY}(x) = \omega_{XZ}(x), \forall x \in X$. Hence, for any $y \in Y$,

$$y = \omega_{XY} \circ (\omega_{XY})^{-1}(y) = \omega_{XZ} \circ (\omega_{XY})^{-1}(y) \in Z.$$

Notice $\omega_{XZ} \circ (\omega_{XY})^{-1}$ is a bijection between $Y$ and $Z$. Therefore, $Y = Z$ and $\xi_X$ is injective.

Next we calculate the radius $\xi_X$ that preserves the distance between any two point clouds. The goal is to find a radius $\varepsilon$ such that for any $Y, Z \in B_W(X, \varepsilon)$, the Wasserstein distance between the point clouds $Y$ and $Z$,

$$d_W(Y, Z) = \left( \sum_{y \in Y} d_E^2(y, \omega_{YZ}(y)) \right)^{\frac{1}{2}},$$

is equal to the Euclidean distance between the embedding $\xi_X(Y)$ and $\xi_X(Z)$,

$$d_E(\xi_X(Y), \xi_X(Z)) = \left( \sum_{i=1}^N d_E^2(\omega_{XY} \circ \tau(i), \omega_{XZ} \circ \tau(i)) \right)^{\frac{1}{2}}$$

$$= \left( \sum_{x \in X} d_E^2(\omega_{XY}(x), \omega_{XZ}(x)) \right)^{\frac{1}{2}}$$

$$= \left( \sum_{y \in Y} d_E^2(y, \omega_{XZ} \circ (\omega_{XY})^{-1}(y)) \right)^{\frac{1}{2}}.$$

Notice it suffices to find a radius $\varepsilon$ such that for any $Y, Z \in B_W(X, \varepsilon)$, $\omega_{YZ} = \omega_{XZ} \circ (\omega_{XY})^{-1}$. Equivalently, the optimal bijection between $Y$ and $Z$ is given by the composition $\omega_{XZ} \circ (\omega_{XY})^{-1} : Y \to X \to Z$. The key idea is that if $Y$ and $Z$ are both sufficiently close to $X$ in the sense of the Wasserstein distance, then the distance between $y \in Y$ and $(\omega_{XY})^{-1}(y)$ and the distance between $z \in Z$ and $(\omega_{XZ})^{-1}(z)$ will be small. Hence, each point $y \in Y$ will be close to $\omega_{XZ} \circ (\omega_{XY})^{-1}(y)$ and thus we will have $\omega_{YZ} = \omega_{XZ} \circ (\omega_{XY})^{-1}$. The situation is illustrated in Figure 15.

Denote $\omega_{XZ} \circ (\omega_{XY})^{-1}$ as $\widetilde{\omega_{YZ}}$. For any point cloud $Y \in B_W(X, \varepsilon)$, we have

$$d_W(X, Y) = \left( \sum_{x \in X} d_E^2(x, \omega_{XY}(x)) \right)^{\frac{1}{2}} < \varepsilon.$$

Hence,

$$\max_{x \in X} d_E(x, \omega_{XY}(x)) < \varepsilon.$$

This suggests for any two point clouds $Y, Z \in B_W(X, \varepsilon)$,

$$\max_{x \in X} d_E(\omega_{XY}(x), \omega_{XZ}(x)) < 2\varepsilon.$$

Equivalently,

$$\max_{y \in Y} d_E(y, \widetilde{\omega_{YZ}}(y)) < 2\varepsilon.$$

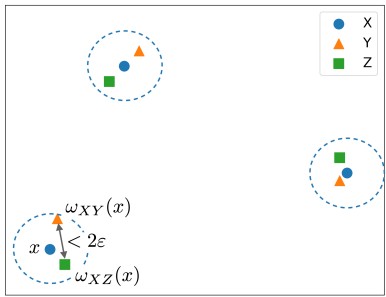 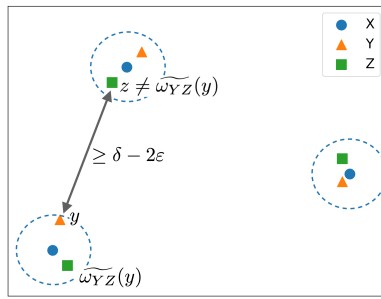

Figure 15: Shown are point clouds $Y$ (orange triangles) and $Z$ (green squares) in a Wasserstein neighborhood of point cloud $X$ (blue circles). Left: both $\omega_{XY}(x) \in Y$ and $\omega_{XZ}(x) \in Z$ are close to $x \in X$, and hence they are close to each other. Right: the distance between $y$ and $z \neq \widetilde{\omega_{YZ}}(y) \in Z$ is lower bounded.

Let $\delta$ denote the minimal pairwise distance of points in $X$:

$$\delta = \min_{x_1, x_2 \in X} d_E(x_1, x_2).$$

Note $\delta$ is strictly greater than zero since points in $X$ are mutually different. For a fixed point $y \in Y$, any point $z$ in $Z$ other than $\widetilde{\omega_{YZ}}(y)$ has a lower-bounded distance from $y$:

$$\min_{z \in Z \smallsetminus \{\widetilde{\omega_{YZ}}(y)\}} d_E(y, z) \geq \delta - 2\varepsilon, \quad \forall y \in Y.$$

Now consider $\varepsilon_X = \frac{\delta}{8}$. We have

$$d_E(y, \widetilde{\omega_{YZ}}(y)) < 2\varepsilon_X = \frac{1}{4}\delta < \frac{3}{4}\delta = \delta - 2\varepsilon_X \leq \min_{z \in Z \smallsetminus \{\widetilde{\omega_{YZ}}(y)\}} d_E(y, z), \quad \forall y \in Y.$$

Equivalently,

$$\widetilde{\omega_{YZ}}(y) = \arg\min_{z \in Z} d_E(y, z), \quad \forall y \in Y.$$

This means $\omega_{YZ} = \widetilde{\omega_{YZ}}$, which completes the proof. $\qquad\square$

**Corollary 5.** *Let $\mathcal{M}$ be the set containing all point clouds in $\mathbb{R}^D$ with $N$ distinct points. $\mathcal{M}$, together with the metric topology induced by $2$-Wasserstein distance, forms a manifold of dimension $D \times N$.*

*Proof.* By Proposition 4, for every $X \in \mathcal{M}$ one can find a neighborhood $B_W(X, \varepsilon_X)$ and an injective mapping $\xi_X : B_W(X, \varepsilon_X) \to \mathbb{R}^{D \times N}$ satisfying

$$d_W(Y, Z) = d_E(\xi_X(Y), \xi_X(Z)), \quad \forall Y, Z \in B_W(X, \varepsilon_X).$$

Notice $\xi_X$ is a bijective isometry between $(B_W(X, \varepsilon_X), d_W)$ and $(\xi_X(B_W(X, \varepsilon_X)), d_E)$. Hence, $\xi_X$ is open and continuous. Consequently, $\xi_X$ is a homeomorphism, and $\{\xi_X : B_W(X, \varepsilon_X) \to \mathbb{R}^{D \times N}\}_{X \in \mathcal{M}}$, serving as an atlas, endows $\mathcal{M}$ with the manifold structure. $\qquad\square$

## B.2 Riemannian metric structure

Next we introduce the Riemannian metric structure for the manifold of point clouds, and show the distance induced by the Riemannian metric coincides with the Wasserstein distance. To this end, consider an alternative definition for the space of point clouds:

$$\mathcal{M}' = \{X : [N] \to \mathbb{R}^D \mid X \text{ is injective}\}/\sim_{S_N}.$$

The equivalence relation is defined by:

$$X_1 \sim_{S_N} X_2 \quad \Leftrightarrow \quad \exists \nu \in S_N : X_1 \circ \nu = X_2 \quad \Leftrightarrow \quad \mathrm{Im}(X_1) = \mathrm{Im}(X_2).$$

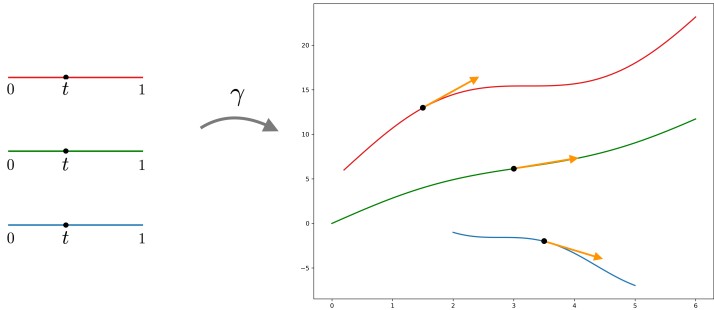

Figure 16: Shown is a curve in the point cloud space $[\gamma] \in \mathcal{C}$, which maps the set $\{1, 2, 3\} \times [0, 1]$ (left) to the plane $\mathbb{R}^2$ (right). In the right panel, the collection of black dots represent the image of $\gamma$ at a specific time $t \in [0, 1]$, which forms a point cloud in $\mathbb{R}^2$; the collection of orange arrows represents the velocity tangent vector at $\gamma_t$ along the curve $\gamma$.

Here $S_N$ denotes the $N$-symmetric group and $\mathrm{Im}(X)$ denotes the image of mapping $X$. Two mappings are deemed equivalent when their images are identical. Note the image of each mapping $X : [N] \to \mathbb{R}^D$ is a point cloud in $\mathbb{R}^D$ as we defined earlier, i.e. $\mathrm{Im}(X) \in \mathcal{M}$. In fact, the mapping $\mathcal{M}' \to \mathcal{M}; [X] \mapsto \mathrm{Im}(X)$ gives the identification between the original and new definitions for the space of point clouds. For simplicity, we use one representative $X$ to denote the equivalence class $[X]$ and also refer to $X$'s as point clouds.

This definition allows defining smooth curves in the point clouds space. Specifically, a smooth curves in $\mathcal{M}'$ is an element of the following set:

$$\mathcal{C} = \{\gamma : [N] \times I \to \mathbb{R}^D \mid \gamma(\cdot, t) \text{ is injective}, \; \forall t \in I; \gamma(k, \cdot) \in C^\infty(I), \forall k \in [N]\}/ \sim_{S_N} .$$

Here $I$ denote the closed unit interval $[0, 1]$ and the equivalence relation is defined by

$$\gamma_1 \sim_{S_N} \gamma_2 \quad \Leftrightarrow \quad \exists \nu \in S_N : \gamma_1(\nu(\cdot), \cdot)) = \gamma_2(\cdot, \cdot)) \quad \Leftrightarrow \quad \mathrm{Im}(\gamma_1(\cdot, t)) = \mathrm{Im}(\gamma_2(\cdot, t)), \forall t.$$

A curve in point cloud space $[\gamma] \in \mathcal{C}$ is essentially a collection of $N$ curves in $\mathbb{R}^D$ such that at each time $t$ the $N$ points on those curves form a point cloud (see Figure 16). For simplicity we use $\gamma$ to denote the equivalence class $[\gamma]$.

The *tangent space* at point cloud $X$ is defined as

$$T_X \mathcal{M}' = \{V \mid_{\mathrm{Im}(X)} : \mathrm{Im}(X) \to \mathbb{R}^D \mid V \text{ a vector field on } \mathbb{R}^D\}.$$

A *Riemannian metric* is a manifold structure that smoothly assigns a positive-definite inner product $g_X(\cdot, \cdot)$ on the tangent space $T_X \mathcal{M}$ at each $X \in \mathcal{M}$. We introduce a Riemannian metric for $\mathcal{M}$ as follows: $T_X \mathcal{M}'$:

$$g_X(V, W) = \sum_{x \in \mathrm{Im}(X)} \langle V(x), W(x) \rangle.$$

Endowed with this Riemannian metric, $\mathcal{M}'$ is a Riemannian manifold, denoted by $(\mathcal{M}', g)$. We now discuss the distance induced by this Riemannian metric. For a curve in point cloud space $\gamma \in \mathcal{C}$, the *velocity tangent vector* at $\gamma_t$ along curve $\gamma$ is defined as (see an example in the right panel in Figure 16):

$$\dot{\gamma}_t : \mathrm{Im}(\gamma_t) \to \mathbb{R}^D; \quad x \to \partial_t \gamma(k_x, t).$$

Here $k_x \in [N]$ is the preimage of point $x$ under $\gamma_t$, i.e., $\gamma_t(k_x) = x$. The *length of curve* is given by the length functional $L$:

$$L : \mathcal{C} \to \mathbb{R}; \quad \gamma \mapsto \int_I \sqrt{g_{\gamma_t}(\dot{\gamma}_t, \dot{\gamma}_t)} dt.$$

The distance induced by the Riemannian metric is the minimal length of curves between two points on the manifold.

**Definition 6** (Riemannian distance)**.** Let $(\mathcal{M}, g)$ be a Riemannian manifold. The Riemannian distance between two points $X, Y \in \mathcal{M}$ is defined as

$$d_g(X, Y) = \inf\{L(\gamma) : \gamma \text{ a smooth curve in } \mathcal{M} \text{ connecting } X \text{ and } Y\}.$$

We point out the Riemannian distance coincides with the Wasserstein distance. To see this, we need the following result by Do Carmo & Flaherty Francis (Lemma 2.3, Chap. 9, 1992).

**Lemma 7** (Do Carmo & Flaherty Francis, 1992)**.** Let $X_0, X_1$ be two points in Riemannian manifold $\mathcal{M}$, and $\gamma$ a curve joining $X_0$ to $X_1$. Then $\gamma$ minimizes the length functional if and only if $\gamma$ minimizes the energy functional defined as follows:

$$E : \mathcal{C} \to \mathbb{R}; \quad \gamma \mapsto \int_I g_{\gamma_t}(\dot{\gamma}_t, \dot{\gamma}_t) dt.$$

Moreover, when $\gamma$ is the minimizer, $L(\gamma) = \sqrt{E(\gamma)}$.

Now we proceed to introduce the main statement of this subsection.

**Proposition 8.** *Let $(\mathcal{M}', g)$ be the Riemannian manifold of point clouds. Then the Riemannian distance is equivalent to Wasserstein distance, $d_g = d_W$.*

*Proof.* Consider two point clouds $X_0, X_1 \in \mathcal{M}'$. Let $L$ be the length functional, and $E$ be the energy functional of all curves connecting $X_0$ and $X_1$. For any curve $\gamma$ joining $X_0$ to $X_1$, we have

$$
\begin{aligned}
E(\gamma) &= \int_I g_{\gamma_t}(\dot{\gamma}_t, \dot{\gamma}_t)\, dt \\
&= \int_I \sum_{x \in \mathrm{Im}(\gamma_t)} \langle \dot{\gamma}_t(x), \dot{\gamma}_t(x) \rangle dt \\
&= \int_I \sum_{x \in \mathrm{Im}(\gamma_t)} \|\partial_t \gamma(k_x, t)\|^2 dt \\
&= \int_I \sum_{k=1}^N \|\partial_t \gamma(k, t)\|^2 dt \\
&= \sum_{k=1}^N \int_I \|\partial_t \gamma(k, t)\|^2 dt \\
&= \sum_{k=1}^N E[\gamma(k, \cdot)].
\end{aligned}
$$

The above equations along with Lemma 7 indicate the minimizer of the length functional $L$ coincides with the minimizer of $\sum_{k=1}^N E(\gamma(k, \cdot))$. Recall for each $k$, $\gamma(k, \cdot)$ is a curve in $\mathbb{R}^D$ joining $X_0(k)$ to $X_1(k)$. In Euclidean spaces, it is known that straight lines minimize the length functional, which implies, by Lemma 7, they also minimize $E(\gamma(k, \cdot))$. Specifically for fixed end points $X_0(k)$ and $X_1(k)$, $E(\gamma(k, \cdot))$ has minimal value $d_E^2(X_0(k), X_1(k))$. Therefore, finding the minimizer of $\sum_{k=1}^N E(\gamma(k, \cdot))$ is equivalent to finding the bijection between $\mathrm{Im}(X_0)$ and $\mathrm{Im}(X_1)$ that produces the minimal value of the sum of squared distances between points paired by the bijection. This exactly coincides with the optimal transport problem. In conclusion, we have

$$
\begin{aligned}
d_g(X_0, X_1) &= \min_{\gamma} L(\gamma) \\
&= \min_{\gamma} \sqrt{E(\gamma)} \\
&= \min \left( \sum_{k=1}^N E[\gamma(k, \cdot)] \right)^{\frac{1}{2}} \\
&= \min_{\omega \in \Omega(\mathrm{Im}(X_0), \mathrm{Im}(X_1))} \left( \sum_{x \in \mathrm{Im}(X_0)} d^2(x, \omega(x)) \right)^{\frac{1}{2}} \\
&= d_W(\mathrm{Im}(X_0), \mathrm{Im}(X_1)).
\end{aligned}
$$

This is what was claimed. $\qquad\square$

### B.3 Vector fields

In this section, we provide definitions for perturbation vector fields and gradient vector fields for the manifold of point clouds.

Assume $\pi : \mathcal{M} \to \mathcal{M}$ is a perturbation mapping. As introduced in Section 3.2, the perturbation vector fields $V_\pi$ in the Euclidean space is defined as:

$$V_\pi(X) = \pi(X) - X, \quad \forall X \in \mathcal{M}.$$

The main idea behind this definition is that each tangent vector $V_\pi(X)$ specifies the direction of the straight line connecting $X$ and $\pi(X)$. For general Riemannian manifolds, the notion of straight lines is generalized by minimizing geodesics. Formally, the curve between two points on manifold that minimizes the length functional is a *minimizing geodesic*. Then the question arises whether there exists a minimizing geodesic between any two points on the manifold. To address this, we introduce the concepts of geodesic completeness and the Hopf-Rinow theorem.

As introduced in Appendix B.2, the Riemannian metric induces a distance $d_g$ on the manifold $\mathcal{M}$. A sequence $\{X_i\}_{i \in \mathbb{Z}^+}$ of points on $(\mathcal{M}, d_g)$ is a $d_g$-*Cauchy sequence* if for any positive number $\varepsilon$ there exists a positive integer $N$ such that $d_g(X_i, X_j) < \varepsilon, \forall i, j > N$.

**Definition 9** (Geodesically complete manifold)**.** The Riemannian manifold $(\mathcal{M}, g)$ is geodesically complete if any $d_g$-Cauchy sequence $\{X_i\}_{i \in \mathbb{Z}^+}$ converges in $\mathcal{M}$: $\exists Y \in \mathcal{M}$ such that $\lim_{i \to \infty} d_g(X_i, Y) = 0$.

Since the Euclidean space $\mathbb{R}^D$ is a complete metric space, automatically the Riemannian manifold of point clouds is geodesically complete. Also notice this manifold is connected, since there exists a path connecting any two point clouds. The Hopf-Rinow theorem ensures the existence of minimizing geodesics between any two point clouds.

**Theorem 10** (Hopf-Rinow theorem)**.** *Let $(\mathcal{M}, g)$ be a connected Riemannian manifold. If $(\mathcal{M}, d_g)$ is geodesically complete, there exists a minimizing geodesic between any two points on $\mathcal{M}$.*

Now we formally define the perturbation vector fields.

**Definition 11** (Perturbation vector field)**.** Let $(\mathcal{M}, g)$ be a geodesically complete manifold, and $\pi : \mathcal{M} \to \mathcal{M}$ be a perturbation mapping. The perturbation vector field $V_\pi$ is defined as

$$V_\pi : \mathcal{M} \to \sqcup_X T_X \mathcal{M}; \quad X \mapsto V_\pi(X) = \dot{\gamma}_{X,\pi(X)}(0),$$

where $\gamma_{X,\pi(X)}$ is a minimizing geodesic $\gamma_{X,\pi(X)} : I \to \mathcal{M}$ with $\gamma_{X,\pi(X)}(0) = X$ and $\gamma_{X,\pi(X)}(1) = \pi(X)$.

We proceed to define the Riemannian gradient vector field. Assume $\rho$ is a real-valued smooth function on $\mathcal{M}$. In the cases of Euclidean spaces, the gradient vector can be characterized by the following property:

$$\langle \nabla \rho(X), v \rangle = \frac{\partial}{\partial v} \rho, \quad \forall X \in \mathbb{R}^m, v \in T_X \mathbb{R}^m = \mathbb{R}^m.$$

For general Riemannian manifolds, the notion of directional derivatives is generalized by derivations.

**Definition 12** (Derivative)**.** Let $\mathcal{M}$ be a manifold, and $C^\infty(\mathcal{M})$ be the space of smooth functions on $\mathcal{M}$. A derivative at $X \in \mathcal{M}$ is a linear map $\partial : C^\infty(\mathcal{M}) \to \mathbb{R}$ satisfying the Leibniz identity:

$$\partial(fg) = \partial(f) \cdot g(X) + \partial(g) \cdot f(X).$$

For a fixed point $X \in \mathcal{M}$, it turns out that each tangent vector $v \in T_X \mathcal{M}$ can be uniquely associated with a derivative, denoted by $\partial_v$, in the sense that $\partial_v(\rho)$ measures the rate of change of the function value $\rho(X)$, moving through $X$ with the velocity specified by $v$. Detailed discussion regarding the equivalence between tangent vectors and derivations can be found in the work of Tu (Chapter 8, 2011). Now we provide the definition of the Riemannian gradient.

**Definition 13** (Gradient vector field). Let $(\mathcal{M}, g)$ be a Riemannian manifold, and $\rho : \mathcal{M} \to \mathbb{R}$ a smooth function. The gradient vector field of $\rho$, denoted by $\nabla \rho$, is defined as the vector field

$$\nabla \rho : \mathcal{M} \to \sqcup_X T_X \mathcal{M}; \quad X \mapsto \nabla \rho(X)$$

satisfying the property:

$$g_X(\nabla \rho(X), v) = \partial_v(\rho), \quad \forall X \in \mathcal{M}, v \in T_X \mathcal{M}.$$

Note that Definition 11 and Definition 13 are applicable to other types of data, provided that the data space can be equipped with a geodesically complete Riemannian manifold structure.

### B.4 Pull-back metric

In this section, we provide definitions for pull-back metric for general encoding mappings between Riemannian manifolds.

Let $(\mathcal{M}, g_{\mathcal{M}})$ be the Riemannian manifold for input data, $(\mathcal{N}, g_{\mathcal{N}})$ be the Riemannian manifold for output data, and $f : \mathcal{M} \to \mathcal{N}$ be a differential encoding mapping between the input space and output space. Recall that for any $X$ in $\mathcal{M}$ the Jacobian of $f$ at $X$, denoted by $J_X^f$, is a linear mapping between $T_X \mathcal{M}$ and $T_{f(X)} \mathcal{N}$. The *pull-back metric* induced by $f$, denoted by $g^f$, is the structure that assigns the following inner-product on the tangent space $T_X \mathcal{M}$ for each $X \in \mathcal{M}$:

$$g^f(V, W) = g_{\mathcal{N}}(J_X^f(V), J_X^f(W)).$$

Then the *pull-back norm* for any tangent vector $V$ at $X$ is defined as

$$\|V\|_f = \sqrt{g^f(V, V)}.$$

Please note that when one considers $\mathcal{N}$ as a vector space and $g_{\mathcal{N}}$ as the Euclidean metric, the above definition for pull-back metric reduces to the one that we introduced in Section 3.3. Meanwhile, we point out that, while the Riemannian metric on $\mathcal{M}$ does not affect the pull-back metric, it is still necessary in our approach since it's essential in defining the perturbation vector field and gradient vector field (see Definition 11 and Definition 13).

## C Differentiability of the mapping from point clouds to PIs

We provide details for the differentiability of the mapping from point cloud data to PIs and the computation of the Jacobian. The encoding mapping from the point clouds to PIs can be viewed as the composition of the following two maps:

$$\begin{aligned} f : \quad & \mathcal{M} \to \mathrm{Bar} \to \mathcal{N} \\ & X \mapsto \mathrm{PD}_k(\mathrm{Filt}(X)) \mapsto \mathrm{PI}(\mathrm{PD}_k(\mathrm{Filt}(X))). \end{aligned}$$

Here Bar denotes the set of PDs, $\mathrm{Filt}(X)$ denotes a filtration on a point cloud $X$, and $\mathrm{PD}_k(X)$ denotes the $k$-dimensional PD on filtration $\mathrm{Filt}(X)$.

Recall that by definition, a persistence diagram is a multiset consisting of bounded and infinite persistence intervals, with its elements in $\mathbb{R} \times \bar{\mathbb{R}}$ where $\bar{\mathbb{R}} \triangleq \mathbb{R} \cup \{+\infty\}$. As a space of multisets, Bar does not naturally come equipped with a differential structure. However, one can study the differentiability for maps from or to Bar by equipping Bar with a *diffeology* structure (Leygonie et al., 2022, Section 3.5). Specifically, for any positive integers $i$ and $j$, Bar is covered by the map $\mathbb{R}^{2i} \times \mathbb{R}^j \xrightarrow{S_{i,j}} \mathrm{Bar}$, where $\mathbb{R}^{2i} \times \mathbb{R}^j$ can be thought of as the space of *ordered* PDs consisting of a fixed number $i$ (resp. $j$) of bounded (resp. infinite) persistence intervals and $S_{i,j}$ is the quotient map modulo the order. Note the action of $S_{i,j}$ would turn vectors into multisets. Then a map $A : \mathcal{M} \to \mathrm{Bar}$ is said to be differentiable at $X \in \mathcal{M}$ if there exists an open neighborhood $U$ of $X$,

positive integers $i, j$ and a differential map $\tilde{A} : U \to \mathbb{R}^{2i} \times \mathbb{R}^j$ such that $A = S_{i,j} \circ \tilde{A}$:

$$
\begin{array}{ccc}
 & & \mathbb{R}^{2i} \times \mathbb{R}^j \\
 & \overset{\tilde{A}}{\nearrow} & \big\downarrow S_{i,j} \\
\mathcal{M} & \overset{A}{\longrightarrow} & \text{Bar}
\end{array}
$$

Similarly, a map $B : \text{Bar} \to \mathcal{N}$ is said to be differentiable at $Z \in \text{Bar}$ if for all positive integers $i, j$ and all vectors $\tilde{Z} \in \mathbb{R}^{2i} \times \mathbb{R}^j$ such that $S_{i,j}(\tilde{Z}) = Z$, the map $B \circ S_{i,j} : \mathbb{R}^{2i} \times \mathbb{R}^j \to \mathbb{N}$ is differential on an open neighborhood of $\tilde{Z}$:

$$
\begin{array}{c}
\mathbb{R}^{2i} \times \mathbb{R}^j \\
\big\downarrow S_{i,j} \\
\text{Bar} \overset{B}{\longrightarrow} \mathcal{N}
\end{array}
$$

We refer reader to Leygonie et al. (2022) for a detailed introduction to the analysis above.

**Differentiability for the map from point clouds to PDs** Let $X \in \mathcal{M}$ be a point cloud, $U$ be an open neighborhood of $X$, and $\text{Cl}(X, R)$ be the clique complex on the $R$-neighborhood graph of $X$. A filtration Filt would induce a total order on the simplices in $\text{CL}(X, R)$ in the sense that $\sigma \preceq \tau$ if and only if $\phi(\sigma) \leq \phi(\tau)$ where $\phi$ is the filtration function determining the filtration. First let us assume that 1) Filt induces a strict total order at point cloud $X$, i.e., simplices in $\text{Cl}(X, R)$ have distinct filtration values, and 2) the distance between any two points in $X$ does not equal to $R$. The filtration Filt then defines a map sending a point cloud to a vector of filtration values as follows:

$$
\tilde{A}_1 : \quad X \mapsto \big(\phi(\sigma_1), \phi(\sigma_2), \ldots, \phi(\sigma_{|\text{Cl}(X,R)|}\big) \in \mathbb{R}^{|\text{Cl}(X,R)|},
$$

where $\phi(\sigma_1) < \phi(\sigma_2) < \cdots < \phi(\sigma_{|\text{Cl}(X,R)|})$. Note that, according to our second assumption, i.e., pairwise distances of points in $X$ do not equal to $R$, one can assume that all point clouds in $U$ would have the same number of simplices in the corresponding clique complexes. Specifically, consider any point cloud $Y \in U$ and the optimal transport plan $\omega_{XY}$ from $X$ to $Y$ (see Appendix B.1). There exists a bijection between $\in \text{Cl}(X, R)$ and $\text{Cl}(Y, R)$:

$$
\Delta_{XY} : \sigma^X \mapsto \{\omega_{XY}(x) : x \in \sigma^X\} \in \text{Cl}(Y, R).
$$

Hence, the map $\tilde{A}_1$ is well-defined on $U$. Note also that $\tilde{A}_1$ is differentiable at $X$ if the filtration value $\phi(\sigma)$ is differentiable at $X$ for any simplex $\sigma \in \text{Cl}(X, R)$. Then given a homology degree $k$, the filtration would define a *barcode template*, which is a multiset $P_k(X)$ of pairs of simplices in $\text{Cl}(X, R)$ and a multiset $U_k(X)$ of simplices in $\text{Cl}(X, R)$, such that (Leygonie et al., 2022):

$$
\text{PD}_k(\text{Filt}(X)) = \{(\phi(\sigma), \phi(\tau))\}_{(\sigma,\tau) \in P_k(X)} \cup \{(\phi(\sigma'), +\infty)\}_{\sigma' \in U_k(X)}.
$$

This defines a map $\tilde{A}_2$ sending the vector of ordered filtration values to a vector of ordered PD:

$$
\tilde{A}_2 : \quad \tilde{A}_1(X) \mapsto \tilde{Z} = \Big(\big((\phi(\sigma_1), \phi(\tau_1)), \ldots, (\phi(\sigma_i), \phi(\tau_i))\big), \quad \big(\phi(\sigma'_1), \ldots, \phi(\sigma'_j)\big)\Big) \in \mathbb{R}^{2i} \times \mathbb{R}^j.
$$

In the case where the order induced by the filtration is a strict total order, the barcode template is uniquely defined, and hence the map $\tilde{A}_2$ is well defined (see Brüel-Gabrielsson et al. 2019 for a detailed discussion). Meanwhile, one can assume that the barcode template would be stable on $U$ in the sense that if $(\sigma, \tau) \in P_k(X)$ (resp. $\sigma' \in U_k(X)$), then $(\Delta_{XY}(\sigma), \Delta_{XY}(\tau)) \in P_k(Y)$ (resp. $\Delta_{XY}(\sigma') \in U_k(Y)$). Hence, the map $\tilde{A}_2$ can be viewed as a fixed permutation on $\tilde{A}_1(U)$ and is hence differentiable. In practice, the *barcode template* can be obtained via the matrix reduction algorithm; we refer readers to Leygonie et al. (2022) for more details regarding the existence and construction of the barcode template. Please note that the map $\tilde{A} = \tilde{A}_2 \circ \tilde{A}_1$ would be the desired differentiable map such that $\text{PD}_k(\text{Filt}(X)) = S_{i,j} \circ \tilde{A}(X)$ and with this map we have shown the map $\text{PD}_k(\text{Filt}(\cdot))$ is differentiable at $X$ if the filtration defines a strict total order on $\text{Cl}(X, R)$ and the pairwise distances of points in $X$ do not equal to $R$.

At certain point clouds, the filtration could induce a non-strict total order since multiple simplices can have the same filtration value. In such cases, the map $\tilde{A}_2$ may no longer be well-defined since the barcode template is not unique. Consequently, the derivative of $\tilde{A}_2$ should be considered as a subderivative (Brüel-Gabrielsson et al., 2019) and the map $\text{PD}_k(\text{Filt}(\cdot))$ is not differentiable at such point clouds. In practice, this can be resolved by refining the total order into a strict order deterministically or randomly (see, e.g., Skraba et al., 2017), which corresponds to selecting an element in the subderivative. On the other side, at a point cloud $X$ in which there are two points with a distance between them equal to $R$, a small perturbation on these critical points can result in removing simplices in $\text{Cl}(X, R)$. Hence, the the map $\tilde{A}_1$ may not be continuous at $X$ even if one considers its codomain as a space of sequences, and the map $\text{PD}_k(\text{Filt}(\cdot))$ would be non-differentiable at such point cloud. However, please note that the set of point clouds with such property has a dense and open complement in $\mathcal{M}$.

**Differentiability for the map from PDs to PIs** The map from PDs to PIs is guaranteed to be differentiable everywhere in Bar (Leygonie et al., 2022, Proposition 7.3). We note that here one needs to consider the subset of Bar consisting of only bounded intervals. In practice, for computations, one sets the death value of infinite intervals to the maximum filtration value $R$.

**Generic differentiability for PH encoding** As shown above, the PH encoding is differentiable at point clouds satisfying 1) the filtration defines a strict total order and 2) the pairwise distances of points in the point cloud do not equal to $R$. We now point out that for Vietoris-Rips filtration and Height filtration, the PH encoding is generically differentiable, i.e., the set where PH encoding is differentiable is dense in $\mathcal{M}$. As we indicated above, the set of point clouds that satisfies the second requirement is open and dense in $\mathcal{M}$. Hence, it is suffices to show the set of point clouds where the filtration defines a strict total order is also dense and open, as the intersection of open, dense sets would still be dense.

For Height filtration, by Leygonie et al. (Proposition 5.2, 2022) we claim that the filtration defines a strict total order for point clouds outside the following set

$$\tilde{\mathcal{M}} = \{X \in \mathcal{M} : \exists \sigma_1, \sigma_2 \in \text{Cl}_0(X, R), \phi(\sigma_1) = \phi(\sigma_2)\},$$

where $\text{Cl}_0(X, R)$ denotes the set of all vertices in $\text{Cl}(X, R)$. Hence, the PH encoding would be generically differentiable since the complement of $\tilde{\mathcal{M}}$ is dense in $\mathcal{M}$. We point out that Leygonie et al. (2022) consider a slightly different setting where the input space $\mathcal{M}$ is considered as a parameter space for the map $\phi_0 \triangleq \phi \mid_{\text{Cl}_0(X,R)}$. For example, consider $\mathcal{M} = \mathbb{S}^{D-1}$ and for each $\theta \in \mathcal{M}$ define $\phi_0(v; \theta) = \langle v, \theta \rangle, \forall v \in \text{Cl}_0(X, R)$. However, one can easily adapt our setting to be consistent with the their setting as follows. Assume the height filtration function $\phi_0$ is determined by the vector $w$, i.e., $\phi_0(v) = \langle v, w \rangle$. Consider any point cloud $X \in \mathcal{M}$ and a neighborhood $U$ of $X$. The map $\phi_0$ can be viewed as parametrized by $U$ in the sense that for any $Y \in U$, define $\phi_0(v; Y) = \langle \omega_{XY}(v), w \rangle, \forall v \in \text{Cl}_0(X, R)$, where $\omega_{XY}$ is the optimal transport plan from $X$ to $Y$ (see Appendix B). Intuitively, as moving a vertex $v \in \text{Cl}_0(X, R)$ would result in a change in $\phi_0(v)$, the coordinates of all the vertices in $\text{Cl}_0(X, R)$ can be viewed as a parametrization for the map $\phi_0$.

For Vietoris-Rips filtration, we will need the concept of *general position* for point clouds. Specifically, a point cloud $X = \{x_1, \ldots, x_N\}$ is said to be in *general position* if $\forall \{i, j\} \neq \{k, l\}$, where $i, j, k, l \in \{1, \ldots, n\}, \|x_i - x_j\|_2 \neq \|x_k - x_l\|_2$. Following Leygonie et al. (Corollary 5.9, 2022), we claim that the Rips filtration would define a strict order for point clouds in general position. As the subspace of such point clouds is dense in $\mathcal{M}$, the PH encoding is generically differentiable. We point out that Leygonie et al. (Corollary 5.9, 2022) consider the domain of the barcode valued map as the vector space $\mathbb{R}^{D \times N}$. However, their results can be extended to the above claim by noticing that the manifold of point clouds in our setting is locally homeomorphic to $\mathbb{R}^{D \times N}$ (see Appendix B.1).

**Computation for the partial derivatives** We now derive the partial derivatives for the PH encoding $f : \mathcal{M} \to \mathcal{N} \subset \mathbb{R}^{P \times P}$. First, assume $f$ is differentiable at $X \in \mathcal{M}$. As discussed in Appendix B.1, there exists a neighborhood $U$ of $X$ and a coordinate map $\xi : U \to \mathbb{R}^{D \times N}$. Specifically, assume $\tau : [N] \to X$ is the total order that characterizes $\xi(X)$, i.e., $\xi(X) = [\tau(1), \tau(2), \ldots, \tau(N)]$ (see Appendix B). Recall that when constructing PIs from transformed PDs, one needs to choose a smooth kernel $g_{(b,l)}$, e.g., Gaussian kernel, and a smooth

weighting function $\alpha(b, l)$, e.g., linear function of $l$. The persistence surface $\psi$ would then be defined as:

$$\psi(x, y) = \sum_{(b,l) \in \eta(\text{PD})} \alpha(b, l) g_{(b,l)}(x, y).$$

Then, one can obtain a persistence image in the form of a $P \times P$ matrix whose $(i, j)$-th entry is:

$$\text{PI}_{ij} = \int_{\text{pixel }_{ij}} \psi(x, y).$$

Here $\cup_{ij}$ pixel $_{ij}$ forms a grid subdivision of a subdomain of $\psi$. In our experiments, we consider a rectangle subdomain $[x_{\min}, x_{\max}] \times [y_{\min}, y_{\max}]$ and evenly-spaced rectangle pixels. Specifically, consider grid points $x_k = x_{\min} + k\Delta x$ for $k = 0, 1, \ldots, P$, where $\Delta x = \frac{x_{\max} - x_{\min}}{P}$, and $y_l = y_{\min} + l\Delta y$ for $l = 0, 1, \ldots, P$, where $\Delta y = \frac{y_{\max} - y_{\min}}{P}$. Set $\text{pixel}_{ij} = [x_i, x_{i+1}] \times [y_j, y_{j+1}]$ for $i = 0, 1, \ldots, P - 1$ and $j = 0, 1, \ldots, P - 1$. We then estimate the integral value by

$$\text{PI}_{ij} = s \cdot \psi(x_i, y_i),$$

where $s$ is the area of $\text{pixel}_{ij}$, i.e., $s = \Delta x \Delta y$. For any $(i, j) \in [P] \times [P]$, $k \in [N]$, the partial derivative $\frac{\partial \text{PI}_{ij}}{\partial \tau(k)}$ can be formulated as

$$\frac{\partial \text{PI}_{ij}}{\partial \tau(k)} = \sum_{\phi(\sigma) \in \text{PD}_k(\text{Filt}(X))} \frac{\partial \text{PI}_{ij}}{\partial \phi(\sigma)} \frac{\partial \phi(\sigma)}{\partial \tau(k)}.$$

We point out that $\text{PD}_k(\text{Filt}(X))$ is a multiset consisting of bounded and infinite persistence intervals and the notation $\phi(\sigma) \in \text{PD}_k(\text{Filt}(X))$ is used to denote the summation is taken over all the filtration values that appear in these intervals.

For point clouds where the filtration only defines a non strict total order, $\frac{\partial \phi(\sigma)}{\partial \tau(k)}$ should be viewed as a subderivative as we discussed above and one needs to select an element in the subderivative. For example, consider a point cloud $X_0$ consisting of points evenly-spaced on $\mathbb{S}^1$ (see Figure 17) and the PH encoding that outputs the 1-dimensional PI on the Rips filtration. Notice the distance between any two adjacent points takes the same value. Therefore, the *barcode template* is not unique. In the first row of Figure 17, we show the top eigenvector and the Jacobian matrix at $X_0$ and other two point clouds near $X_0$. As shown in the figure, both eigenvectors and Jacobian matrices are indeed unstable. We point out, however, if we perturb $X_0$ even slightly, then we can expect the information contained in the Jacobian to be reliable, because the set of points clouds for which the encoding is differentiable is dense in $\mathcal{M}$. As shown in the second row of Figure 17, the eigenvector and the Jacobian matrix become stable near point cloud $X_1$, which is obtained by adding a slight noise on $X_0$. A similar result can be observed for a point cloud whose points are uniformly sampled from $\mathbb{S}^1$ (see the third row of Figure 17). We also note that in practice, due to numerical approximation errors, it might still be the case that, e.g., even for points in generic position, one might have that two edges in the Vietoris-Rips complex might have the same filtration value. This is an issue that needs to be considered in applications, and that could inform specific choices in computations.

Meanwhile, we point out that while the information of Jacobian on specific point clouds can be unstable, the computational results in this work are still reliable since these quantities, e.g., average inner-product between perturbation vector fields and the top eigenvectors (Section 4.2) and average pull-back norm (Section 4.3), are computed over diverse point clouds within a dataset and those undesirable point clouds are very sparse in the input space as we have previously shown. To demonstrate this clearer, we experimentally investigate the relation between average pull-back norm and the number of point clouds considered in the computation (see Figure 18). We obeserve that, on both RPF and brain artery tree data set, the pull-back norm quickly converges to a stable value once a certain amount of point clouds are included in the computation. This demonstrates the stability and reliability of the insights obtained via our methodology.

To compute the Jacobian in our experiments, we use the Gudhi library (The GUDHI Project, 2020) version 3.8.0 and Tensorflow version 2.12.0. Specifically, we use Gudhi library to collect PDs, especially `Gudhi.tensorflow.RipsLayer` class to collect PDs with respect to the Rips filtration. We then manually compute PIs as described above using Tensorflow. Finally, we collect the Jacobian for the whole pipeline with `tensorflow.GradientTape.jacobian` function.

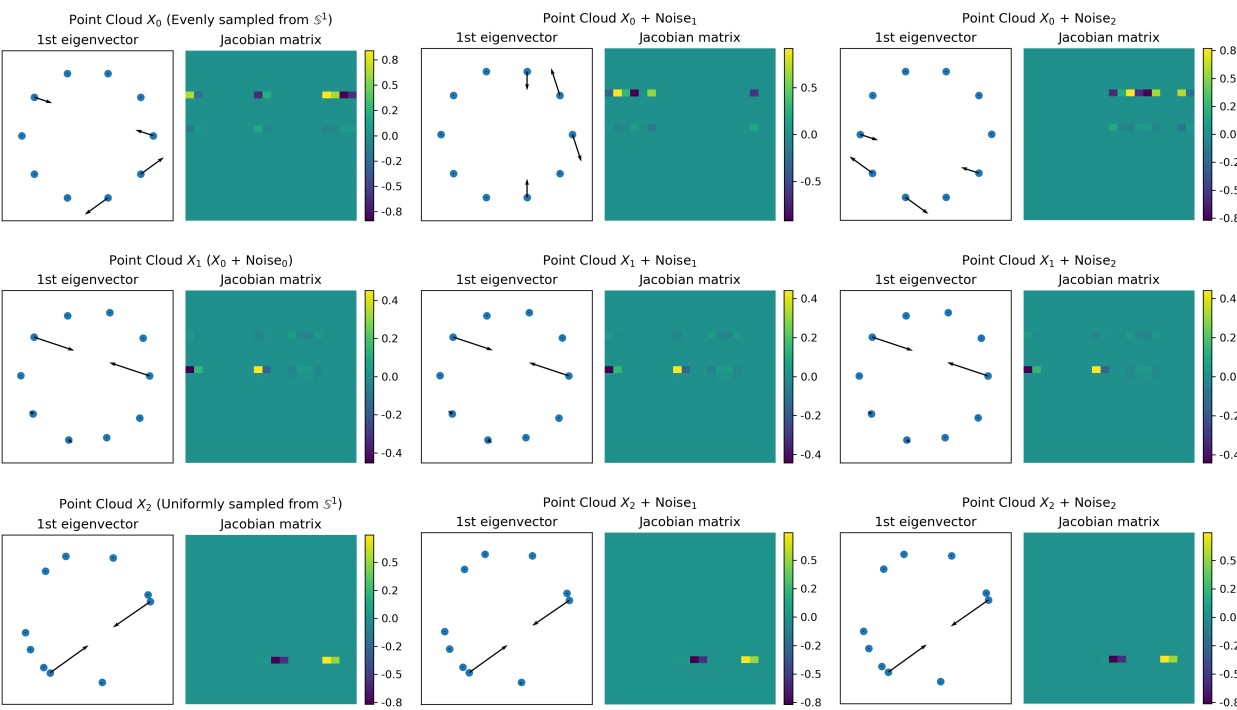

Figure 17: Stability of the Jacobian at three point clouds: $X_0$, comprising points evenly spaced on $\mathbb{S}^1$ (first row); $X_1$, obtained by adding slight noise to $X_0$ (second row); and $X_2$, consisting of points uniformly sampled from $\mathbb{S}^1$ (third row). We consider the PH encoding that outputs the 1-dimensional PI on the Vietoris-Rips filtration. In the first two columns in each row, we show the top eigenvector of the Jacobian and the Jacobian matrix for the original point cloud. In the next two sets of columns we show the same information for the point cloud obtained by adding $\text{Noise}_1$ resp. $\text{Noise}_2$ to the original point cloud. $\text{Noise}_1$ and $\text{Noise}_2$ are consistent across rows. Note that $X_0$ is not sampled randomly from the circle, but in a deterministic manner which ensures the points are equidistant, so that the pair of simplices that yields the birth and death of the loop is not unique. This causes the non-differentiability of the encoding map and the instability of the Jacobian information. However, the Jacobian is stable at $X_1$ and $X_2$, which illustrates that the Jacobian is generically stable in the input space.

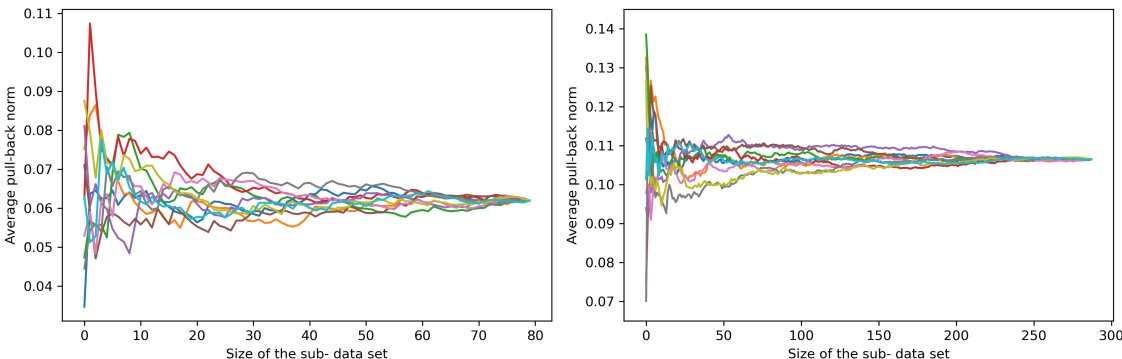

Figure 18: Evaluation of the average pull-back norm of the noising perturbation vector field with respect to the 1-dimensional PH encoding on Rips filtration over subsets of the RPF data set (left) and the average pull-back norm of the gradient vector field of the *sex* feature with respect to the same PH encoding over subsets of the brain artery tree data set (right). In both cases, we begin from one random point cloud and iteratively include one more random point cloud until the entire dataset is exhausted. We record the average pull-back norm as the size of the sub- data set increases. This process is repeated 10 times. We observe that as more point clouds are included, the average pull-back norm converges to a stable value.

## D  Visualizing the Jacobian of the encoding over the data manifold

In this section we visualize the singular value decomposition (SVD) of Jacobian on a toy data set, aiming at providing further intuition for the eigenvectors of Jacobian mappings. We consider point clouds that are uniformly sampled from axis-aligned ellipses of width $w$ and height $h$. These are illustrated in the left panel of Figure 19. To visualize the Jacobian at an input point cloud $X$, we plot the pull-back unit ball around $X$ in the data manifold,

$$B^*(X, 1) = \{v \in T_X \mathcal{M} : \|v\|_f = 1\}.$$

This corresponds to the preimage of a unit ball in $T_{f(X)}\mathcal{N}$. Notably, the equation $1 = \|v\|_f = \sum \lambda_i \langle v, q_i \rangle^2$ indicates that the pull-back unit ball forms an $m$-dimensional ellipsoid with semi-axes $q_1, q_2, \ldots, q_m$, and the lengths of these semi-axes are given by $\frac{1}{\sqrt{\lambda_1}}, \frac{1}{\sqrt{\lambda_2}}, \ldots, \frac{1}{\sqrt{\lambda_m}}$.

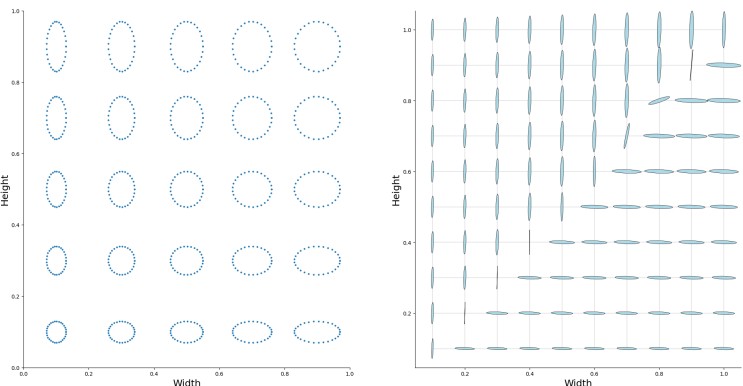

Figure 19: Visualization of the pull-back metric on a toy data set. Left: The space of point clouds sampled from ellipses of width $w$ and height $h$. Right: The pull-back unit ball at different locations on the space of ellipses. Shorter semi-axes of the pull-back unit ball correspond to larger eigenvalues of the Jacobian.

In the right panel of Figure 19 we plot the pull-back unit balls for the encoding $f$ given by the 1-dimensional PI with respect to the Vietoris-Rips filtration. The plot reveals that the eigenvectors of the encoding associated with the larger eigenvalue consistently align with the direction of increasing $\min\{w, h\}$. This alignment is

in accordance with what we would expect the Vietoris-Rips filtration to capture on this specific data set, since the death value depends on the radius of the inner circumcircle of a hole. Consequently, variations in the length of the major axis have minimal impact on PI. Conversely, altering the length of the minor axis directly affects the death parameter and changes the PI.

## E    Point saliency maps for PH encodings

In the context of interpretability, point saliency maps are commonly used tools to explain the decisions made by trained models (Montavon et al., 2018; Zheng et al., 2019). These maps assign importance scores to each point in an input point cloud (or to each pixel in an input image), indicating their significance in relation to the model's prediction. In our study, we employ a similar strategy to visualize the importance of each point in an input point cloud with respect to the PH encoding.

Given a point cloud $X$ and an encoding map $f$, we define the encoding saliency score for each point $x_i \in X$ as

$$s_f(x_i) = \left\| \frac{\partial f}{\partial x_i} \right\|_F,$$

where $\| \cdot \|_F$ denotes the Frobenius norm, which is the generalized Euclidean norm for matrices. Here, $\frac{\partial f}{\partial x}$ represents the Jacobian matrix of the encoding, which is a matrix of format $P^2 \times D$ obtained by flattening the PI into a vector in $\mathbb{R}^{P^2}$ and taking the partial derivative of each output pixel with respect to each coordinate of each point in the input point cloud. Similarly, $\frac{\partial f}{\partial x_i}$ is the vector of partial derivatives of the encoding with respect to the coordinates of the $i$-th point $x_i$ in the point cloud. This score quantifies the sensitivity of the representation to variations on the coordinates of $x$.

In Figure 20, we plot the point saliency score with respect to the Rips used in Section 5 for point clouds sampled from eight synthetic curves in $\mathbb{R}^2$. For each point cloud we highlight individual points according to their saliency scores. We observe that endpoints and points related to the inner circumcycle are often highlighted. These points correspond to the simplices that create/destroy certain homology classes in the filtration. For instance, in the curved rhombus in the first-row, second-column panel, the endpoints (yellow) on the upper right side are highlighted. Note that the edge between these two endpoints will connect the gap and create a homology class. Therefore, variations on these two points can significantly change the birth parameter of the corresponding homology class. Meanwhile, the two points (green) on the upper left and lower right sides are highlighted. They are related to the inner circumcycle of the rhombus, and are vertices of the triangle that destroys the homology class. Therefore variations on these two points can change the death parameter of the homology class significantly.

Before concluding this section, it's important to note that the saliency scores shown in Figure 20 are heavily influenced by the sampling process. For example, many of the highlighted endpoints shown in Figure 20

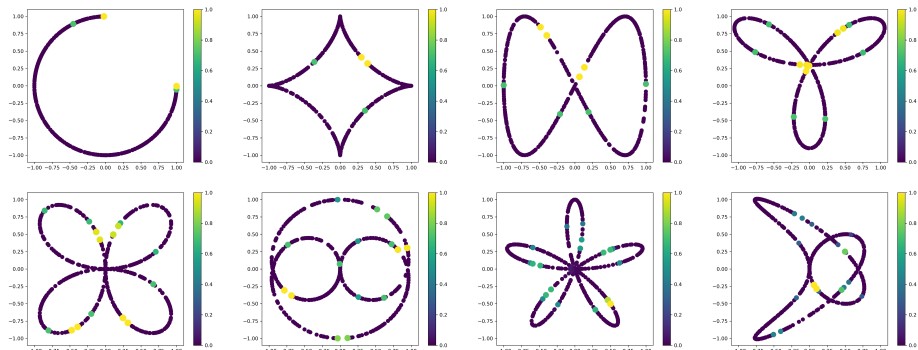

Figure 20: Point saliency score with respect to 1-dimensional PIs on Vietoris-Rips filtration for synthetic 2-dimensional point clouds.

are a result of sparse sampling. Consequently, the resulting saliency scores might be more indicative of the sampling rather than the underlying shape itself. To better comprehend the shape, one potential approach is to compute the average saliency score across multiple different samplings.

## F  Details on the experiments

### F.1  Reproducibility

The data and code developed for this research are made available at `https://github.com/shuangliang15/pullback-geometry-persistent-homology`.

### F.2  Jacobian normalization

Throughout our work, we use pull-back norm to quantify the sensitivity of the encoding method to data variations. One needs to be careful when comparing the pull-back norms induced by different encodings, since encodings may have different scaling levels. For example, consider Euclidean data space $\mathcal{M} = \mathbb{R}^2$, and two encoding mappings $f_1 = x + 0.1y$, $f_2 = x + 10y$. The pull-back norms of a tangent vector $v = [1,0]^T$ with respect to $f_1$ and $f_2$ are

$$\|v\|_{f_1} = \|J^{f_1} v\| = \left\| \begin{pmatrix} 1 & 0 \\ 0 & 0.1 \end{pmatrix} \cdot [1,0]^T \right\| = 1,$$

$$\|v\|_{f_2} = \|J^{f_2} v\| = \left\| \begin{pmatrix} 1 & 0 \\ 0 & 10 \end{pmatrix} \cdot [1,0]^T \right\| = 1.$$

In terms of "absolute sensitivity", $f_1$ and $f_2$ has the same level of sensitivity to variation $v$. More specifically, when variation $v$ is applied to a data point $X$, both $f_1(X)$ and $f_2(X)$ would change with distance approximately 1 in the representation space. However, in terms of "relative sensitivity", $f_1$ is more sensitive to $v$. The reason is that for $f_1$, the vector $v$ is the eigenvector of the Jacobian with the largest eigenvalue; while for $f_2$, $v$ is the eigenvector with the smallest eigenvalue. Equivalently, for $f_1$, $v$ has the largest pull-back norm among all tangent vectors with the same norm as $v$, whereas for $f_2$, $v$ has the smallest pull-back norm.

In Section 4.3, our goal is to study and compare the *focus* of different encodings. In Section 5.1.2 we search for the encoding whose primary *focus* is on the data variations of interest. Hence, in both sections we remove the scaling factor by considering the normalized Jacobian. Specifically, we divide the Jacobian matrix by its largest singular value, $\tilde{J}^f = J^f / \lambda_1^f$. For vector fields $V$ we consider the normalized average pull-back norm:

$$\frac{1}{|\mathcal{D}|} \sum_{X \in \mathcal{D}} \|\tilde{J}^f \cdot V(X)\|.$$

Returning to the previous example, the normalized pull-back norm for vector $v$ with respect to $f_1$ and $f_2$ are $\frac{\|J^{f_1} v\|}{\lambda_1^{f_1}} = \frac{1}{1} = 1$ and $\frac{\|J^{f_2} v\|}{\lambda_1^{f_2}} = \frac{1}{10} = 0.1$, respectively.

### F.3  Identifying what is recognized

We provide details for the experiments in Section 4.

**Radial Frequency Pattern data set**  Figure 21 shows some examples in the Radial Frequency Patterns (RFP) data set $\mathcal{D}_{\mathrm{RFP}}$.

**PH parameters**  For the Vietoris-Rips filtration, we set *maximal_edge_length* as 1. For the DTM filtration, we set *maximal_edge_length* as 0.5 and parameter $m$ as 0.02. For the Height filtration, we set *maximal_edge_length* as 0.1. We note that we need to set the maximum edge length to a small value to be able to capture topological features of interest (for instance, one wants to avoid connecting different outer regions of petals); alternatively, one could use geodesic distances or cubical complexes, see also (Turkeš et al.,

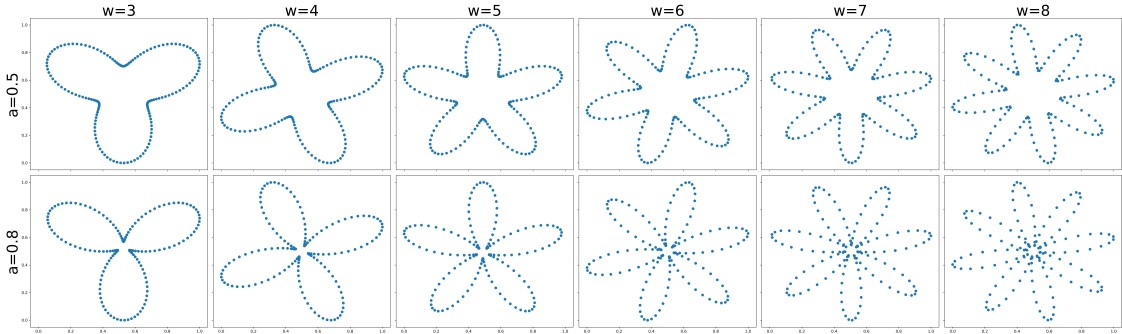

Figure 21: Visualization of the Radial Frequency Pattern (RFP) data.

2022, Fig.20) for a detailed discussion. Notice there's no need to set a small maximum edge length for Rips and DTM filtrations since the filtration value of an edge in Rips and DTM filtrations takes the distance between its two vertices into account. In contrast, in Height filtration the filtration value of an edge is defined by the maximal filtration value of its vertices, which implies any two points will be immediately connected by an edge between them, if exists, after they appear in the filtration. For the construction of PI, we set the resolution $P$ as 20, variance $\gamma^2$ of the Gaussian kernel as $10^{-4}$, and the range of the image as $[0, 1] \times [0, 1]$. The weighting function is set as $\alpha(b, l) = l$. The implementation utilized Tensorflow version 2.12.0 and Gudhi (The GUDHI Project, 2020) version 3.8.0.

**Visualization of the effects of perturbation on PH**   We visualizes the effects of *shearing* perturbation and *convex* perturbation on the persistent homology with respect to Rips filtration (see the upper panel in Figure 22) and Height filtration (see the lower panel in Figure 22). We omit plots associated with DTM filtration as they closely resemble those associated with Rips filtration. Please note that while these visualizations provide insights into the effects of perturbations on PH, they are highly dependent on the specific point clouds under consideration. In fact, the effects of perturbations on PH can vary significantly across different point clouds.

As shown in the upper penal in Figure 22, the *shearing* changes the sparsity of points in the point clouds and hence changes the birth values ($x$ coordinates) of certain points in PD with respect to Rips filtration. Meanwhile, *shearing* changes the size of petals and hence changes the death values ($y$ coordinates) of certain PD points with respect to Rips filtration. On the other hand, *convex* has the effect of "opening up" the central region of the point clouds. Consequently, some loops appear later in the filtration, notably the five loops that already exist in the second column of the first row but do not show up in the second column of the third row. Consequently, *convex* also induces changes in the birth values of PD points associated with these loops. We note that, in comparison to *shearing*, the *convex* perturbation has a more significant impact on PI. This is consistent with the findings illustrated in Figure 8, where Rips is more sensitive to *convex* compared to *shearing*.

In the lower panel of Figure 22, we can observe distinct effects of these two perturbations on the PH associated with the Height filtration. The *shearing* perturbation directly changes the $x$ coordinates of the points in the original points clouds, which correspond to their filtration values, and consequently changes the PI noticeably. On the other hand, the *convex* perturbation changes the PH in a more significant way. Specifically, certain small loops that initially appeared in the filtration (as seen in the first row of the lower panel in Figure 22) no longer appear in the *entire* filtration after perturbation (as seen in the third row of the lower panel in Figure 22). This observation arises from the small value we assigned to the *maximal_edge_length* parameter for Height filtration. In fact, for some point clouds, the central points originally have a distance smaller than *maximal_edge_length*. Then *convex* perturbation can pull these central points apart, causing their distance to exceed the *maximal_edge_length* threshold. Consequently, edges connecting these central points vanish from the filtration, resulting in the disappearance of certain small loops. In such cases, as shown in Figure 22, the *convex* can significantly change the PI associated with Height filtration. However, for other point clouds where the central points initially have a distance greater than *maximal_edge_length*, the corresponding PI

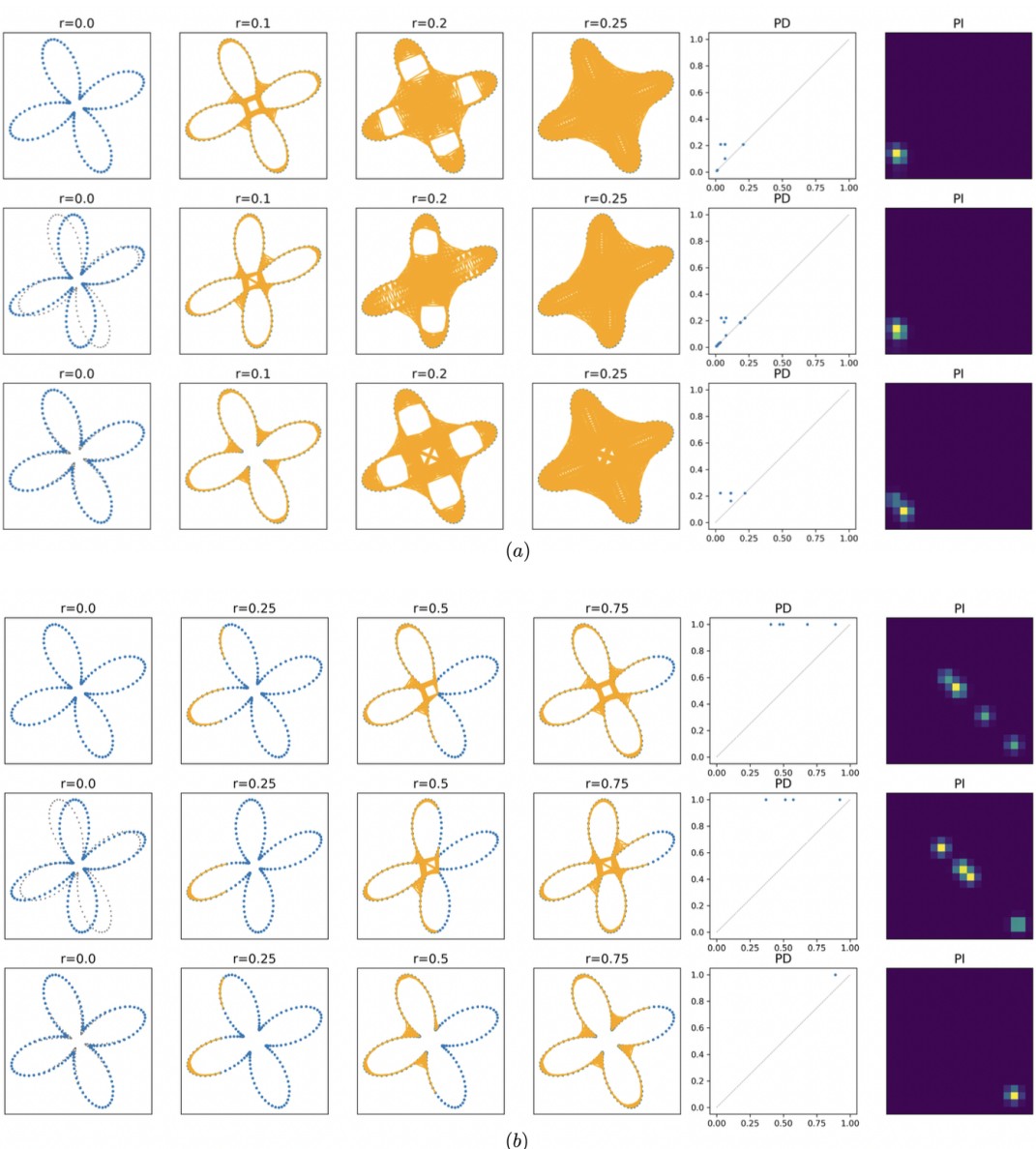

Figure 22: Figure (*a*) visualizes the effects of different perturbations on persistent homology with respect to Rips filtration. The first four columns illustrate the simplicial complex $K_r$ in the Rips filtration with parameter $r$ ranging from 0.0 to 0.25. The fifth and sixth columns show the corresponding PDs and PIs, respectively. The first row is associated with the original point cloud; while the second and third rows are associated with the point cloud perturbed by *shearing* and *convex*, respectively. Figure (*b*) visualizes the effects of perturbations on persistent homology with respect to Height filtration, following the same row-column arrangement as in Figure (*a*).

may remain relatively unchanged after the *convex* perturbation. This is again consistent with the results shown in Figure 8, where the pull-back norms of the *convex* perturbation associated with Height filtration exhibit a significant range of variation.

**PointNet architecture and training details** In our experiments, we labeled each point cloud data $X \in \mathcal{D}_{\mathrm{RFP}}$ with the number of petals, i.e. the parameter $w$ of the curve $\mathrm{RFP}_{a,w}$ from which $X$ is sampled.

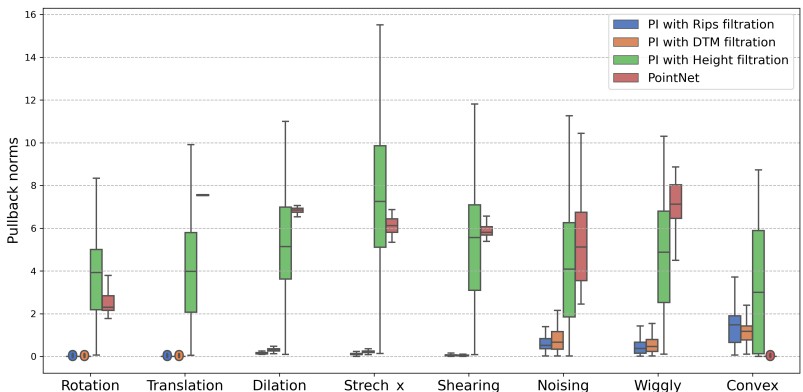

Figure 23: Unnormalized average pull-back norm of different perturbation vector fields with respect different encodings.

This produces 8 classes in total and we train the PointNet model to classify each point cloud in the data set. The PointNet model consists of a 1-dimensional convolutional layer with 64 filters and kernel size 1, followed by batch normalization and rectified linear unit (ReLU) activation. Then global max pooling is applied to obtain a permutation-invariant representation. This is followed by two fully connected layers with 128 and 64 hidden units, respectively, with ReLU activation. The final output layer uses softmax activation to produce theprobability distribution over the output classes. We trained the model with a batch size of 32 for 100 epochs, using a learning rate of 0.001. The optimization algorithm used was Adam, and the model was trained using the cross-entropy loss function. We note data augmentation techniques are used for training PointNet in literatures (see, e.g., Qi et al., 2017). However, we do not augment the data during training, in order to ensure a fair comparison with other encodings. The implementation utilized Tensorflow version 2.12.0.

**Perturbation vector field estimation** Let $\mathcal{D} = \{X_i\}_{i \in \mathcal{I}}$ be a finite data set of point clouds, and $\pi : \mathcal{M} \to \mathcal{M}$ a perturbation mapping defined on the data manifold. In the case where the data lies in Euclidean space, i.e. $\mathcal{M} = \mathbb{R}^m$, one can compute the perturbation vectors as following:

$$V_\pi(X) = \pi(X) - X, \quad \forall X \in \mathcal{D}.$$

However, in the case of general Riemannian data manifold, the subtraction between any two points on the manifold may not be well-defined. To address this, we control the perturbation mapping such that for every $X \in \mathcal{D}$ the perturbed point cloud $\pi(X)$ lies in a small neighborhood of $X$ and calculate the perturbation vector via the local coordinate system. Specifically, as shown in Appendix B.1, for each point cloud $X \in \mathcal{D}$, one can find a neighborhood $X \in U_X \subset \mathcal{M}$ and an injective isometry $\xi_X : U_X \to \mathbb{R}^m$. We control the perturbation mapping sends every point $X$ to $U_X$, i.e.,

$$\pi(X) \in U_X, \quad \forall X \in \mathcal{D}.$$

Then the perturbation vectors can computed with the subtraction on the Euclidean domain $\xi_X(U_X)$ as follows:

$$V_\pi(X) = \xi_X(\pi(X)) - \xi_X(X), \quad \forall X \in \mathcal{D}.$$

**Unnormalized pull-back norms** As discussed in Appendix F.2, we consider the normalized average pull-back norms for perturbation tangent vector fields in Section 4.3. We present the results of unnormalized average pull-back norms in Figure 23. Note that one could conclude from Figure 23 that, on average, Height is more sensitive to *convex* than Rips in terms of "absolute sensitivity". However, we reach the opposite conclusion from Figure 8 in terms of "relative sensitivity".

### F.4 Selecting hyperparameters

We provide further details on Section 5.

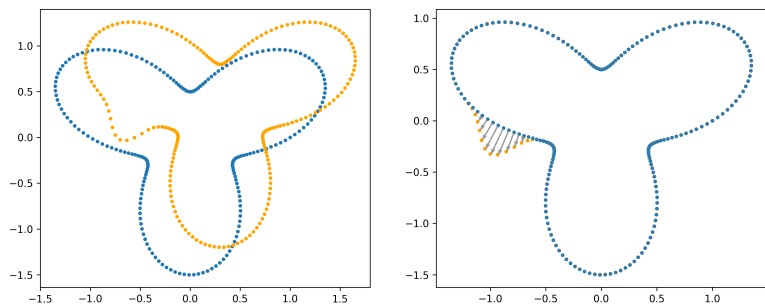

Figure 24: Left: shown are two point clouds in the plane. Right: the orange point cloud is transformed by iterated closest point algorithm and the grey vectors represent the difference vector between blue point clouds and the transformed orange point cloud.

**PH parameters**    In this section, we focus on PH encoding constructed on Vietoris-Rips filtration. We set the parameter *maximal_edge_legnth* as 0.25, and the range of PI as $[0.0, 0.25] \times [0.0, 0.25]$. The implementation utilized Gudhi (The GUDHI Project, 2020) version 3.8.0 and Tensorflow version 2.12.0.

**Gradient vector field estimation**    Let $\mathcal{D} = \{X_i\}_{i \in \mathcal{I}}$ be a finite data set of point clouds, and $\rho(X_i)$ be the corresponding feature values of point clouds in $\mathcal{D}$. In the case when the data lies in Euclidean space, we can estimate the gradient vectors with the finite difference method (FDM) as follows:

$$\overline{\nabla \rho(X)} = X' - X, \quad X' = \underset{Y \in \mathcal{D}}{\mathrm{argmax}} \frac{|\rho(Y) - \rho(X)|}{d_E(X, X')}, \quad \forall X \in \mathcal{D}.$$

For binary categorical feature, i.e. $\rho(X) \in \{0, 1\}$, the formula can be modified as

$$\overline{\nabla \rho(X)} = X' - X, \quad X' = \underset{Y \in \{Z \in \mathcal{D}: \rho(Z) \neq \rho(X)\}}{\mathrm{argmin}} d_E(X, Y), \quad \forall X \in \mathcal{D}.$$

However, in the case of general Riemannian data manifold, the subtraction between any two points on $\mathcal{M}$ may not be well-defined. To address this, when estimating gradient vector located at $X$, we send other point clouds in the data set to a neighborhood of $X$ via transformation that preserves the feature value. In our experiment, we use Euclidean transformation since the *sex* feature is irrelevant to the position or orientation of the brain artery trees. Then FDM can be applied to the transformed data set, via the coordinate system on that neighborhood.

Specifically, we utilize the iterated closest point (ICP) method (Chen & Medioni, 1992; Besl & McKay, 1992). Let $E(\mathbb{R}^D)$ be the collection of all Euclidean transformations in $\mathbb{R}^D$

$$E(\mathbb{R}^D) = \{\iota : \mathbb{R}^D \to \mathbb{R}^D : d_E(x, y) = d_E(\iota(x), \iota(x)), \forall x, y \in \mathbb{R}^D\}.$$

Note each $\iota$ can be naturally extended to a transformation on point clouds: $\iota(X) \triangleq \{\iota(x) : x \in X\}$. Let $\zeta$ be a error function in the sense that $\zeta(X, Y)$ measures the "difference" between $X$ and $Y$. Given two point clouds $X$ and $Y$, the ICP algorithm searches the Euclidean transformation that gives the minimal error value:

$$\iota_{X,Y} = \underset{\iota \in E(\mathbb{R}^D)}{\mathrm{argmin}} \zeta(X, \iota(Y)).$$

Here $\iota_{X,Y}$ is Euclidean transformation found by ICP algorithm for point clouds $X$ and $Y$. Define the ICP discrepancy between point clouds $X$ and $Y$ (not necessarily a distance) as

$$d_{\mathrm{ICP}}(X, Y) = d_W(X, \iota_{X,Y}(Y)),$$

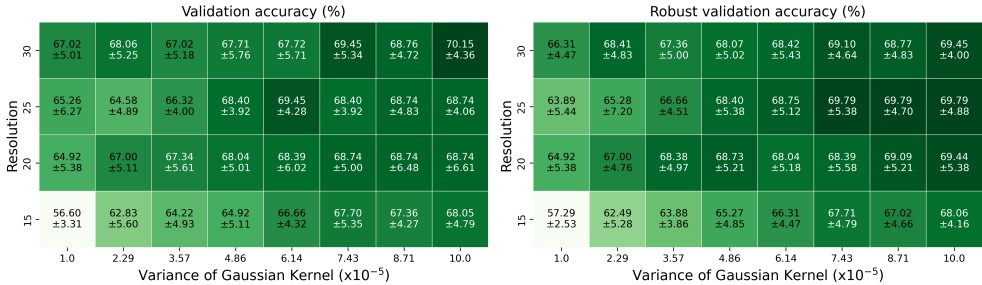

Figure 25: Right: test accuracy of the convolutional neural network (CNN) trained with PIs produced by different parameter settings. Left: robust test accuracy of the convolutional neural network (CNN) trained with PIs produced by different parameter settings.

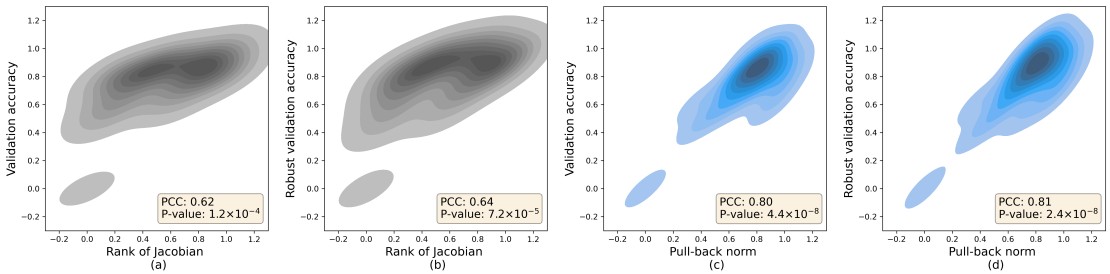

Figure 26: Gaussian kernel density estimation of the joint distribution of four pairs of variables: (a) Jacobian rank vs. validation accuracy; (b) Jacobian rank vs. robust validation accuracy; (c) pull-back norms vs. validation accuracy; and (d) pull-back norms vs. robust validation accuracy, where the downstream models are chosen as convolutional neural networks.

where $d_W$ is Wasserstein distance between point clouds. Let $U_X$ be a neighborhood of $X$ and $\xi_X$ a coordinate map $\xi_X : U_X \to \mathbb{R}^m$. Estimate the gradient vector field for binary categorical feature $\rho$ as follows (see an illustration in Figure 24):

$$\overline{\nabla \rho(X)} = \xi_X(\iota_{X,X'}(X')) - \xi_X(X), \quad X' = \underset{Y \in \{Z \in \mathcal{D} : \rho(Z) \neq \rho(X)\}}{\operatorname{argmin}} d_{\mathrm{ICP}}(X, Y), \quad \forall X \in \mathcal{D}.$$

The implementation utilized python library Open3D (Zhou et al., 2018) version 0.17.0 and POT (Flamary et al., 2021) version 0.9.0.

**Downstream tasks and performance** In Section 5, we fed PIs produced by different parameter settings to logistic regression models to predict the *sex* feature. Specifically, we normalize the PIs such that the pixel values range within $[0, 1]$. We implemented logistic regression models using Scikit-learn (Pedregosa et al., 2011) version 1.2.2 with default hyperparameters. When evaluating the model, we use a 7-fold cross validation. And for robust evaluation in Section 5.1.3, we add identically and independent distributed Gaussian noise with variance $10^{-2}$ to each coordinate of each point in input point clouds.

Here, we also investigate the effects of resolution and variance of Gaussian kernel on the downstream performance of convolutional neural network (CNN). The CNN model takes PIs as inputs, then begins with a convolutional layer with 32 filters and a $3 \times 3$ kernel, followed by a ReLU activation function. Then max pooling with a pool size of $2 \times 2$ is applied. Subsequently, another convolutional layer with 64 filters and a $3 \times 3$ kernel is added, also followed by a ReLU activation and max pooling. The resulting outputs are then flattened and passed through a fully connected layer with 64 neurons and ReLU activation. Finally, a single neuron with a sigmoid activation function is used for binary classification. To train the model, we employ Adam optimizer with the cross-entropy loss function. The implementation utilized Tensorflow version 2.12.0.

We represent the downstream performance of the CNN model in Figure 25. Shown in Figure 26 is the kernel density estimation of the joint distribution of four pairs of variables: Jacobian rank vs. validation accuracy, Jacobian rank vs. robust validation accuracy, pull-back norm vs. validation accuracy, and pull-back norm vs. robust validation accuracy. Additionally, the Pearson correlation coefficient and p-value of a two-sided test are presented at the lower right corner of each point in Figure 26. We observe that the correlation between pull-back norms and downstream performances remains significant.

We conjecture that complex models, such as CNNs, are able to obtain good downstream performance even if the average pull-back norm is low, so long as it is not zero. Intuitively, when the feature information is indeed contained in the encoded representation but is not significantly pronounced, a simple model may not be able to extract this information, but a complex model, along with appropriate training techniques, can still learn to extract and utilize this information.

## G    Investigating which part of the data is highlighted by PH encodings

In this section, we demonstrate how our method facilitates investigating which *part* of the data is the focus of PH encodings. To this end, we will introduce noising perturbation on different *parts* of the point clouds and examine the average pull-back norm of these perturbation vector fields. Moreover, we also consider the beta weighting function for PIs and investigate the effects of the beta mean parameter on the pull-back geometry, which allows comparing the focus of long and short persistence intervals.

**Human body data**   We utilize the benchmark mesh segmentation data (Chen et al., 2009). This data set consists of meshes representing 19 different types of 3-dimensional shapes, each annotated with manually added segmentation labels. For our analysis, we focus on the subset of meshes representing the human body, which encompasses various gestures such as standing, walking, and sitting. We randomly subsample three point clouds from the vertices of each human body mesh, with each point cloud containing $N = 500$ points.

**PH encoding**   We focus on the PH encoding that sends each point cloud to its 2-dimensional PI with respect to the Vietoris-Rips filtration. We choose 2-dimensional PH because we note that the underlying geometric objects of human body meshes are 2−dimensional surfaces whose 1-dimensional homology is typically zero, and whose reduced 0-dimensional homology is zero as well. For the construction of PIs, we set the PI hyperparameters as $P = 20, \gamma = 1 \times 10^{-4}$, *maximal_edge_legnth*= 0.3, and the range of PI as $[0.0, 0.3] \times [0.0, 0.3]$. For the weighting function, we again employ the beta weighting function that we introduced in Section 5.2. We set the variance parameter for the beta weighting function $s^2$ as 0.04 and consider the mean parameter $k$ ranging from 0.1 to 0.5. We present a visualization of the impact of the mean parameter on the PIs in Figure 27.

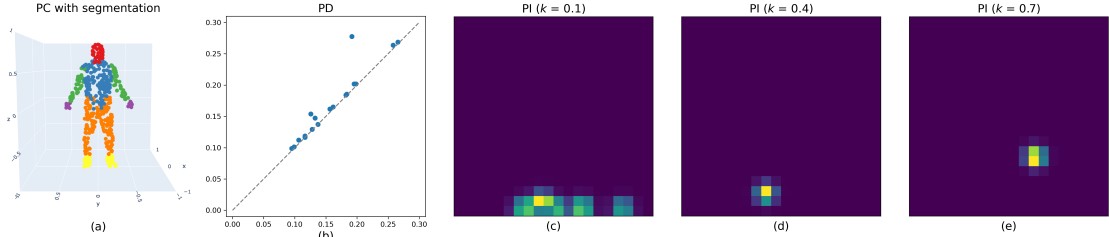

Figure 27: Illustration of the human body data set: (a) Point cloud with body parts segmentation; (b) 2-dimensional persistence diagram with respect to the Vietoris-Rips filtration; (c) to (e): persistence images obtained with beta functions with the beta mean parameter set as 0.1, 0.4, and 0.7 respectively.

**Perturbation**   We merge the segmentation labels for the sampled human body point clouds into 6 categories: head, torso, arms, hands, legs and feet (see panel (a) in Figure 27 for an illustration). Accordingly, we consider 6 types of perturbations, each adding independent Gaussian noise to points in one body part. We provide visualizations of some of the perturbations in Figure 28.

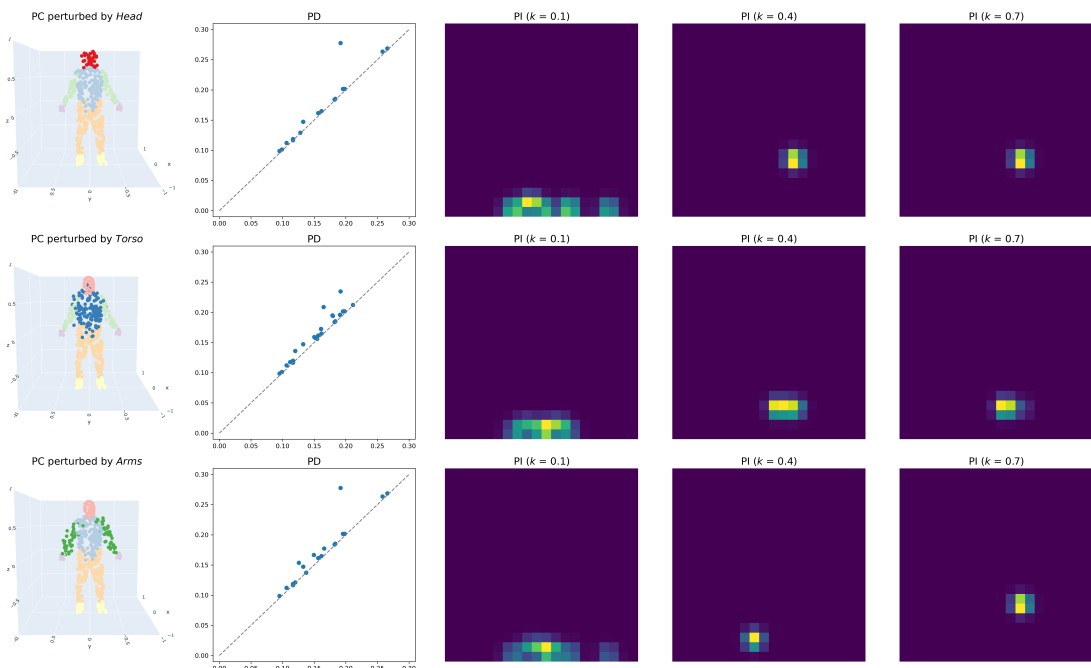

Figure 28: Illustration of the effects of noising perturbation on *Head*, *Torso*, and *Arms* on the point cloud, PD and PIs with different weighting functions .

**Pull-back norm**   We evaluate the average pull-back norms of the noising perturbation on 6 body parts and present the results in Figure 29.

It is noticeable in Figure 29 that persistence intervals of varying lengths capture distinct aspects of the data. Specifically, when the mean parameter is set to 0.1, the encoding exhibits significant sensitivity to perturbations on *legs*, when $k = 0.2$, the focus of the encoding switches to *head*. For larger mean parameters, the encoding becomes most sensitive to perturbations on *torso*. This could be explained by observing that shorter persistence intervals in 2-dimensional PDs on Rips filtration are more related to smaller hollow shapes in the data, such as arms and head, while longer intervals relate more to larger hollow shapes, such as the torso.

These findings can also guide the selection of hyperparameters for PH encodings. For instance, in a face recognition task, we know from above that shorter persistence intervals are sensitive to data variations on *head*. Hence, one should choose a value around 0.2 for the beta mean parameter in order to obtain a data presentation that is most suitable for this task. At the same time, we note once again that long persistence intervals do not always contain the most important information and that the optimal choice of the weighting function (and other hyperparameters) depend on the specific task at hand.

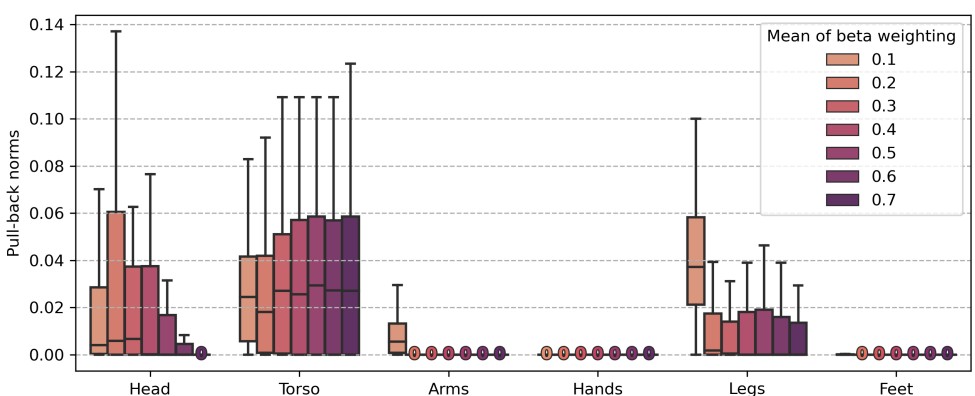

Figure 29: Averaged pull-back norms of noising perturbations on 6 body parts.

