# OpenReview forum: "Pull-back Geometry of Persistent Homology Encodings"
_TMLR — Accepted by TMLR_

### Review · Reviewer_DuMc · 2023-11-25

**Summary Of Contributions:**

This paper focuses on examining persistence homology (PH) encodings, which are mappings from data to vectorized PH representations. Persistence homology is a powerful tool in topological data analysis (TDA) that captures structural features in data (e.g., holes, voids) at multiple scales. The output of PH is often mapped into a vector space for further analysis or machine learning tasks. Persistence images (PIs) represent a popular example of such mapping. This paper considers PH encodings derived from PIs based on three distinct filtrations: Vietoris-Rips, distance-to-measure, and height filtration. Each of these filtrations offers a different perspective on the data, capturing different aspects of its topological structure at different scales.

The authors explore how the choice of filtration impacts the resulting PH encoding and, consequently, the information captured about the data. They demonstrate that through the measurement of various perturbations on the data manifold in relation to pull-back geometry, it becomes possible to identify which of them are captured and which are overlooked by the PH encodings.

Furthermore, this paper addresses the issue of parameter selection for the PH encodings. Effectively selecting parameters is crucial in obtaining meaningful and task-specific representations of the data. The authors demonstrate, on real-world data, the effectiveness of selecting PH encoding parameters by taking into account the pull-back norm of features of interest.

**Audience:**

Yes

**Claims And Evidence:**

Yes

**Requested Changes:**

On page 1, PH encoding is defined as the mapping from data to vectorized representations of the associated PH. In a few sentences later persistence images (PIs) are called examples of PH encodings, but PIs are mappings from persistence diagrams and not from the original data. Also, on page 8, the three filtrations were also called PH encodings.  While I understand what the authors meant by that, at first it caused some confusion, thus all these instances have to be carefully rephrased.

On page 2, I suggest changing "vectorized representations of the associated PH" to "vectorized PH representations".

In the paper, the authors say “it is unclear which aspects of the data are highlighted or which ones are suppressed in the PH representations’’ and there are similar statements throughout the paper. This is not completely true since the representative cycles can help interpret the PH results by identifying and visualizing essential topological features and patterns in data. I suggest rephrasing these sentences and maybe the authors can consider adding a short discussion of representative cycles.

On page 4, the authors mention “k-dimensional PD” and “k-dimensional homology class”. However, the paper lacks the definition of k-dimensional homology. Including the definition of k-dimensional homology would be beneficial, particularly for readers who may not be familiar with TDA.

**Strengths And Weaknesses:**

The paper addresses important TDA questions and brings value to both TDA and machine learning community. While it is well organized and written, there are a few improvements that can be made to enhance the paper's clarity.

---

> ### Author Response · Authors · 2024-01-27
> **Part I. Improvement on terminology**
>
> > Requested change 1) On page 1, PH encoding is defined as the mapping from data to vectorized representations of the associated PH. In a few sentences later persistence images (PIs) are called examples of PH encodings, but PIs are mappings from persistence diagrams and not from the original data. Also, on page 8, the three filtrations were also called PH encodings. While I understand what the authors meant by that, at first it caused some confusion, thus all these instances have to be carefully rephrased.
>
> Thank you so much for pointing this out. We went through the paper once again in detail, and made sure to carefully rephrase all instances of *encoding* and *representation*. Regarding *PH encoding* and *PI*, we have rewritten the corresponding part in the Introduction:
>
> "... Here, a PH *encoding* denotes the mapping from data to a vectorized PH representation."
>
> and
>
> "We center our attention on a widely used PH representation known as the persistence image (PI) [1], which is an image-like representation of the input data in terms of multiscale homological features."
>
> Regarding PH encoding and filtration, we have rephrased the "PH encoding" paragraph in Section 4:
>
> "We investigate PH encodings constructed on $3$ different filtrations: Vietoris-Rips (Rips) filtration, DTM filtration, and Height filtration (with respect to the hyperplane with normal vector $[1,0]^T$)."
>
> and also
>
> "In the following discussion, we sometimes refer to these encodings by the name of the filtration on which they are constructed, denoting for instance the PH encoding constructed on Rips filtration simply as Rips."
>
> &nbsp;
>
> > Requested change 2) On page 2, I suggest changing "vectorized representations of the associated PH" to "vectorized PH representations".
>
> Done! Please see our response to your previous Requested change 1).
>
> &nbsp;
>
> [1] Adams, Henry, et al. "Persistence images: A stable vector representation of persistent homology." Journal of Machine Learning Research 18 (2017).

---

> ### Author Response · Authors · 2024-01-27
> **Part II. Representative cycles and interpretability**
>
> > Requested change 3) In the paper, the authors say "it is unclear which aspects of the data are highlighted or which ones are suppressed in the PH representations" and there are similar statements throughout the paper. This is not completely true since the representative cycles can help interpret the PH results by identifying and visualizing essential topological features and patterns in data. I suggest rephrasing these sentences and maybe the authors can consider adding a short discussion of representative cycles.
>
> We agree that the mentioned statement might have been confusing, we have removed it from the paper, and have rewritten the Abstract and Introduction to better position our work. In the revised manuscript, we now also explicitly mention the representative cycles in the Related work:
>
> "Finally, an important line of work in the study of PH encodings is concerned  with developing computable notions of optimal representative cycles for persistent homology classes, see, e.g., the survey of [1]."
>
> &nbsp;
>
> [1] Li, Lu, et al. "Minimal cycle representatives in persistent homology using linear programming: An empirical study with user's guide." Frontiers in Artificial Intelligence 4 (2021): 681117.

---

> ### Author Response · Authors · 2024-01-27
> **Part III. Definition for k-dimensional homology**
>
> > Requested change 4) On page 4, the authors mention “k-dimensional PD” and “k-dimensional homology class”. However, the paper lacks the definition of k-dimensional homology. Including the definition of k-dimensional homology would be beneficial, particularly for readers who may not be familiar with TDA.
>
> Thank you for pointing this out. We now include a brief description of k-dimensionaly homology and k-dimensional homology class in Section 2:
>
> "The k-dimensional homology group, or k-dimensional homology, of a simplicial complex characterizes the k-dimensional holes in the complex. Each non-zero k-dimensional homology class in the k-dimensional homology group uniquely characterizes a k-dimensional hole. As we introduced earlier, the set $K_r$ includes more and more simplices as the parameter $r$ increases. In persistent homology, we are interested in how the homology groups of $K_r$ changes as we vary the parameter $r$."

---

> ### Author Response · Authors · 2024-02-08
> **Regarding our previous responses**
>
> Thank you again for your comments on our initial submission, helping us in making the revisions outlined above. We have sought to systematically address all concerns and requested changes and believe this has strengthened the paper. Please be informed that we have made additional modifications and have now uploaded the latest version of the manuscript. We would kindly like to inquire if you have found the revisions adequately addressed your concerns or if you have any additional suggestions or feedback.

---

### Review · Reviewer_LaLp · 2023-11-27

**Summary Of Contributions:**

The authors consider some case studies in evaluating the Jacobian of persistence images generated by particular sets of point cloud data. In particular, they consider the RFP and brain artery tree datasets, and look at the spectrum and eigenvectors of the Jacobian. Roughly speaking, high rank is desired for the Jacobian and high magnitude eigenvectors provide some intuition of the point cloud perturbations that are detected by persistence images.

**Audience:**

Yes

**Broader Impact Concerns:**

Not applicable.

**Claims And Evidence:**

Yes

**Requested Changes:**

I do not currently feel like recommending the submission for acceptance due to the above questions and weaknesses. The major point to address is the need for more impactful applications and insights in those particular instances.

Overall, I do find the positioning of the work to be a little perplexing. The method of analysis is not at all restricted to embeddings generated by persistence homology, so it might be useful to consider its use in other settings (like the neural network ones considered). Moreover, embeddings via persistence homology are some of the most interpretable, I feel, at least relative to those from deep learning.

**Strengths And Weaknesses:**

Strengths:
1. The presentation is sensibly organized, with much background given on persistence homology/diagrams, and various filters.
2. The basic perspective is sound, and it's clear that some insights might be garnered by looking at the Jacobian and its spectrum.

Weaknesses:
1. The contribution is limited by the fact that they only look at particular datasets, and the insights garnered are apparent or might be more easily gained with performance-based testing. As a result, I feel like the work is overclaiming slightly with the generality that it alludes to in its abstract and intro.
    1. I think it's a stretch to claim that the interpretations of what persistence homology and its representations capture is unclear. It is even stated several times that it captures "holes" of various dimensions in the data, as they are formed with respect to the filtration at hand. Each of the filtrations is also relatively easily understood. This leads to conclusions in many of the experiments that are relatively obvious: that higher-resolution persistence images lead to higher-rank Jacobians and that Rips and DTM are invariant to rotation and translation, for example.
    2. It's unclear how many empirical datasets or point clouds are to be considered before one has a fair idea of what perturbations are valued. Doing a performance-based evaluation would simultaneously achieve this (empirical) aim and train the model at hand. This can be seen in the evaluation of the PointNet model in the RFP experiments.
2. While the presentation is well-organized, the writing of the definitions and the perspective of the work leaves something to be desired.
    1. It would seem relevant to at least note briefly that persistence diagrams lie above the diagonal $y=x$ by definition, and to show some examples, to give readers intuition and so that they know the information in a $P \times P$ image is half of what they'd expect it to be. Actually, it's a little unclear to me how the kernel smoothing behaves with respect to this limited domain, and this should be clarified.
    2. In Def. 3, I think it'd be useful to remind readers that $X$ denotes a point cloud and that $\mathcal{D}$ denotes a collection of point clouds that form your data. Given that a tangent vector in the space of point clouds corresponds to a "vector field" in the sense of a perturbation vector per point, it's easy for someone to think one might be calculating norms of these individual perturbation vectors.
    3. When considering the input space of point clouds, I think it's important to note the fact that these point clouds are determined up to permutation, and they must be quotiented by the action of $S_n$. On the other hand, it's also unclear to me why the Wasserstein distance is referenced, as one intends to simply consider evaluation of the pullback metric. Lastly, as footnote 2 indicates, one should really probably consider the Wasserstein metric on the persistence images, and it's unfortunate that the method sticks to the less natural and informative Euclidean distance on persistence images.
    4. A perturbation vector field that simply points directly from a point cloud to its perturbed point cloud is only feasible when the target manifold is Euclidean, as it is here. It also may only be reflective of the perturbation if it is small (a 180 degree rotation would not generate a rotational vector field).
    5. Figure 1 is highly confusing without prior introduction of RFPs and what subset of them (or datasets sampled from them) is represented by a torus. Please clarify.

---

> ### Author Response · Authors · 2024-01-26
> **Part I: More data, Performance-based testing, Data-dependence**
>
> > Weakness 1) The contribution is limited by the fact that they only look at particular datasets, and the insights garnered are apparent or might be more easily gained with performance-based testing. As a result, I feel like the work is overclaiming slightly with the generality that it alludes to in its abstract and intro.
>
> > Weakness 1.2) It's unclear how many empirical datasets or point clouds are to be considered before one has a fair idea of what perturbations are valued. Doing a performance-based evaluation would simultaneously achieve this (empirical) aim and train the model at hand. This can be seen in the evaluation of the PointNet model in the RFP experiments.
>
> **More data**: Beside the synthetic RFP data and the real-world brain artery data set, in the revised manuscript we are now also including experiments on another real-world data set of human body postures. For details, please see [our response to Reviewer 4uUj](https://openreview.net/forum?id=7yswRA8zzw&noteId=Il9jaSA4fe).
>
> **Performance-based testing**: In the case of a specific problem and a specific solution, performance-based testing of that specific solution is definitely the best option. However, the main point here is that we want to evaluate the data encoding by itself. In this case, one may regard the pull-back norm of a vector field as the performance metric. For the case that we are interested in a particular downstream task, we posit that evaluating the data representation by itself still is useful, as this can help us better understand possible opportunities for improvement in the pipeline that would not be apparent when evaluating the entire model in a typical performance based analysis.
>
> **Data-dependence of methodology**: We recognize that the insights gained through our methodology are specific to the data sets that we considered. This is necessarily the case, since, for example, Gaussian noise and outliers can change 1-dimensional PH of a circle more than that of a disk. This would affect a performance-based evaluation in the same way. In the revised Introduction, we acknowledge the data-specificity of the insights in the following way:
>
> "We note that the insights about the (PH) encodings obtained through our approach depend on the specifics of the data set. Nonetheless, by evaluating different datasets one may be able to draw certain conclusions that hold with some generality: for instance, conclude that a particular encoding captures a particular feature in datasets of a particular type. This is an interesting prospect that can be facilitated by our proposed approach."
>
> Moreover, we highlight this once again in Section 4: "... we emphasize that this section is dedicated to investigating 'what is recognized' by PH encodings *within specific datasets*".

---

> ### Author Response · Authors · 2024-01-26
> **Part II: Low interpretability of PH and relatively obvious experiments**
>
> > Weakness 1.1) I think it's a stretch to claim that the interpretations of what persistence homology and its representations capture is unclear. It is even stated several times that it captures "holes" of various dimensions in the data, as they are formed with respect to the filtration at hand. Each of the filtrations is also relatively easily understood. This leads to conclusions in many of the experiments that are relatively obvious: that higher-resolution persistence images lead to higher-rank Jacobians and that Rips and DTM are invariant to rotation and translation, for example.
>
> We have rewritten the Abstract, Introduction and Conclusion to avoid this perception about interpretability. The goal of this paper is to present a methodology to gain intrinsic and data-specific understandings for PH encodings. Specifically, our proposed method help to address questions such as how to identify which data variations are captured by PH encodings, how to quantify the effectiveness of these representations in detecting particular data features, and how to select the parameters of PH encodings for solving a particular task.
>
> We agree that some of our experimental observations recover relatively intuitive facts, which is a good thing! We do not want to overstate our findings, however, and address this concern in two different ways. Firstly, regarding the Rips and DTM, we now discuss these experiments as a proof of concept. Secondly, when it comes to the more obvious resolution parameter of persistence image, we now consider some more interesting scenarios for the brain artery data. Rather than focusing only on the resolution and Gaussian variance of persistence images, we now also consider the less obvious parameter: the weight function. We introduce the beta weighting function which allows assigning weights to persistence intervals of different length. We study the effects of intervals of different length on detecting the *sex* feature (Figure 14 in the revised manuscript). Our result 1) strengthens our conclusion that pull-back norm is predictive for downstream task performance in classifying *sex* labels on brain artery tree data and 2) confirms persistence intervals with medium length are more important for this task, which is also observed by Bendich et al. (2016).
>
> Furthermore, the revised manuscript includes additional experiments on the real-world human body data set (Appendix G in the revised manuscript). For the 2-dimensional PI on Rips filtration, we found that noising perturbations on certain body part, for example arms, can only be detected by shorter persistence intervals.

---

> ### Author Response · Authors · 2024-01-27
> **Part III. Input space and output space**
>
> > Weakness 2.3) When considering the input space of point clouds, I think it's important to note the fact that these point clouds are determined up to permutation, and they must be quotiented by the action of $S_n$. On the other hand, it's also unclear to me why the Wasserstein distance is referenced, as one intends to simply consider evaluation of the pullback metric. Lastly, as footnote 2 indicates, one should really probably consider the Wasserstein metric on the persistence images, and it's unfortunate that the method sticks to the less natural and informative Euclidean distance on persistence images.
>
> In Section 3.1 and Appendix B.1, we defined a point cloud as an unordered finite subset in $\mathbb{R}^D$, and in Appendix B.2 we define the space of point clouds as the collection of maps from $\{1,2,...,N\}$ to $\mathbb{R}^D$ quotient by the action $S_n$. Both definitions are compatible with the fact that point clouds are determined up to permutation. To emphasize this, we rephrased and add the following sentence in Section 3.1:
>
> "A point cloud $X$ is finite subset in $\mathbb{R}^D$ with cardinality $|X| = N$. A point cloud may be regarded as an unordered list of points, determined only up to permutation."
>
> You are right that the original metric would not effect the pullback metric. However, the Wasserstein distance is essential for constructing the Riemannian manifold structure on the input space and for defining perturbation vector fields and gradient vector fields. We highlighted this better in the revised manuscript (Appendix B.4 in the revised manuscript):
>
> "Meanwhile, we point out that, while the Riemannian metric on $\mathcal{M}$ does not affect the pull-back metric, it is still necessary in our approach since it's essential in defining the perturbation vector field and gradient vector field (see Definition 11 and Definition 13)."
>
> Please note that our method can be directly applied to any other choice of distance on PI as we pointed out in Section 3.1:
>
> "Here again, other choices of the metric are possible.(footnote 4)
>
> Footnote 4: This can be of interest to try to establish more general stability results for PIs. An example would be a Wasserstein distance between persistence images assigning an appropriate cost to  $b$ and $l$ directions."
>
> We added a new Appendix section in the revised manuscript discussing this matter (Appendix B.4). We have also highlighted this more in the introduction: "... Our methodology, however, extends to other PH representations and, more broadly, to any differentiable encoding whose representation space can be endowed with a Riemannian manifold structure."

---

> ### Author Response · Authors · 2024-01-27
> **Part IV. Perturbation vector fields and local variations**
>
> > Weakness 2.4) A perturbation vector field that simply points directly from a point cloud to its perturbed point cloud is only feasible when the target manifold is Euclidean, as it is here. It also may only be reflective of the perturbation if it is small (a 180 degree rotation would not generate a rotational vector field).
>
> Note, please, we have provided the definition for perturbation vector field in the setting of general Riemannian manifolds (Definition 11 in Appendix B.3). Specifically, we define a perturbation tangent vector as the tangent vector induced by the geodesic connecting the original data point the perturbed one.
>
> Meanwhile, you are right that our method can only applied to small perturbations (here to measure "small" one also needs a distance in the space of point clouds as we addressed in [our response to your previous Weakness 2.3](https://openreview.net/forum?id=7yswRA8zzw&noteId=EfTYbCxFG4)). Throughout the manuscript we talk about "local variations", and this is explicitly acknowledged as a limitation in the Conclusion:
>
> "Our analysis is based on the structure of the Jacobian of the data encoding, which by nature focuses only on local variations of the input data. In future it will be interesting to further advance these methods in regard to non-local data variations, where synthetic notions of derivatives such as our empirical evaluation of the vector fields, and ideas such as the application of the iterated closest point method could serve as a point of departure. The analysis of non-linear transformations via Gram matrices has seen a number of recent advances in the context of artificial neural networks. It will be interesting to explore possible synergies between those investigations and PH data encodings. "

---

> ### Author Response · Authors · 2024-01-27
> **Minor comments**
>
> > Weakness 2.1) It would seem relevant to at least note briefly that persistence diagrams lie above the diagonal $y=x$ by definition, and to show some examples, to give readers intuition and so that they know the information in a $P \times P$ image is half of what they'd expect it to be. Actually, it's a little unclear to me how the kernel smoothing behaves with respect to this limited domain, and this should be clarified.
>
> In Section 2 of the revised manuscript, we added Figure 2 which illustrates the PD, transformed PD and PI for an example point cloud, that is accompanied with the following paragraph:
>
> "Note that, since the death time $d$ cannot be smaller than the birth time $b$, the birth-death pairs $(b,d)$ always lie above the diagonal line, i.e., $\text{PD}\subseteq \{(x,y)\in\mathbb{R}^2: y\geq x \geq 0\}$. The transformed birth-lifespan pairs $(b,l)$ lie in the first quadrant, i.e., $\eta(\text{PD})\subseteq \{(x,y)\in\mathbb{R}^2: x\geq 0, y \geq 0\}$ (see an illustration in Figure 2)."
>
> &nbsp;
>
> > Weakness 2.2) In Def. 3, I think it'd be useful to remind readers that $X$ denotes a point cloud and that $\mathcal{D}$ denotes a collection of point clouds that form your data. Given that a tangent vector in the space of point clouds corresponds to a "vector field" in the sense of a perturbation vector per point, it's easy for someone to think one might be calculating norms of these individual perturbation vectors.
>
> We added the following sentence below Definition 3: "Please note that in Definition 3, $V(X)$ denotes a tangent vector at $X$ in the space of point clouds. Specifically, a vector $V(X)\in T_X\mathcal{M}$ corresponds to a "vector field" on $X$ which assigns a vector to each point $x\in X$ in the point cloud $X$."
>
> &nbsp;
>
> > Weakness 2.5) Figure 1 is highly confusing without prior introduction of RFPs and what subset of them (or datasets sampled from them) is represented by a torus. Please clarify.
>
> We have edited the caption of the figure for clarity (Figure 3 in the revised manuscript): "The space of point clouds forms a manifold, which in this figure is depicted as a torus". We also added a comment to explain that: "Figure 3 is a schematic illustration. It is not intended to imply that the kind of depicted point clouds indeed form a torus."
>
> &nbsp;
>
> > I do not currently feel like recommending the submission for acceptance due to the above questions and weaknesses. The major point to address is the need for more impactful applications and insights in those particular instances. Overall, I do find the positioning of the work to be a little perplexing. The method of analysis is not at all restricted to embeddings generated by persistence homology, so it might be useful to consider its use in other settings (like the neural network ones considered). Moreover, embeddings via persistence homology are some of the most interpretable, I feel, at least relative to those from deep learning.
>
> In the revised manuscript, we have systematically worked on addressing each of your concerns. We hope that you will find the improvements satisfactory and remain attentive to any additional feedback or suggestions that you might have.

---

> ### Author Response · Authors · 2024-02-08
> **Regarding our previous responses**
>
> Thank you again for your comments on our initial submission, helping us in making the revisions outlined above. We have sought to systematically address all concerns and requested changes and believe this has strengthened the paper. Please be informed that we have made additional modifications and have now uploaded the latest version of the manuscript. We would kindly like to inquire if you have found the revisions adequately addressed your concerns or if you have any additional suggestions or feedback.

---

### Review · Reviewer_4uUj · 2023-12-28

**Summary Of Contributions:**

The paper delves into the complex realm of data representation using persistent homology (PH), a technique that accentuates topological features of data. It uniquely investigates how PH encodings affect data's manifold geometry by analyzing the Jacobian spectrum and pull-back geometry. This approach allows for assessing which data features are emphasized or overlooked by PH without needing specific task training. A significant finding is the correlation between the pull-back norm and performance in downstream tasks, suggesting a new method for selecting appropriate PH encodings. The study stands out in the field of topological data analysis, as it not only enhances the understanding of PH properties but also provides practical guidance for optimizing PH representations for varied applications.

**Audience:**

Yes

**Broader Impact Concerns:**

The paper's focus on specialized aspects of persistent homology could limit its broader impact, as its highly technical nature might restrict its accessibility and applicability beyond the niche field of topological data analysis. Expanding its scope to demonstrate relevance and potential applications in wider areas of data science and machine learning could significantly enhance its overall impact.

**Claims And Evidence:**

Yes

**Requested Changes:**

I think this is a good paper. The following general changes are recommended to enhance its contribution to the field:

1. Given the paper's advanced mathematical content, particularly around Jacobian spectrum and pull-back geometry, a more detailed explanation or simplification of these concepts would aid comprehension. This could include illustrative examples or diagrams that break down these complex ideas.

2. While the study's experimental validation, such as the analysis of brain artery trees, is robust, extending these experiments to a wider range of datasets would demonstrate the versatility of the proposed methods.

3. A section comparing the paper's approach with other existing methods in topological data analysis, especially in terms of recognizing or ignoring data variations, could contextualize its efficacy and novelty.

4. The paper would benefit from an expanded discussion on the implications of its findings in broader data science contexts, particularly in how PH encodings could influence machine learning models or data representation strategies.

5. Incorporating detailed case studies where the proposed method is applied, especially in contexts different from the brain artery trees dataset, would provide practical insights into the method's application.

6.. A more in-depth analysis of potential limitations, such as computational demands or scenarios where the method might not perform optimally, would provide a balanced view of the approach.

7. Given the paper's focus on geometry and topology, additional visual aids or enhanced graphical representations would significantly aid in understanding the complex methodologies and results.

**Strengths And Weaknesses:**

+++++**Strengths**

1.  The paper introduces a novel method to analyze persistent homology (PH) encodings by investigating the Jacobian spectrum and pull-back geometry. This approach is a significant contribution to the field of topological data analysis, offering a new lens to understand PH.

2. By focusing on how different PH encodings capture or ignore data variations, the study has practical implications. The ability to predict performance on downstream tasks using the pull-back norm is particularly useful for selecting suitable PH encodings.

3. The paper includes experimental demonstrations that bolster the theoretical findings. This empirical approach not only validates the proposed methods but also demonstrates their applicability in real-world scenarios, such as in the analysis of brain artery trees.

4. The research is presented with clarity and thoroughness, making complex concepts accessible while providing depth in analysis. This balance enhances the paper's appeal to both experts and those newer to the field.

-----**Weaknesses**

1. While the paper is well-explained, the methods' complexity could be a barrier for some readers, particularly those who are not well-versed in advanced topological concepts or mathematical formulations.

2. The focus on PH and its specific applications might limit the paper's appeal to a broader audience. Its relevance is primarily confined to specialists in topological data analysis and related fields.

3.While the paper situates itself well within its specific research domain, it could benefit from a broader contextualization in terms of its implications for other areas of data analysis or machine learning.

4. While the experiments conducted are robust, additional validation across more diverse datasets and scenarios could strengthen the findings. This would ensure the generalizability of the research.

In summary, the paper makes a significant contribution to the understanding of persistent homology encodings, with practical applications for selecting PH encodings based on the pull-back norm. Its innovative approach and thorough experimental validation are notable strengths. However, the complexity of the methods and the niche focus of the application might limit its accessibility and appeal. Further validation in diverse scenarios could enhance the robustness of the findings. Overall, the paper's strengths make it a valuable addition to the field, and its acceptance is recommended.

---

> ### Author Response · Authors · 2024-01-26
> **Part I. Clarification of methodology**
>
> > Weakness 1) While the paper is well-explained, the methods' complexity could be a barrier for some readers, particularly those who are not well-versed in advanced topological concepts or mathematical formulations.
>
> > Requested Change 1) Given the paper's advanced mathematical content, particularly around Jacobian spectrum and pull-back geometry, a more detailed explanation or simplification of these concepts would aid comprehension. This could include illustrative examples or diagrams that break down these complex ideas.
>
> > Requested Change 7) Given the paper's focus on geometry and topology, additional visual aids or enhanced graphical representations would significantly aid in understanding the complex methodologies and results.
>
> To improve the understanding of our methodology, we have now included a schematic diagram that visualizes the pipeline of the proposed method (Figure 1 in the revised manuscript).
>
> Secondly, we include a figure (Figure 2 in the revised manuscript) that visualizes the pipeline for constructing PIs.
>
> Thirdly, we include a figure (Figure 4 in the revised manuscript) that visualizes the Jacobian of the encoding and the pull-back norm.
>
> Please also note that as a way to provide more intuitions, Appendix D visualizes the pull-back geometry by plotting the pull-back unit ball on a toy data manifold.

---

> ### Author Response · Authors · 2024-01-26
> **Part II. Applicability of methodology beyond TDA**
>
> > Weakness 2) The focus on PH and its specific applications might limit the paper's appeal to a broader audience. Its relevance is primarily confined to specialists in topological data analysis and related fields.
>
> > Weakness 3) While the paper situates itself well within its specific research domain, it could benefit from a broader contextualization in terms of its implications for other areas of data analysis or machine learning.
>
> In the paper we indeed focus on PH, but the methodology could be applied well beyond TDA.
>
> Please note that in Section 4 we applied our method to the deep neural network architecture PointNet, and we included a detailed discussion comparing the PointNet encoding and other three PH encodings in Section 4.2 (Figure 7 in the revised manuscript) and Section 4.3 (Figure 8 in the revised manuscript). Specifically, we have found that: "For PointNet encoding, the top eigenvector has a relatively strong alignment with *translation*" and "PointNet is robust under *convex* perturbations".
>
> This had been discussed in the Introduction of the original submission, and in the revised manuscript we highlight it more clearly:
>
> "We center our attention on a widely used PH representation known as the persistence image (PI) [1], which is an image-like representation of the input data in terms of multiscale homological features. Our methodology, however, extends to other PH representations and, more broadly, to any differentiable encoding whose representation space can be endowed with a Riemannian manifold structure. In our experiments, we illustrate this generality by applying our approach to the PointNet encoding, a benchmark deep learning model for point clouds."
>
> We believe investigating other encodings would be valuable future work, which can be facilitated by our method.
>
> &nbsp;
>
> [1] Adams, Henry, et al. "Persistence images: A stable vector representation of persistent homology." Journal of Machine Learning Research 18 (2017).

---

> ### Author Response · Authors · 2024-01-26
> **Part III. Additional experiments**
>
> > Weakness 4) While the experiments conducted are robust, additional validation across more diverse datasets and scenarios could strengthen the findings. This would ensure the generalizability of the research.
>
> > Requested Change 2) While the study's experimental validation, such as the analysis of brain artery trees, is robust, extending these experiments to a wider range of datasets would demonstrate the versatility of the proposed methods.
>
> The revised manuscript includes additional experiments. Firstly, we consider some more interesting scenarios for the brain artery data: rather than focusing only on the resolution and Gaussian variance of persistence images, we now also consider the less obvious parameter: the weight function. We introduce the beta weighting function which allows assigning weights to persistence intervals of different length. We study the effects of intervals of different length on detecting the *sex* feature (Figure 14 in the revised manuscript). Our result 1) strengthens the conclusion that pull-back norm is predictive for downstream task performance in classifying *sex* labels on brain artery tree data and 2) confirms persistence intervals with medium length are more important for this task, which is also observed in [1].
>
> Secondly, we consider the new real-world human body dataset to exemplify how our methodology facilitates investigating which *part* of the input data is the focus of PH encodings (Appendix G in the revised manuscript). We consider the 2-dimensional PI on Rips filtration and the beta weighting function for constructing PIs. We found that noising perturbations on certain body part, for example arms, can only be detected by shorter persistence intervals.
>
> &nbsp;
>
> [1] Bendich, Paul, et al. "Persistent homology analysis of brain artery trees." The annals of applied statistics 10.1 (2016): 198.

---

> ### Author Response · Authors · 2024-01-26
> **Part IV. Comparison with existing approaches**
>
> > Requested Change 3) A section comparing the paper's approach with other existing methods in topological data analysis, especially in terms of recognizing or ignoring data variations, could contextualize its efficacy and novelty.
>
> In the Introduction of the original submission, we already discussed performance-based testing as the main existing approach to assess to what extent PH recognizes the given perturbation or feature, where we discussed several works in topological data analysis looking into recognizing data variations, commenting for instance on the work in [1] and the work in [2]. In the experiments we also discuss the results from our method in the light of previous investigations related to the sensitivity of PH. Please let us know if you have any particular work in mind that we should add.
>
> Following your comment, in the revised Introduction we have still sought to make the comparison even more explicit and have also added a flowchart in Figure 1 to make the comparison more visible.
>
> Furthermore, please also notice that in section 5 we do employ a downstream classifier, namely the logistic regression model, on the PH representations. We have demonstrated that the pull-back norm of the gradient vector field of the *sex* feature is predictive for the performance of the logistic regression model in classifying the same feature (Figure 11 and Figure 14 in the revised manuscript).
>
> &nbsp;
>
> [1] Turkeš, Renata, et al. "Noise robustness of persistent homology on greyscale images, across filtrations and signatures." Plos one 16.9 (2021): e0257215.
>
> [2] Bubenik, Peter, et al. "Persistent homology detects curvature." Inverse Problems 36.2 (2020): 025008.

---

> ### Author Response · Authors · 2024-01-26
> **Part V. Broader implications**
>
> > Requested change 4) The paper would benefit from an expanded discussion on the implications of its findings in broader data science contexts, particularly in how PH encodings could influence machine learning models or data representation strategies.
>
> PH encodings already find numerous applications in data science, some of which we described in the introduction. Yet, TDA practitioners often report as one of the main challenges in practice the appropriate selection of a filtration and other parameters of PH. We suggest that our proposed methods can help streamline this process. We have demonstrated this for the case of selecting the parameters of PH.  In principle, one can push this further, for instance introduce further parameters into the PH computation pipeline and use our approach to optimize these.
>
> In the context of data representations, deciding what makes a good data representation is a bit of an art. Nonetheless, some of the most frequent desiderata are that the representation method "disentangles" key factors of variation and implements dimensionality reduction by discarding the irrelevant variations of the data. We regard our study as taking important steps precisely towards cataloguing these properties for an important family of data representation techniques as are PIs.
>
> Data representations are generally considered to be useful for interpreting data and identifying trends and relationships. Our work is motivated in a good part by the question: When and why can PIs be useful as a data representation technique? We have shown that we can identify which interpretable trends or factors of variation in the data are captured by PI encodings and quantify the degree to which PI encodings capture features of interest. An interesting prospect facilitated by our methods is the empirical evaluation of the types of features that PH can capture in different types of data sets. As we have illustrated, our methods can also be used to compare between different data representation techniques.

---

> ### Author Response · Authors · 2024-01-26
> **Minor comments**
>
> > Requested change 5) Incorporating detailed case studies where the proposed method is applied, especially in contexts different from the brain artery trees dataset, would provide practical insights into the method's application.
>
> We have included additional experiments on the human body data set. Please see our response [Part III: Additional experiments](https://openreview.net/forum?id=7yswRA8zzw&noteId=Il9jaSA4fe) for further details.
>
> &nbsp;
>
> > Requested change 6) A more in-depth analysis of potential limitations, such as computational demands or scenarios where the method might not perform optimally, would provide a balanced view of the approach.
>
> Thank you for pointing this out. In the Conclusions of the original submission, we end the paper addressing one of the main limitations - the proposed methodology can only work when the perturbation is small, and the codomain of the encoding must have a Riemannian structure. We have now also added a comment on the computational demands of these methods.

---

> ### Author Response · Authors · 2024-02-08
> **Regarding our previous responses**
>
> Thank you again for your comments on our initial submission, helping us in making the revisions outlined above. We have sought to systematically address all concerns and requested changes and believe this has strengthened the paper. Please be informed that we have made additional modifications and have now uploaded the latest version of the manuscript. We would kindly like to inquire if you have found the revisions adequately addressed your concerns or if you have any additional suggestions or feedback.

---

### Review · Reviewer_8VK4 · 2024-01-09

**Summary Of Contributions:**

Persistent Homology (PH) provides a way to extract topological information from structured object, such as point clouds represented as elements of $\mathbb{R}^{N \times D}$ ($N$ points in dimension $D$) by the mean of a _filtration_ which, in the case of point clouds, may be encoded as a map $f : \mathbb{R}^{N \times D} \times \mathbb{R}^D \to \mathbb{R}$.
One such example is the _Rips_ filtration defined as $f(X, x) \coloneqq \min_{i=1,\dots,N} |x - x_i|$, where $X = (x_1,\dots,x_N)$.

Each choice of filtration $f(X,\cdot)$ for a given object $X$ yields a topological descriptors called a _persistence diagram_ (PD). The space of PDs lacking linear structure, it is usual to embed these PDs in a Euclidean space $\mathbb{R}^n$ by the mean of a vectorization. The _persistence image_ (PI) embedding is one standard option to do so.

Eventually, one obtains a map $\Phi_f : \mathbb{R}^{N \times D} \ni X \mapsto \mathrm{PI}(\mathrm{PH}(f(X,\cdot)) \in \mathbb{R}^n$. A natural question, considered by this work, is then _given a perturbation of $X$, how is $\Phi_f(X)$ affected?_ Or, somewhat equivalently, what kind of information $\Phi_f(X)$ is ``encoding'' about $X$ given the choice of filtration $f$?

This map is known to be locally Lipschitz hence differentiable almost everywhere. Therefore, the natural idea considered in this work is to consider the Jacobian $J_f$ of $\Phi_f$ and quantify how a perturbation of $X$ encoded as a (``tangent'', if points belong to a manifold) vector fields $V$ are changing $\Phi_f$, which can directly be deduced from looking at the spectral composition of $J_f(V(X))$ and quantified using appropriated norms.

The current paper investigates this idea and illustrates it through various numerical experiments.

**Audience:**

Yes

**Broader Impact Concerns:**

No specific concerns in terms of ethical implications for the current work.

**Claims And Evidence:**

Yes

**Requested Changes:**

I would like to see an updated version of the work that would be much stronger on the formalism / theoretical side, bringing much more general insights and as such being more impactful and useful for the targeted audience.

I can admit that it is too much work for the current paper. In that case, I let the other reviewers and the editors assessing whether it is worth for publication in TMLR in its current state (up to minor modifications).

**Strengths And Weaknesses:**

# Global summary

This is a competent paper supported by nice experiments. However, my main feeling is that it is somewhat oversimplifying the problem it considers and only scratches the surface. I am not saying that the paper is inherently bad, but that it could be much better by bringing much more insights (especially from a theoretical viewpoint), formalizing and hopefully addressing important questions.

To me, the paper looks like ''interesting experimental findings'', while it could be ''bringing deep insights and important questions on the variational sturcture of the persistent homology map'', and the latter would be significantly more impactful and useful for the TDA community in my opinion.

(edit on 01/13 : removing backquotes that yielded improper formatting)

# Strengths

- Very nice illustrations!
- Overall well written (though some choices are disputable, see below).
- Extensive set of experiments.

# Major Weaknesses

While I like its main purpose, I feel that the paper is frustratingly superficial from a theoretical standpoint and does not bring proper insight on the topic it covers. Many non-trivial aspects or questions are put under the rug. For instance:
- The paper relies on an (arbitrary) embedding of PDs in an Euclidean space to avoid the much harder though much more insightful question _how to quantify the variation of $X \mapsto \mathrm{PH}(f(X,\cdot))$ for a given filtration $f$?_ This requires to equip the space of PDs with a differentiable structure (and understanding the impact of the non-smoothness of the map $\mathrm{PH}$). The (significant) benefit would be to get an analysis that would be independent of the choice of the vectorization.
- As said above, the map $X \mapsto \mathrm{PH}(f(X,\cdot))$ is not smooth (only differentiable a.e.), which means in practice that for a very small perturbation $V$, the generators of $\mathrm{PH}(X)$ and $\mathrm{PH}(X + V(X))$ can be quite different, i.e. eigenvectors of the Jacobian/Gram matrix may not be stable. As such I am not sure how the proposed interpretation is reliable.
- I would like having more insight on the relation between perturbations and filtrations. That would help the practitioner to address the question _how should I design a filtration $f$ that would be sensitive/unsensitive to a given type of perturbation $V$?_ or, conversely, _given this $f$, which are the perturbations $V$ that are detected/undetected by $f$?_ To me, the important point is thus to formalize, in a general abstract setting, a way to quantify the variation of $V \mapsto J_f(V(X))$. Typically, I would like to have an insight on _why_ the Rips filtration is good at reflecting ``convexification'' (this may be related to the notion of _convexity defect_), or to know _in advance_ that a given filtration is insensitive to some transformations (e.g. the Rips being insensitive to translation-rotation; though it is obvious in that particular case). That requires to consider the variations of the map-valued map $X \mapsto f(X,\cdot)$. For instance, the Rips PI is insensitive to translations and rotations because the map $X \mapsto \mathrm{dist}_X(\cdot)$ is equivariant under this action and the operator $\mathrm{PH}$ has its own invariances. To me, that's an important question raised by the paper, but not addressed nor even formalized.

# Minor comments

- In section 3, the choice of mixing the Manifold viewpoint and the Euclidean viewpoint on the space of point clouds is a bit confusing: its too sophisticated for the Euclidean case and too simplistic for the Manifold setting. E.g. Definition 1 and 2 "are for the case that $\mathcal{M}$ is a Euclidean space" : in that case, the notion of tangent space is not required, etc. On the other hand, as acknowledged by the authors, this definition is too simplistic for the actual manifold setting.
- Peyré et al., --> Should be Peyré and Cuturi. It is published in _Foundations and Trends in machine Learning_. Other citations may be checked as well.
- Not sure that considering the Wasserstein distance between persistence images (end of $\S$3.1) is meaningful/useful. If one use the standard Wasserstein distance (as defined in Optimal Transport), then one has to normalize the images, which is dubious in my opinion. If one use the Wasserstein distance as defined in TDA (using the diagonal as a trash bin), then why not simply doing that in the space of persistence diagrams directly? (And this boils down to the harder problem of computing variations in a metric space that I mention above)

---

> ### Author Response · Authors · 2024-01-26
> **Part I. Dependence on the choice of vectorization**
>
> > Weakness 1) The paper relies on an (arbitrary) embedding of PDs in an Euclidean space to avoid the much harder though much more insightful question how to quantify the variation of $X \mapsto \mathrm{PH}(f(X, \cdot))$ for a given filtration $f$? This requires to equip the space of PDs with a differentiable structure (and understanding the impact of the non-smoothness of the map PH). The (significant) benefit would be to get an analysis that would be independent of the choice of the vectorization.
>
> We acknowledge that the scope of this paper is less ambitious than quantifying the variation of $X \mapsto \operatorname{PH}(f(X, \cdot))$ *in general*, as we focus on persistence images.
> We agree investigating other PH representations, including PDs, is very interesting. At the same time, we think the investigation of PI is very interesting too, since these representations are widely used in applications, and certainly a good point of departure for future endeavors in that direction.
> Please note our methodology can be applied to any differentiable encoding whose output space can be equipped with a Riemannian manifold structure. This had been discussed in the Introduction and Conclusion of the original submission, and in the revised manuscript we highlight it more clearly:
>
> "We center our attention on a widely used PH representation known as the persistence image (PI), ... Our methodology, however, extends to other PH representations and, more broadly, to any differentiable encoding whose representation space can be endowed with a Riemannian manifold structure. In our experiments, we illustrate this generality by applying our approach to the PointNet encoding, a benchmark deep learning model for point clouds."
>
> "Another limitation of the proposed methodology is the assumption about the differentiability of the encoding, and the need for a Riemannian manifold structure for the representation space. For this reason, our methodology cannot be directly applied to analyze PH representations such as the most common persistence diagrams, since these cannot be endowed with a smooth structure [1]. We note, however, that a Riemannian framework for *approximated* PDs has been introduced in [2]."
>
> &nbsp;
>
> [1] J. Leygonie, S. Oudot, and U. Tillmann. A framework for differential calculus on persistence barcodes. Foundations of Computational Mathematics, 22:1069–1131, 2022. doi: https://doi.org/10.1007/s10208-021-09522-y
>
> [2] Anirudh, Rushil, et al, "A Riemannian framework for statistical analysis of topological persistence diagrams", Proceedings of the IEEE Conference on Computer Vision and Pattern Recognition Workshops (2016).

---

> ### Author Response · Authors · 2024-01-26
> **Part II. Non-smoothness of PH**
>
> > Weakness 2) As said above, the map $X \mapsto \mathrm{PH}(f(X, \cdot))$ is not smooth (only differentiable a.e.), which means in practice that for a very small perturbation $V$, the generators of $\mathrm{PH}(X)$ and $\mathrm{PH}(X+V(X))$ can be quite different, i.e. eigenvectors of the Jacobian/Gram matrix may not be stable. As such I am not sure how the proposed interpretation is reliable.
>
> Indeed the map $X\mapsto \operatorname{PH}(f(X,\cdot))$ is not smooth. However, in this work we focus on the map from point clouds to PIs. We have shown that this map is differentiable everywhere given the Riemannian manifold structure that we introduce on the space of point clouds (Appendix C). Specifically, we show that given any point cloud there exists a neighborhood such that the PI mapping is differentiable. When evaluating the pull-back norm of perturbation vectors, we do require the norm of the perturbation vectors to be small enough to guarantee the perturbed point cloud would still stay in this neighborhood. Hence, our investigation only focuses on local variations. We have highlighted this in the Limitations in the original (and revised) submission:
>
> "Our analysis is based on the structure of the Jacobian of the data encoding, which by nature focuses only on local variations of the input data. In future it will be interesting to further advance these methods in regard to non-local data variations, where synthetic notions of derivatives such as our empirical evaluation of the vector fields, and ideas such as the application of the iterated closest point method could serve as a point of departure. ..."
>
> Regarding the second part of your comment, we note that, under appropriate assumptions,  stability theorems for PH guarantee that for small perturbations $V,$ $\operatorname{PH}(X)$ and $\operatorname{PH}(X+V)$ remain close.

---

> > ### Comment · Reviewer_8VK4 · 2024-01-27
> > **-> Non-smoothness of PH**
> >
> > I may not have been clear, but my point was not that $\mathrm{PH}(X)$ and $\mathrm{PH}(X+V)$ may differ (here $\mathrm{PH}$ implicitly refers to the Vietoris-Rips filtration, and indeed in that case the difference is controlled by $V$), but that the _generators_, i.e. points in $X$ which are responsible for the presence of an interval in  $\mathrm{PH}(X)$ are unstable.
> >
> > Basically, the sentence that I may be failing to understand is the one appearing in appendix C :
> >
> > > Meanwhile, the filtration mapping is differentiable with respect to the coordinate of every point in $X$.
> >
> > To me, this is only guaranteed if the filtration takes distinct values on each simplex---which is more demanding than just considering the points distinct. In the VR filtration, if two distances $|x_i - x_j|$ and $|x_k - x_l|$ are equal (and happen to be critical), there is an ambiguity on which edge is the one creating (or destroying) a given interval. Sure, (i) this does not make the interval unstable of course, and (ii) this does not happen generically. Nonetheless, it means that _numerically_, small (but not infinitesimally small) perturbations of $X$ can yield different generators, hence different eigenvectors in the context of the paper (Figure 2).
> > And I stress that the "smallness" is not controlled in terms of $\min_{ij} |x_i - x_j|$ (i.e. how distinct the points are), but in terms of $\min_{ij,kl} ||x_i - x_j| - |x_k - x_l||$ (i.e. how distinct the _pairwise distances_ are).
> >
> > For instance, if one samples points _uniformly_ on a circle and compute the $H_1$ PD (and PI), which points should be considered as creating/destroying the $1$-dimensional cycle? What is the Jacobian in that case?
> >
> > I am not saying that what is done in the paper is incorrect. I am saying that I am not sure to understand in which ways the eigenvectors of the Jacobian are reliable.
> >
> > I may be missing something though.

---

> > ### Comment · Reviewer_8VK4 · 2024-01-27
> > **On the Riemanian structure on the space of point clouds**
> >
> > On a side remark, the fact that the Wasserstein-$2$ space exhibits a Riemanian-like structure is well-known and deeper than what is presented in this work; see for instance the Part II of Villani's book (Optimal transport : Old and New), or Chapter 8 in Ambrosio, Gigli and Savaré's book (Gradient flows: in Metric Spaces and in the Space of Probability Measures).
> >
> > Putting this specific construction in the appendix is appreciated as it may help the reader that would not be familiar with these notions, but it should not be considered as a novel contribution of the present work.

---

> ### Author Response · Authors · 2024-01-26
> **Part III. Relationship between perturbations and filtrations**
>
> > Weakness 3) I would like having more insight on the relation between perturbations and filtrations. That would help the practitioner to address the question how should I design a filtration $f$ that would be sensitive/unsensitive to a given type of perturbation $V$? or, conversely, given this $f$, which are the perturbations $V$ that are detected/undetected by $f$? To me, the important point is thus to formalize, in a general abstract setting, a way to quantify the variation of $V \mapsto J_f(V(X))$. Typically, I would like to have an insight on why the Rips filtration is good at reflecting "convexification" (this may be related to the notion of convexity defect), or to know in advance that a given filtration is insensitive to some transformations (e.g. the Rips being insensitive to translation-rotation; though it is obvious in that particular case). That requires to consider the variations of the map-valued map $X \mapsto f(X, \cdot)$. For instance, the Rips PI is insensitive to translations and rotations because the map $X \mapsto \operatorname{dist}_X(\cdot)$ is equivariant under this action and the operator $\mathrm{PH}$ has its own invariances. To me, that's an important question raised by the paper, but not addressed nor even formalized.
>
> Investigating the sensitivity of filtration (function) under perturbations is indeed *another* interesting problem, but in our work we chose to rather study the sensitivity of PH (or more precisely, the mapping from point clouds to PIs). We consider this to be a more interesting question since, invariance of a filtration function under perturbation implies invariance of PH (guaranteed by the stability theorems), but the reverse does not need to be true: the filtration might change under a pertubation, while the PH can remain invariant. Indeed, one could, for example, perturb a circle into a square: the filtration obviously changes, but 0- and 1-dim PH (with respect to the Rips filtration on point clouds, or binary or greyscale filtration on images) can remain the same. In other words, the invariance of a filtration under perturbation is a sufficient, but not necessary condition for invariance of PH.
>
> Regarding your remark about understanding the sensitivity of a filtration or PH to some transformations *in advance*, we say that this a challenging task, since the behavior strongly depends on the data at hand. Consider, for example, Gaussian noise and outliers can change 1-dimensional PH of a circle more than that of a disk. In the revised manuscript, we acknowledge the data-specificity of the insights in the Introduction in the following way:
>
> "We note that the insights about the (PH) encodings obtained through our approach depend on the specifics of the data set. Nonetheless, by evaluating different datasets one may be able to draw certain conclusions that hold with some generality: for instance, conclude that a particular encoding captures a particular feature in datasets of a particular type. This is an interesting prospect that can be facilitated by our proposed approach."
>
> We also highlight this once again in Section 4: "... we emphasize that this section is dedicated to investigating 'what is recognized' by PH encodings *within specific datasets*".

---

> > ### Comment · Reviewer_8VK4 · 2024-01-27
> > **About filtrations**
> >
> > > Investigating the sensitivity of filtration (function) under perturbations is indeed another interesting problem, but in our work we chose to rather study the sensitivity of PH (or more precisely, the mapping from point clouds to PIs).
> >
> > To me, perturbing the point point _is_ (one way of) perturbing the filtration. It is just that the filtration is parametrized by the point cloud; e.g. $X \mapsto \mathrm{dist}_X(\cdot)$ which yields the Cech filtration. Saying that one studies the sensitivity of PH (or PI) with respect to the perturbation of the point cloud _rather than_ the perturbation of the filtration is improper in my opinion.
> >
> > In any case, I agree with you that understanding all of this is a very difficult problem that goes way beyond the scope of this paper. Nonetheless, my point was that the observations provided by the paper (e.g. Rips being sensitive to "convexification") are surely interesting, but not very insightful in that right now I have no clue on _why_ this should be the case ; which is to me the important underlying question.
> > Right now, if I pick some filtration or some perturbation that has not been covered by the current work, I have no idea on what to expect and I such, I do not really know how to make use of this work.

---

> ### Author Response · Authors · 2024-01-26
> **Minor comments**
>
> > In section 3, the choice of mixing the Manifold viewpoint and the Euclidean viewpoint on the space of point clouds is a bit confusing: its too sophisticated for the Euclidean case and too simplistic for the Manifold setting. E.g. Definition 1 and 2 "are for the case that is a Euclidean space" : in that case, the notion of tangent space is not required, etc. On the other hand, as acknowledged by the authors, this definition is too simplistic for the actual manifold setting.
>
> Please note our method can be applied to any differential mapping between Riemannian manifolds. When introducing our method in Section 3, we do not want to lose this generality. Hence, we keep the manifold notation for tangent vectors and tangent spaces. Meanwhile, for better readability, we consider the Euclidean case for the input space when defining perturbation vector fields and gradient fields. However, we have provided rigorous manifold-style definitions for perturbation vector fields and gradient fields in the Appendix (Definition 11 and Definition 13 in Appendix B).
>
> We have discussed this in Section 3 of the original submission:
>
> "For simplicity of presentation, in the following we treat $\mathcal{M}$ as an Euclidean space. Nonetheless, our discussion is consistent with the manifold structure and also applies to other types of data with an appropriate manifold structure (see Appendix B for details).",
>
> and
>
> "Definition 1 and Definition 2 are for the case that $\mathcal{M}$ is a Euclidean space. We provide definitions of perturbation vector field and gradient vector field in the case of general Riemannian manifolds in Appendix B.3."
>
> &nbsp;
>
> > Peyré et al., should be Peyré and Cuturi. It is published in Foundations and Trends in Machine Learning. Other citations may be checked as well.
>
> Done, thank you for noticing this! We made sure to check all the arXiv pre-prints and working papers.
>
> &nbsp;
>
> > Not sure that considering the Wasserstein distance between persistence images (end of Section 3.1) is meaningful/useful. If one use the standard Wasserstein distance (as defined in Optimal Transport), then one has to normalize the images, which is dubious in my opinion. If one use the Wasserstein distance as defined in TDA (using the diagonal as a trash bin), then why not simply doing that in the space of persistence diagrams directly? (And this boils down to the harder problem of computing variations in a metric space that I mention above).
>
> Indeed the choice of the distance between PIs depends on specific interests. We now only mention Wasserstein distance (between PI) in the footnote and highlight its flexibility:
>
> "Here again, other choices of the metric are possible.(footnote 4)
>
> Footnote 4: This can be of interest to try to establish more general stability results for PIs. An example would be a Wasserstein distance between persistence images assigning an appropriate cost to  $b$ and $l$ directions."
>
> Regarding the comment about persistence diagrams, please see our response to your [first remark](https://openreview.net/forum?id=7yswRA8zzw&noteId=hZ07S4SkZ4) above.

---

> ### Author Response · Authors · 2024-02-07
> **Additional comments about filtrations**
>
> > To me, perturbing the point point is (one way of) perturbing the filtration. It is just that the filtration is parametrized by the point cloud; e.g. which yields the Cech filtration. Saying that one studies the sensitivity of PH (or PI) with respect to the perturbation of the point cloud rather than the perturbation of the filtration is improper in my opinion.
>
> We think that studying the relationship between perturbations of point clouds and perturbations of filtrations, or perturbations of filtrations more generally, is an interesting topic. However, we disagree that studying the sensitivity with respect to perturbations of point clouds is improper. Point clouds are a natural domain for PH and perturbations of point clouds are a natural way to induce perturbations of filtrations. Consider in particular that many stability problems and stability results in TDA are formulated in terms of bounds on distances between persistence diagrams with respect to suitable distances between perturbed point clouds.
>
> &nbsp;
>
> > Nonetheless, my point was that the observations provided by the paper (e.g. Rips being sensitive to "convexification") are surely interesting, but not very insightful in that right now I have no clue on why this should be the case ; which is to me the important underlying question. Right now, if I pick some filtration or some perturbation that has not been covered by the current work, I have no idea on what to expect and I such, I do not really know how to make use of this work.
>
> About the second point, concerning other perturbations beside those inspected in our manuscript, please observe that one can use the methodology we have presented to also study other filtrations or perturbations: one can directly compute the average pull-back norm of any perturbation vector field to quantify the sensitivity of the PH encoding to that perturbation.
>
> To demonstrate this more clearly, we have implemented a Python function that enables users to compute the sensitivity of the PH encoding (whose output is 1-dimensional PI constructed on Vietoris-Rips filtration) to an *arbitrary* perturbation. The inputs to the function are simply a list of point clouds, a list of perturbed point clouds, and the parameters for the encoding (e.g., maximal edge length and resolution for PI). The function then returns the average pull-back norm of the perturbation vector field over the given data set. We have updated the repository by including this function in the file "pbn_perturb_func.py" under the folder "Identifying-what-is-recognized". Meanwhile, we will add a pointer to this update in Section 4 of the revised manuscript:
>
> "We provide a numerical function in [repository](https://anonymous.4open.science/r/persistent-homology-0915) that allows automatic computation for the average pull-back norm with respect to Vietoris-Rips filtration for any given perturbation on a given data set. Note that it is not necessary to have the explicit form for perturbation $\pi: \mathcal{M} \rightarrow \mathcal{M}$, since we only require the set of perturbed point clouds, i.e., $\pi(X)$ for all $X$ in the data set."

---

> ### Author Response · Authors · 2024-02-07
> **Additional comments about non-smoothness of PH**
>
> > I may not have been clear, but my point was not that $\mathrm{PH}(X)$ and $\mathrm{PH}(X+V)$ may differ (here $\mathrm{PH}$ implicitly refers to the Vietoris-Rips filtration, and indeed in that case the difference is controlled by $V$ ), but that the generators, i.e. points in $X$ which are responsible for the presence of an interval in $\mathrm{PH}(X)$ are unstable.
>
> > Basically, the sentence that I may be failing to understand is the one appearing in appendix C: "Meanwhile, the filtration mapping is differentiable with respect to the coordinate of every point in $X$."
>
> > To me, this is only guaranteed if the filtration takes distinct values on each simplex---which is more demanding than just considering the points distinct. In the VR filtration, if two distances $\left|x_i-x_j\right|$ and $\left|x_k-x_l\right|$ are equal (and happen to be critical), there is an ambiguity on which edge is the one creating (or destroying) a given interval. Sure, (i) this does not make the interval unstable of course, and (ii) this does not happen generically. Nonetheless, it means that numerically, small (but not infinitesimally small) perturbations of $X$ can yield different generators, hence different eigenvectors in the context of the paper (Figure 2). And I stress that the "smallness" is not controlled in terms of $\min _{i j}\left|x_i-x_j\right|$ (i.e. how distinct the points are), but in terms of $\min _{i j, k l}|| x_i-x_j|-| x_k-x_l||$ (i.e. how distinct the pairwise distances are).
>
> > For instance, if one samples points uniformly on a circle and compute the $H_1$ PD (and PI), which points should be considered as creating/destroying the 1-dimensional cycle? What is the Jacobian in that case?
>
> > I am not saying that what is done in the paper is incorrect. I am saying that I am not sure to understand in which ways the eigenvectors of the Jacobian are reliable.
>
> Thank you for pointing this out! You are right that the PH encoding is only generically differentiable. We will be uploading a revised manuscript very soon (aiming by tomorrow), in which we include a formal analysis of the differentiability for PH encodings.
>
> In particular, we show that the PH encoding is differentiable at point clouds where the filtration defines a strict total order. For Vietoris-Rips filtration and height filtrations, we have shown that the PH encoding is generically differentiable, i.e., the set where PH encoding is differentiable is dense in $\mathcal{M}$. A similar result had already been considered in the work of [1].
>
> Regarding the stability of eigenvectors, please note that whereas the eigenvectors of the Jacobian at a specific point cloud may be unstable, when making general statements about pullback norm and alignment, we are referring to quantities that are computed as expectation values over a distribution of point clouds. In the revision of the manuscript we are including additional experiments that show the average pull-back norm quickly converges to a stable value once a certain amount of data is considered.
>
> We acknowledge that stability of the eigenvectors poses a problem when considering certain specific point clouds, and we will explain this issue in the revised version of the paper, using the example of points evenly-spaced on a circle suggested by the reviewer. If we take evenly-spaced points on a circle, then it is true that the information given by the eigenvectors of the Jacobian would be unreliable. However, if we perturb this point cloud even slightly, then we expect the information contained in the Jacobian to be reliable, because the set of points clouds for which the encoding is differentiable is dense in $\mathcal{M}$. Note in particular that this issue would generally not arise for points sampled uniformly at random from a circle.
>
> Numerical approximation errors can also create issues even for points in generic position. Numerical instability due to rounding errors is a problem that one often needs to take into account in data analysis. In our particular case a situation may arise where two edges in the Vietoris-Rips complex turn out to be rounded to have the same filtration value. We are adding a word of caution about this in the revised manuscript, and we thank the reviewer for raising this important aspect.
>
> &nbsp;
>
> [1] J. Leygonie, S. Oudot, and U. Tillmann. A framework for differential calculus on persistence barcodes. Foundations of Computational Mathematics, 22:1069–1131, 2022. doi: https://doi.org/10.1007/s10208-021-09522-y

---

> ### Author Response · Authors · 2024-02-07
> **Additional comments about the Riemannian structure on the space of point clouds**
>
> > On a side remark, the fact that the Wasserstein-2 space exhibits a Riemanian-like structure is well-known and deeper than what is presented in this work; see for instance the Part II of Villani's book (Optimal transport : Old and New), or Chapter 8 in Ambrosio, Gigli and Savaré's book (Gradient flows: in Metric Spaces and in the Space of Probability Measures).
>
> > Putting this specific construction in the appendix is appreciated as it may help the reader that would not be familiar with these notions, but it should not be considered as a novel contribution of the present work.
>
> We thank the reviewer for sharing these references. Indeed, the specific construction is added for the convenience of interested readers who might not be familiar with the subject. We have added pointers to these more detailed works in Section 3 in the revised manuscript: "For a detailed discussion regarding the Riemannian structure and differential calculus on a Wasserstein space we refer the reader to [1; Chapter 8] and [2]".
>
> &nbsp;
>
> [1] Ambrosio, Luigi, Nicola Gigli, and Giuseppe Savaré. Gradient flows: in metric spaces and in the space of probability measures. Springer Science \& Business Media, 2005.
>
> [2] Villani, Cédric. Optimal transport: old and new. Vol. 338. Berlin: Springer, 2009.

---

### Author Response · Authors · 2024-01-26
**Summary of main improvements**

Dear reviewers,

&nbsp;

Thank you so much for taking the time to go through our paper and for your constructive feedback. We are happy that the relevance of the paper has been recognized ("significant finding/contribution", "the study stands out", "offering a new lens", "addresses important TDA questions and brings value to both TDA and machine learning community", "competent paper... well written, supported by an extensive set of experiments and very nice illustrations"). Below we respond to each of the critical comments in detail. The most important changes are as follows:

- Abstract, Introduction, and Conclusions: To better position our work, and to address some of the main criticisms raised in the reviews, we have significantly rewritten the Abstract, Introduction, and Conclusions:

  - We discuss in detail the advantages of our proposed approach that offers insights that are more intrinsic to the underlying problem compared with performance-based testing that are contingent upon the choice of an additional classifier and necessitate training and hyperparameter tuning. This is also illustrated in the new flowchart that sketches both the proposed pull-back geometry approach and the performance-based testing (Figure 1).
  - We discuss in greater detail the applicability of the method beyond persistence images and indeed TDA.
  - We highlight the data-dependence of the findings, to tone down the generality of the claims perceived in the initial submission.
  - We comment on the computational challenges of the approach.


- Clarification of the methodology: We include a schematic diagram that visualizes the proposed approach (Figure 1), an illustration of the pipeline for computing PIs (Figure 2), as well as a visualization for the Jacobian map and the pull-back norm (Figure 4).

- Additional experiments on less obvious parameters: Beside the PI parameters resolution and variance, in Section 5 we now investigate the effect of the PI weight function on the pull-back geometry on the brain artery data. We demonstrate once again that the pull-back norm is predictive of the downstream task performance, and that the persistence intervals of medium length are crucial for this application. This goes against the popular belief that long intervals are the most important, at the same time confirming the findings from the original paper that employs PH on this data set.

- Additional experiments on other data sets: We perform additional experiments on another real-world data set of 3-dimensional point clouds of human bodies. We demonstrate how our method can also help investigate which parts of the input point cloud are the focus of the PH encoding.
We found that for 2-dimensional PI on Vietoris-Rips filtration, noising perturbations on certain body part, for example arms, can only be detected by shorter persistence intervals.

- Additional analysis of the differentiability for PH encodings: We present a formal analysis of the differentiability for PH encodings in Appendix C. For Vietoris-Rips filtration and height filtration, we show that the PH encoding is generically differentiable, i.e., differentiable on a dense subset in the input space. We further incorporate additional experiments to illustrate the insights and computational results gained through method are stable due to the generic differentiability of PH encodings.

&nbsp;

We believe these changes have made the paper stronger and thank you for helping us in doing so. We hope to have addressed your concerns and that you will find the revised manuscript suitable for publication. We remain attentive and available for further discussions, questions and suggestions.

---

### Decision · Action_Editor_KUTy · 2024-02-22

**Recommendation:** Accept as is

**Comment:**

The paper's main claim and contribution are based on the perturbation analysis and the tools required to obtain novel insights. All the reviewers, including this AE, agreed that this is a valuable contribution to the community. However, there were some concerns, which I mention below.

1. One reviewer suggested that it would be interesting to evaluate how the embedding Jacobian satisfies the general goals of persistence homology: capturing holes of different dimensions in a way that is robust to rotations and minor noise.
2. Another reviewer noted the difficulty in deriving good insights, while also acknowledging the advantages of this work.

The paper has done a fair amount of revision to accommodate the concerns.

Overall, we have decided to **accept** the work.

**Audience:**

Yes, it would be interesting for a section of TMLR's audience.

**Claims And Evidence:**

The paper is on persistent homology encodings, which are methods to create representations of data based on their topological features. The paper proposes a novel methodology to investigate the properties and interpretation of these encodings using the pull-back geometry that they induce on the data manifold. The paper also presents some experimental results to demonstrate the insights gained by this approach.